# Diffusion Classifiers Understand Compositionality, but Conditions Apply

**Yujin Jeong***
TU Darmstadt & hessian.AI
yujin.jeong@tu-darmstadt.de

**Arnas Uselis***
University of Tübingen
arnas.uselis@uni-tuebingen.de

**Seong Joon Oh**
University of Tübingen

**Anna Rohrbach**
TU Darmstadt & hessian.AI

## Abstract

Understanding visual scenes is fundamental to human intelligence. While discriminative models have significantly advanced computer vision, they often struggle with compositional understanding. In contrast, recent generative text-to-image diffusion models excel at synthesizing complex scenes, suggesting inherent compositional capabilities. Building on this, zero-shot diffusion classifiers have been proposed to repurpose diffusion models for discriminative tasks. While prior work offered promising results in discriminative compositional scenarios, these results remain preliminary due to a small number of benchmarks and a relatively shallow analysis of conditions under which the models succeed. To address this, we present a comprehensive study of the discriminative capabilities of diffusion classifiers on a wide range of compositional tasks. Specifically, our study covers three diffusion models (SD 1.5, 2.0, and, for the first time, 3-m) spanning 10 datasets and over 30 tasks. Further, we shed light on the role that target dataset domains play in respective performance; to isolate the domain effects, we introduce a new diagnostic benchmark SELF-BENCH comprised of images created by diffusion models themselves. Finally, we explore the importance of timestep weighting and uncover a relationship between domain gap and timestep sensitivity, particularly for SD3-m. To sum up, diffusion classifiers understand compositionality, but conditions apply! Code and dataset are available at https://github.com/eugene6923/Diffusion-Classifiers-Compositionality.

Figure 1: **Overview of our findings.** *Finding I*: Diffusion models can perform compositional discrimination reasonably on real images, but underperform CLIP, especially on counting tasks (§5.2). *Finding II*: Diffusion models can understand (through classification) the images they can generate (§5.3). *Finding III*: Timestep reweighting improves discrimination by reducing the domain gap between generated and real data (§5.4).

---

*Equal contribution

39th Conference on Neural Information Processing Systems (NeurIPS 2025) Track on Datasets and Benchmarks.

# 1 Introduction

Models like Stable Diffusion [8, 41] have been trained on billions of image-text pairs and can generate highly detailed and coherent images that match textual descriptions. Their ever-increasing ability to generate complex compositional scenes [17, 11] suggests they have developed a strong understanding of image-text relationships and can effectively align visual and textual concepts. Diffusion models are trained with pixel-wise supervision, so they may be less prone to learn shortcuts, compared to discriminatively-trained models like CLIP [38], which often are insensitive to word order [63, 16, 7], struggle with spatial relationships, counting [50, 34], and compositions [49, 27, 52, 53]. It is thus natural to ask: can we transfer the compositional capabilities of generative models to discriminative compositional tasks?

There is growing interest in leveraging the strong generative capabilities of diffusion models for broader discriminative tasks such as classification, shape-texture bias, and depth estimation. Two main approaches have emerged: one line of work treats diffusion models as feature extractors, training task-specific classifiers in a supervised manner [64, 46]. The other approach, known as Diffusion Classifiers, repurposes diffusion models for zero-shot classification using their original training objective [29, 28, 6, 13, 20]. Notably, the latter has outperformed CLIP on compositional benchmarks like CLEVR [23] and Winoground [49], which require reasoning over multiple objects and attributes. However, existing studies are limited in scope, often relying on a small number of benchmarks and lacking a systematic analysis.

A recent work, the Generative AI Paradox [58], explores a key open question: whether strong generation implies strong discrimination. It shows that even if the model can generate, it may not understand, highlighting the disconnect between generative and discriminative capabilities. However, this analysis involves separate models for generation (e.g., Midjourney [19]) and discrimination (e.g., CLIP [38], OpenCLIP [18]) in image classification scenarios, making it difficult to directly assess the relationship between the two. In contrast, diffusion classifiers offer a direct way of probing the generative-discriminative connection by using the same model for both generation and discrimination.

To this end, we formulate three hypotheses to understand when and why Diffusion Classifiers succeed or fail, focusing on: i) diverse, large-scale compositional settings, ii) visual domain gap, and iii) the effect of different timesteps on classification. Our corresponding findings are illustrated in Figure 1.

**Hypothesis 1: Diffusion models' discriminative compositional abilities are better than CLIP's.** This is inspired by the findings in prior works. We conduct an extensive evaluation with three diffusion models, including a new SD3-m [8] model, on ten compositional benchmarks (**covering 33 subset tasks** — a scale not previously explored in this context) spanning four broad task categories (*Object*, *Attribute*, *Position*, and *Counting*). We find that diffusion models often outperform CLIP-based discriminative models, particularly in reasoning over spatial relations. However, they also exhibit *notable weaknesses, e.g., in counting tasks. Interestingly, SD3-m, despite superior generative compositionality, achieves lower discriminative accuracy (39%) compared to earlier versions (43%) in our analysis*. We shed light on this in the following.

**Hypothesis 2: Diffusion models understand (through classification) what they generate.** To explore the relationship between diffusion models' generative and discriminative abilities, we introduce SELF-BENCH, a diagnostic benchmark consisting of model-generated images. The idea is to isolate the image domain and assess the models' ability to understand images most "familiar" to the model. We find that diffusion classifiers perform well "in-domain" (i.e., when evaluated on data generated by the same model). However, their "cross-domain" performance (i.e., on data generated by a different diffusion model) drops significantly, especially for SD3-m. This highlights the domain gap (e.g., data distribution) as one of the critical factors. In in-domain scenarios, we observe a positive correlation between generative and discriminative compositional performance, suggesting that *stronger generative models can transfer their compositional knowledge to discrimination—but only when they can generate the target domain*.

**Hypothesis 3: The domain gap can be mitigated by timestep weighting.** Lastly, we investigate how diffusion timesteps impact discriminative performance by optimizing timestep weights for downstream tasks. Prior work has explored how diffusion models generate images through a structured progression across timesteps [30, 55, 26, 65, 56]. However, in diffusion classifiers, either uniform timesteps or fixed timestep weightings are typically used across all datasets and models—an area that remains underexplored. In contrast, we find that SD3-m is particularly sensitive to timestep

selection. Our results show that even low-shot timestep tuning (using just 5% of the target dataset can significantly mitigate the performance drop of SD3-m on real-world compositional benchmarks). We hypothesize that SD3-m's timestep sensitivity is closely linked to its susceptibility to the domain gap. To further examine this relationship, we incorporate CLIP-based image encoders to quantify visual similarity between domains, and analyze how it correlates with optimal timestep weighting. We find that *timestep weighting is especially effective in scenarios with large domain gaps*.

## 2 Related work

**Diffusion models.** Generative models have demonstrated impressive performance in producing realistic images [39, 60, 5, 61], videos [4, 66, 21], and audio [9, 32, 22]. In particular, text-to-image diffusion models [15] iteratively refine images conditioning on text prompts by adding and removing noise, achieving remarkable quality. A widely used open-source example is Stable Diffusion (SD) [41]. Earlier SD versions [41] are based on a UNet [42] backbone, incorporating ResNet blocks [12] and attention mechanisms [54]. With the rise of transformer-based designs [54], the latest Stable Diffusion 3 series [8] adopts this architecture, further boosting performance. It also introduces new noise sampling strategies for training Rectified Flow models [33, 1, 31]. Our analysis focuses on Stable Diffusion versions 1.5, 2.0, and 3-m, examining their architectural evolution and performance across scales.

**Compositionality in text-to-image models.** Text-to-image generation models, such as diffusion models, are hypothesized to have the capability to generate combinations of objects that were not present in the training data [49, 35]. Later versions of diffusion models, such as Stable Diffusion 3, exhibit improved generative capabilities and can produce scenes with greater compositional complexity [8, 59]. Recent benchmarks, such as CompBench [17] and GenEval [11], confirm this trend. In this work, we explore diffusion models to gain a deeper understanding of compositionality across various tasks, using existing compositional discriminative benchmarks and our newly introduced SELF-BENCH, which consists of images generated by the diffusion models themselves.

**Diffusion classifiers.** Recent studies have explored zero-shot classification using diffusion models' denoising process directly [29, 6, 28]. Li *et al.* [29] introduce the Diffusion Classifier with an adaptive evaluation strategy, demonstrating superiority over CLIP RN-50 [38]. Clark *et al.* [6] propose a universal timestep weighting function, showing effectiveness on attribute binding tasks (e.g., CLEVR [23]). Diffusion-ITM [28] adapts diffusion models for image-text matching, enabling both text-to-image and image-to-text retrieval, and introduces GDBench—a benchmark with seven complex vision-and-language tasks—where it outperforms CLIP baselines. These methods share a common foundation but differ in weighting and sampling strategies. Beyond zero-shot classification, few-shot approaches [62] leveraging diffusion models have also been proposed. Discffusion [13] enhances discrimination using cross-attention maps and LSE Pooling [3], focusing on few-shot learning but applicable to zero-shot settings as well. More recently, Gaussian Diffusion Classifier [37] was proposed as a one-shot or zero-shot method, using features from DINOv2 to improve efficiency. In this work, we primarily study the vanilla zero-shot Diffusion Classifier [29] to better understand its discriminative capabilities.

## 3 Methodology

In this section, we first discuss the prerequisites for diffusion classifiers. We then detail our approach to turning Stable Diffusion 3-m [8] into a classifier; we are the first to explore this, to the best of our knowledge. Last, we describe our approach to learning the optimal weighting function for the diffusion classifiers on given test data.

Given a dataset $\mathcal{D}_N = \{(\mathbf{x}_1, c_1), \dots, (\mathbf{x}_n, c_N)\}$ of $n$ images, where each image $\mathbf{x}_i \in \mathbb{R}^{H \times W \times 3}$ is labeled with a class $c_i \in \{1, \dots, K\}$, we aim to learn a classifier that can effectively handle compositional classification tasks. In practice, we work with latent representations $\mathbf{z} \in \mathbb{R}^d$ by encoding the images using a pretrained autoencoder model.

### 3.1 Preliminaries: diffusion classifiers

Diffusion models [48, 15] are generative models that learn to gradually denoise by reversing a forward diffusion process. For an image-text pair $(\mathbf{x}, \mathbf{c})$, where $\mathbf{x}$ is first encoded into a latent representation $\mathbf{z}$

using a pretrained autoencoder, the core training objective for diffusion models is

$$\mathcal{L}(\mathbf{z}, \mathbf{c}) = \mathbb{E}_{t, \boldsymbol{\epsilon}} \left[ w_t \left\| \boldsymbol{\epsilon} - \epsilon_\Theta(\mathbf{z}_t, t, \mathbf{c}) \right\|^2 \right], \tag{1}$$

where $w_t$ are timestep weights, $\epsilon_\Theta$ is a neural network that predicts the noise $\boldsymbol{\epsilon}$ added to the latent $\mathbf{z}$ at timestep $t \in T$, and $\mathbf{c}$ is a conditioning text prompt. Unless stated otherwise we draw $t \sim$ Uniform($[0, 1]$). This loss is related to the ELBO of the conditional likelihood $p(\mathbf{z}|y)$, where $y$ is the class label, which allows us to use diffusion models for classification, as shown in [29, 28, 6]:

$$\tilde{y} = \arg\max_{y_k} p(y = y_k \mid \mathbf{z}) = \arg\max_{y_k} \log p(\mathbf{z} \mid y = y_k),$$

where the likelihood is estimated using diffusion models through ELBO, approximated by Eq. (1), with conditioning $\mathbf{c}$ representing specific class label $y_k$. In practice, we approximate the expectation in Eq. (1) via Monte Carlo sampling using $T_s$ timesteps by considering fixed timesteps and noises. That is, we assume a fixed set $S = \{(t_j, \boldsymbol{\epsilon}_j)\}_{j=1}^{T_s}, t_j := j/T_s$ with which we compute the expectation.

**Learning the weighting function.** In diffusion models, different timesteps capture varying levels of information [30, 55]. This hierarchical information processing is crucial for compositional tasks, where both global structure (e.g., object relationships) and local details (e.g., attributes) matter.

Recently, [6] has explored universal timestep weighting in discriminative (yet non-compositional) settings. While we adopt several components from their approach, our setting and low-shot smoothing strategy differ. They rely on computationally expensive, high-variance classification estimates. For instance, they assume 100 trials for a single image-text pair. In contrast, we use fixed timesteps and noise to reduce variance in prediction [29] when computing the reconstruction target in Equation (1). We provide details in A.2.

The weighting function $w_t$ can be parameterized in two ways: (a) a piecewise constant function $w_t = v_{\lfloor t \times T_s \rfloor}, \quad t \in S_0$, where we learn individual weights $v_0, \ldots, v_{T_s-1}$, and $S_0$ denotes the set of timesteps; (b) alternatively, to enforce smoothness, as a $p$-degree polynomial $w_t = \sum_{i=0}^{p} a_i t^i, \quad t \in S_0$; (a) is generally used for achieving the upper-bound performance. However, in the low-shot learning setting (5% of the full training set), we use (b) to prevent overfitting.

## 3.2 SD3-m as a classifier

SD3-m is a rectified flow model [33, 1] trained with a conditional flow matching (CFM) loss [31], which differs from the standard diffusion objectives used in SD1.5 and SD2.0. As a result, we cannot directly apply the same classifier objective used in earlier versions. By reparameterizing the CFM objective as a noise-prediction loss (see, e.g., [8]), we can obtain

$$\mathcal{L}_{\mathrm{RF}}(\mathbf{x}_0) = \mathbb{E}_{t, \boldsymbol{\epsilon}}[w_t \left\| \epsilon_\Theta(\mathbf{z}_t, t, \mathbf{c}) - \boldsymbol{\epsilon} \right\|^2] \tag{2}$$

Using this formulation, we can use SD3 as classifiers, despite its different underlying architecture. The only difference lies in the weighting function $w_t$, which for SD3 follows a logit-normal distribution rather than the uniform weighting used in SD1.5/2.0. However, we empirically find that uniform weighting performs better for classification. Details are given in Section A.

## 4 Self-Bench: a diagnostic benchmark

As shown in Figure 2 (left), existing benchmarks are diverse in terms of domains. However, we empirically observe that the generation results of diffusion models are not as diverse, having a preferred "native" style. For example, in SD3-m, objects are usually well-centered/focused and have a glossy aesthetic touch; images are in high resolution (see Figure 2, right). Additional examples in Figure D.3 (Supplemental) further support that SD3-m consistently produces images with a similar style.

This raises the question: can diffusion classifiers understand images from different domains in discriminative scenarios? (Note that we use the word "domain" in a fairly relaxed sense.) Moreover, what is the best possible performance on in-domain data? This relates to our Hypothesis 2, that the domain plays a critical role in discriminative performance.

To answer these questions, we try to isolate the domain effect and compare the performance across in-domain and out-of-domain scenarios. Here, we *posit that if a diffusion model can generate images*

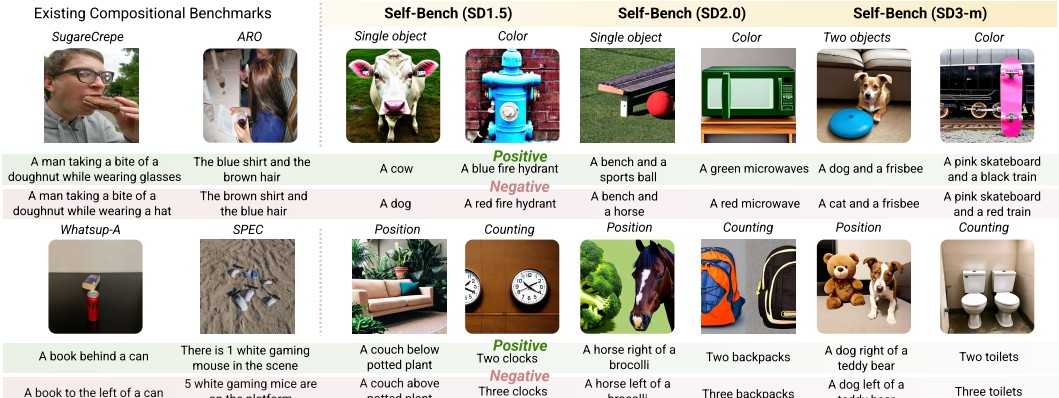

Figure 2: **Examples of standard benchmarks vs. SELF-BENCH.** Each benchmark is categorized into four broad task groups: *Object*, *Attribute*, *Position*, and *Counting*. Each group consists of one or more tasks, and we present one example per task for illustration. We indicate positive / negative captions, where the task involves matching the positive caption with its corresponding image. Notably, standard benchmarks and SELF-BENCH feature domain distinctions, incorporating the factors like style, resolution, and object scale.

*of a certain type, it can also discriminate them.* Therefore, we define *in-domain* as the data that diffusion models can generate. Namely, we propose SELF-BENCH, a benchmark for evaluating diffusion classifiers on diffusion models' own generated data.

**Diagnosing with SELF-BENCH.** We construct and evaluate the benchmark as follows (see Figure 3):

**1. Prompt collection.** We use text prompts from *GenEval* [11], a benchmark for compositional *generation*. GenEval includes 80 object classes and six task types: *Color*, *Color Attribution*, *Counting*, *Single Object*[2], *Two Objects*, and *Position*.

**2. Image generation.** For each prompt, we generate four images using SD1.5, SD2.0, and SD3-m (with guidance scale 9.0). We manually filter out failed generations.

**3. Discriminative prompt construction.** For each image, we keep the original generation prompt as the positive and create negative prompts. For example, for the *Position* task, if the original prompt is "a parking meter **left of** a teddy bear," we construct three additional negative prompts using other predefined spatial relations ("right of," "above," and "below").

**4. Evaluation.** We evaluate how well diffusion classifiers can match generated images with the correct prompts among distractors.

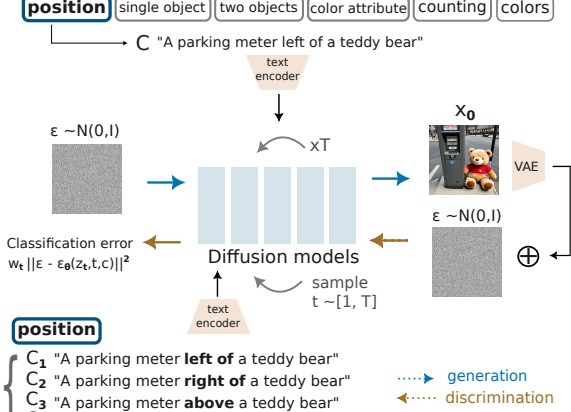

Figure 3: **Diagnosing with SELF-BENCH.** (i) Using Geneval's prompts from six categories, generate images. (ii) For each generated image, create discriminative tasks within its type from the prompts used in the generation process. (iii) Given the generated images (filtered by humans) and the discriminative tasks, benchmark the diffusion classifier.

**Filtering.** The generation process may produce failures, such as ambiguous (e.g., with over half of an object missing or mixed colors) or incorrect images. Thus, we define two sets: `Full`, containing all generated images, and `Correct`, the filtered high-quality subset. Three human annotators evaluate each image,

Table 1: **SELF-BENCH Statistics**: For each task, we show the number of images in `Full` (F) and `Correct` (C) sets.

| Task | Single Obj. | | Two Obj. | | Colors | | Color Attrib. | | Position | | Counting | |
|---|---|---|---|---|---|---|---|---|---|---|---|---|
| Filter | F | C | F | C | F | C | F | C | F | C | F | C |
| SD1.5 | 320 | 271 | 396 | 105 | 376 | 219 | 400 | 18 | 400 | 6 | 320 | 98 |
| SD2.0 | 320 | 271 | 396 | 129 | 376 | 263 | 400 | 36 | 400 | 19 | 320 | 111 |
| SD3-m | 320 | 314 | 396 | 306 | 376 | 314 | 400 | 252 | 400 | 113 | 320 | 230 |
| Total | 960 | 856 | 1188 | 540 | 1128 | 796 | 1200 | 306 | 1200 | 138 | 960 | 439 |

---

[2]Although *Single Object* is not traditionally considered compositional, we follow GenEval's definition, which includes it as part of a complexity spectrum.

and only samples approved by all are deemed `Correct`. Table 1 reports the number of images in both sets. The ratio of `Correct` images varies notably across models and tasks.

**In-domain and cross-domain settings in SELF-BENCH.** We define *in-domain* as the setting where we evaluate a diffusion classifier on images the same diffusion model generated. Conversely, *cross-domain* refers to images produced by other models from the diffusion family. We define generation accuracy as `#Correct/#Full`, where # denotes the number of images in each set. We primarily use the `Correct` dataset in further analysis.

## 5 Experiments

As mentioned in the Introduction, we aim to investigate three hypotheses. In Section 5.1, we describe the general experimental setup. Section 5.2 (Hypothesis 1) presents a comprehensive evaluation across ten compositional benchmarks covering 33 tasks using three diffusion models, including the new Stable Diffusion 3-m. Section 5.3 (Hypothesis 2) uses our SELF-BENCH benchmark to explore the relationship between generative and discriminative abilities and the role of image domain. Section 5.4 (Hypothesis 3) examines how diffusion timesteps influence discriminative performance.

### 5.1 Experimental setting

**Evaluation settings.** We consider two approaches to turn generative diffusion models into discriminative models: (i) Diffusion Classifier [29] and (ii) Discffusion [13] (see Section A.3 of the Supplemental). However, as shown in Figure A.1 of the Supplemental, Discffusion performs significantly worse than Diffusion Classifiers on SD1.5 and comparably on SD 2.0 and SD3-m. Therefore, we primarily focus on Diffusion Classifiers [29] in our analysis.

**Stable Diffusion baselines.** For the evaluation of diffusion classifiers, we use three versions of Stable Diffusion models: SD1.5, SD2.0 [41], and SD3-m [8]; each new version increases the number of parameters. We selected the baselines based on specific criteria, which are detailed in Section C.1 of the Supplemental. Although we also considered distilled variants of diffusion models (e.g., SDXL-Turbo [45]), we excluded them from our main evaluation due to their architectural and generative differences. Additional analysis is provided in Section E.2 of the Supplemental. For SD1.5 and SD2.0, we use the Euler Discrete scheduler [25] and uniformly sample the timesteps. For SD3-m, we use the Flow Matching EulerDiscrete scheduler [8] for flow matching diffusions, which was designed specifically for the SD3 series. We sample 30 timesteps from each model uniformly, following [13]. (The effect of different numbers of timesteps is discussed in Table B.2 in the Supplemental.)

**CLIP baselines.** We use vanilla CLIP models as key baselines for comparison on discriminative tasks. We use five different versions of CLIP models: RN50x64, ViT-B/32, ViT/L14, ViT/H14, and ViT/g14. We follow OpenAI's implementation for the first three: RN50x64, ViT/B32, and ViT/L14 [38]. On the other hand, we follow OpenCLIP's implementation for ViT-H/14 and ViT-g/14 [18]. Where appropriate, we also report SigLIP and SigLIP2 for completeness, but their behavior closely mirrors that of CLIP.

**Metrics.** The benchmarks vary in structure. Some present paired image-text inputs (i.e., two images and two prompts), while others use a single image with multiple candidate prompts. Across all setups, we primarily evaluate using image-to-text retrieval accuracy, which measures whether the model assigns the highest score to the correct prompt given an image. For paired settings, we compute retrieval accuracy based on whether the matching image-prompt pairs are correctly ranked relative to distractors. Further details on task-specific evaluation protocols are provided in Section C.2.

### 5.2 Scaling evaluation to ten benchmarks

Krojer *et al.* [28] highlight a counterintuitive finding: more capable generative models may perform worse on discriminative tasks. However, their analysis does not include compositional benchmarks. To address this gap, our evaluation includes Stable Diffusion 3 [8], which is very capable in terms of compositional generation. We hypothesize that diffusion models with stronger compositional generation capabilities are more effective on compositional discriminative tasks.

Existing works have explored a limited set of compositional benchmarks (e.g., Winoground [49], ARO [63], CLEVR [23]), demonstrating the strengths of diffusion models compared to CLIP. However, compositional tasks span a broad range of challenges, and it remains unclear whether these

findings generalize across diverse compositional scenarios. To address this, we expand our evaluation to ten complex compositional benchmarks.

**Benchmarks.** Our analysis incorporates ten benchmarks: Vismin [2], EQBench [57], MMVP [50], CLEVR [23], Whatsup [24], Spec [36], ARO [63], Sugarcrepe [16], COLA [40], and Winoground [49]. Each benchmark contains different tasks. For example, Vismin includes tasks related to Object, Attribute, Position, and Counting, whereas COLA is focused on Attribute tasks. The left block in Figure 2 presents examples of these benchmarks. Further details on how the tasks are classified within our compositional task categories are provided in Table C.1 in the Supplemental. In total, we analyze 33 tasks in our main study.

**Results.** Figure 4 shows the average performance of diffusion classifiers on the compositional benchmarks (see Figure E.1 in the Supplemental for the complete results).[3] For the *Position* task, SD3-m performs the best compared to other diffusion models and CLIP models. However, in other tasks, CLIP models usually show better performance than diffusion classifiers, contrary to the findings of previous works [29, 6]. Sim-

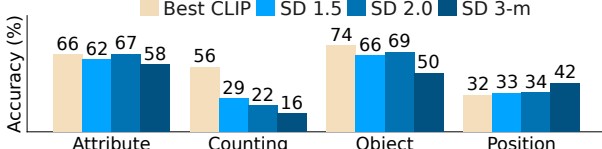

Figure 4: **Evaluating compositional generalization across different categories**. The bars represent average classification accuracies across all tasks within each category. Notably, SD3-m does not generally outperform other Stable Diffusion models in most benchmarks, and CLIP usually outperforms diffusion models.

ilar results hold for SigLIP (see E.7 in Appendix). Surprisingly, among diffusion classifiers, SD3-m is not the best; often SD1.5 or SD2.0 models show better results. These results motivate us to deepen our analysis of when and why diffusion classifiers may underperform.

> **Takeaway hypothesis 1:** Diffusion classifiers excel in spatial position tasks, perform on par with CLIP in "complex" attribute tasks, but underperform in object recognition and counting tasks. Thus, Hypothesis 1 is only partially supported. Additionally, a more capable diffusion model (SD3-m) does not necessarily perform better on compositional discriminative tasks than earlier models (SD1.5, SD2.0).

### 5.3 Studying domain effects via Self-Bench

Here, we take a closer look at domain gap as a possible factor.

**In-domain evaluation.** First, we examine how diffusion classifiers behave in-domain, which may serve as an upper-bound performance estimate, as domain effects are isolated. As shown in Figure 5, they perform better on `Correct` (blue bars) than on `Full` (bright yellow bars) samples. This indicates that the classifiers do not simply choose prompts used for generation; rather, they indeed distinguish between the correct and incorrect prompts.

Moreover, we observe that generation accuracy and discrimination accuracy are positively correlated, contrary to what we saw in other benchmarks in Section 5.2. Specifically, we note that generation accuracy (pink bar in Figure 5) increases

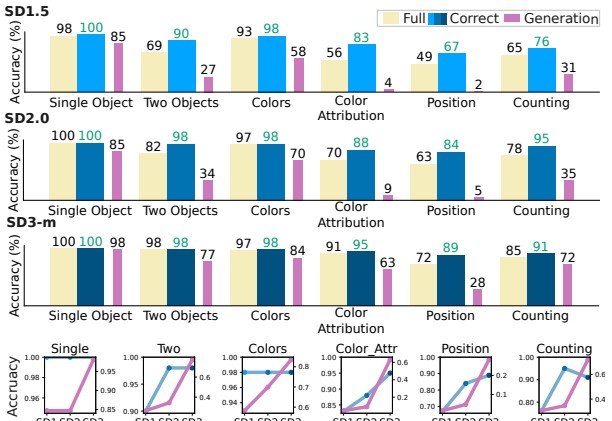

Figure 5: **SELF-BENCH In-domain performance**. (Top three plots) Each row represents the classification accuracy of a diffusion classifier from a specific SD model when evaluated on its own generated data. (Bottom) A positive correlation is observed between generative and discriminative performance. Left axis: discrimination; right axis: generation accuracy.

from SD1.5 to SD3-m, and the discrimination accuracies in both the `Full` and `Correct` categories also rise nearly in parallel from SD1.5 to SD3-m. The correlation coefficient between generation and

---

[3]Tables E.1, E.2, E.3, E.4, E.5, E.6, E.7, E.8, E.9 and E.10 report the quantitative results for all benchmarks used in our evaluation.

`Correct` discrimination accuracy is 0.77. The results suggest that in-domain generation accuracy and discrimination accuracy appear to be positively correlated. The comparison with CLIP is in Figure E.2a of Appendix.

**Comparison In-Domain vs. Cross-Domain.** Next, we focus on comparing *in-domain* and *cross-domain* performance to judge how well models generalize across different domains.

To quantify this, we measure the accuracy drop in models tested on other domains vs. their in-domain SELF-BENCH accuracy, defined as $\Delta A(\text{model}, \text{domain}) = A_{\text{in}}(\text{model}, \text{domain}) - A_{\text{cross}}(\text{model}, \text{domain})$. The differences are averaged across both other domains.

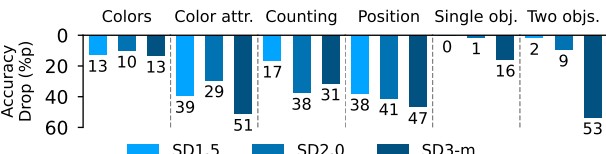

Figure 6: **SELF-BENCH: Cross-domain performance degradation.** The bars represent average drop rate between *in-domain* and *cross-domain* evaluation, averaged over different cross-domain settings. We observe significant accuracy drops when evaluating models on *cross-domain* data. SD3-m shows the most severe degradation, with up to 38% accuracy loss in two-object tasks and 33-40% drops in color and spatial tasks.

Figure 6 illustrates the average accuracy drop when moving from in-domain to cross-domain evaluation on SELF-BENCH. We observe accuracy degradation across all tasks, with SD3-m showing the most drop. While part of the performance gap may be due to task-specific weaknesses in each model, we assume the domain gaps, such as different data distribution, also play a key role in this degradation. Full results can be found in Figure E.7 in the Appendix.

> **Takeaway hypothesis 2**: Diffusion classifiers perform well in-domain, but their accuracy drops significantly in cross-domain settings, highlighting the strong influence of domain shifts.

## 5.4 Timestep weighting effects

Previous works have shown that diffusion models generate images from coarse to fine details over timesteps [30, 55]. However, in classification settings, it is still unclear how different noise levels (i.e., timesteps) affect performance across tasks (e.g., object recognition vs. attribute binding) or different domains (e.g., image style). Diffusion classifiers typically use uniform timestep weighting [29, 28] or a fixed timestep weighting scheme (e.g., $w_t = \exp(-7t)$) [6, 20] across all models (e.g., Imagen [44] and SD). In generation, however, recent works have shown that non-uniform timestep sampling can substantially affect sample quality and training dynamics [26, 65, 56].

We hypothesize that neither strategy is universally optimal (see ablation studies of uniform weighting and fixed timestep weighting in Sec. B.3 in the Supplemental) and that timestep weighting should be adapted to the model and task differently. Here, we investigate how different timesteps contribute to classification decisions. (Figure 10 in the Supplemental provides an intuitive explanation of why timesteps matter from a generative perspective.)

**Important timesteps vary by task and model, and SD3 is especially sensitive.** Figure 7 illustrates the timestep-wise classification accuracy. In this setting, the SD2 and SD3-m models are evaluated on some cross-domain SELF-BENCH tasks. Interestingly, while all timesteps yield non-zero accuracy for SD2, more than 50% of timesteps result in zero accuracy for SD3-m when evaluated on SD2.0 generations. This

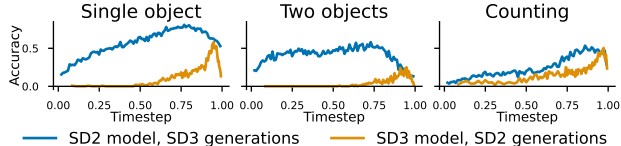

Figure 7: **SELF-BENCH: Single-timestep reconstruction error and classification accuracy.** While SD2.0 maintains good performance on SD3-m's generations, SD3-m exhibits near-zero accuracy for the majority of initial timesteps on SD2.0's generations, particularly for object recognition tasks.

highlights that SD3-m is significantly more sensitive to timestep choice than SD2.0. A key observation is that different timesteps contribute unequally to classification performance, depending on the model and task.

**Reweighted SD3 performs best in real-world benchmarks.** Since we know the optimal timestep is different based on the model and the task, we assess the applicability of our approach to real-world

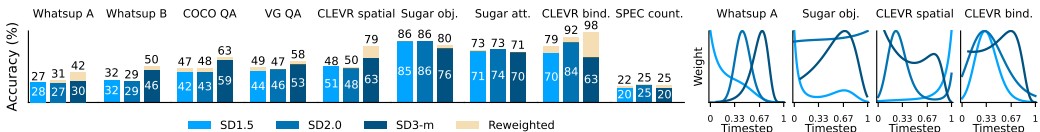

Figure 8: **Low-shot timestep reweighting is effective in real-world benchmarks.** *Left:* Accuracy gains on diverse compositional tasks achieved by reweighting diffusion timesteps for Stable Diffusion variants (SD1.5, SD2.0, SD3-m). Reweighted models consistently outperform the baseline; the gains are most pronounced for the SD3-m model. The numbers above the bars indicate the scores after reweighting, while the numbers inside the bars show the original scores. Positive deltas are highlighted using the reweighted color. *Right:* Learned timestep weighting schemes indicating task-dependent emphasis on specific diffusion steps (early, middle, or late), demonstrating the importance of tailoring timestep selection to the task structure.

scenarios; we also follow Sec. 3.1, but assume only 5% of the data is used for training and 5% for validation, and we report test results. Evaluating our low-shot learning approach on standard benchmarks (Figure 8, left), we find that reweighted SD3-m consistently outperforms both baseline models and their reweighted variants. The improvements are substantial across diverse tasks: 98% on CLEVR binding (vs. 63% baseline), and 42% on WhatsupA spatial task (vs. 30% baseline). Learned weight curves (Figure 8) exhibit diverse patterns that vary depending on the model and task, further underscoring the need for task-specific timestep optimization. Additionally, we find that SD1.5 and SD2.0 do not significantly benefit from timestep weighting. We hypothesize that SD1.5 and SD2.0 perform near-optimally with uniform weighting, while SD3-m may suppress certain timesteps due to training on a smaller, more filtered, and human-aligned dataset than LAION-5B [47].

**Timestep weighting helps mitigate the domain gap.** In Sec. 5.2, we show that SD3-m underperforms SD1.5 and SD2.0 on real-world datasets, and in Sec. 5.3, SD3-m exhibits the largest drop in the cross-domain setting. Since timestep weighting significantly improves SD3-m on real-world tasks, this raises an important question: Does timestep weighting partially help mitigate the domain gap?

To study this further, we conduct an experiment to approximate the effect of domain differences. We generate images using the original prompts from real-world compositional benchmarks. This yields two image sets for each task: (i) the original real-world dataset, and (ii) a synthetic variant generated using the same prompts. Both sets target the same task but differ in visual domain. Using a CLIP image encoder (ViT-B/32) [38], we aim to capture the domain gap[4] between the two, by computing the L2 distance between average embeddings with randomly sampled 50 images. The datapoints in Figure 9 are based on the same real-world benchmarks used in Figure 8. As shown, for SD3, larger CLIP embedding distances (i.e.,

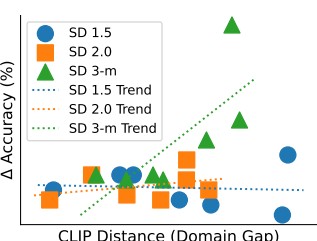

Figure 9: **Timestep Weighting and Domain Gap.** CLIP distances between real-world datasets and SELF-BENCH generations, and corresponding accuracy gains from timestep weighting. Larger domain gaps correlate with greater improvements, but only for SD3.

greater domain gap) are associated with greater reliance on timestep weighting. This suggests a positive correlation between domain gap and the effectiveness of timestep weighting for SD3. In contrast, SD1 and SD2 do not exhibit such a trend. We hypothesize that, as shown in Figure 7, most timesteps in SD2 (also in SD1) are already effective, resulting in limited benefit from reweighting and thus obscuring any correlation with the domain gap. Details are in Section E.6 of the Appendix.

**Qualitative illustration.** To build intuition for how timestep selection impacts classification, we visualise generations from the SD3-m model in Figure 10, starting from a *Self-Bench* image generated by SD2.0 ("a parking meter and a teddy bear"). For each timestep $t$, we corrupt the original image with Gaussian noise corresponding to $t$, and then generate an image by running the diffusion model for 20 denoising steps from $t$ down to 0, conditioning on either a correct or incorrect prompt. We set the classifier-free guidance coefficient to 0.0 (i.e., using only the conditional prompt) to match the discriminative setting used in classification with diffusion.

At very early timesteps (e.g., $t = 0.1$), the generation remains nearly identical regardless of the prompt, suggesting that the model ignores the conditioning - we believe such timesteps are *non-discriminative*. At very late timesteps (e.g., $t = 0.96$), the model strongly reacts to the prompt but also *overwrites* the original image, shifting it out of domain - again rendering the timestep non-

---

[4]Indeed CLIP may capture both stylistic as well as semantic shifts, broadly referred to as "domain."

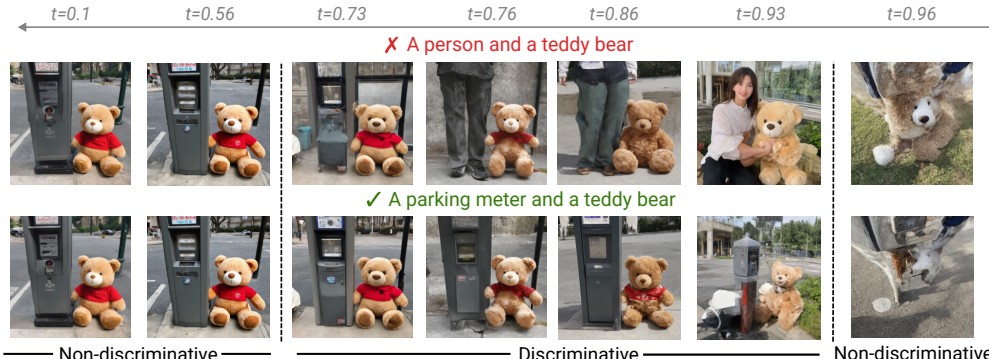

Figure 10: **Intuition for timestep utility.** We visualize SD3-m generations starting from different noise levels applied to a Self-Bench image. (Top) Conditioning on an incorrect caption. (Bottom) Conditioning on the correct one. Only for intermediate timesteps (e.g., $t \in [0.73, 0.93]$) does the model apply meaningful edits without overwriting the original image. See main text for explanation.

discriminative. Only at intermediate timesteps (e.g., $t \in [0.73, 0.93]$ in this case) do we see the model make meaningful edits that reflect the caption while retaining the original structure. We interpret these as *discriminative timesteps* - where the prompt meaningfully affects the output without erasing the original content.

This figure complements our quantitative findings (see Figure 7), offering a visual explanation for why certain timesteps are more informative for classification. In particular, the low accuracy of SD3-m at early timesteps mirrors what we observe qualitatively: early generations fail to reflect the prompt and remain unchanged, making them unsuitable for discrimination.

> **Takeaway hypothesis 3**: Finding optimal timestep weights for a given downstream task in a low-shot setting is an effective way to improve the performance of diffusion classifiers. It helps mitigate the domain gap between diffusion models' generations and real-world test datasets.

## 6 Conclusion

Our work analyzed diffusion classifiers through the lens of compositionality. First, we conducted a comprehensive evaluation across diverse compositional tasks, showing that diffusion classifiers demonstrate compositional understanding in some cases (e.g., Position but not Counting), and revealed a divergence between generative and discriminative abilities. Next, we introduced SELF-BENCH, a diagnostic benchmark of self-generated images, showing that domain shifts significantly affect performance. Finally, we proposed a simple low-shot strategy for mitigating the domain gap.

Despite progress in image generation, our study shows that the zero-shot discriminative ability of diffusion models still falls short of strong discriminative baselines like CLIP, which often remain better on compositional evaluations. We identify domain shift and timestep sensitivity as the decisive factors behind this gap, and we delineate when diffusion classifiers help and when they do not. In response, SELF-BENCH and low-shot timestep reweighting provide practical tools for diagnosing and narrowing the gap through domain-aware evaluation and lightweight adaptation. In short, diffusion classifiers can exhibit compositional understanding, but only under specific, well-aligned conditions.

## Acknowledgements

For compute, Y. Jeong and A. Rohrbach gratefully acknowledge support from the hessian.AI Service Center (funded by the Federal Ministry of Education and Research, BMBF, grant no. 01IS22091) and the hessian.AI Innovation Lab (funded by the Hessian Ministry for Digital Strategy and Innovation, grant no. S-DIW04/0013/003). The research was also supported by the Tübingen AI Center. A. Uselis thanks the International Max Planck Research School for Intelligent Systems (IMPRS-IS) for support.

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

# Appendix

This supplemental material includes extended preliminaries in Section A.1 and Section A.2 and a discussion of design choices and performances for diffusion classifiers in Section A.3 and Section A.4. Ablation studies are presented in Section B. Section C outlines the experimental settings, implementation details, and the full construction of SELF-BENCH. Style alignment experiments are described in Section E.3. We analyze distilled models in Section E.2, and revisit timestep weighting strategies from prior work in Section B.3. Section E.5 illustrates timestep weighting applied to SELF-BENCH, Section B.4 presents zero-shot classification experiments using only later timesteps, and Section E.6 examines CLIP distance between real-world datasets. Finally, Section E includes additional results for all compositional benchmarks.

## Contents

# A  Diffusion classifiers: details and discussion

## A.1  Deriving diffusion classifiers under a unified loss framework

**Unified Loss Formulation.** For a given data sample $\mathbf{x}_0$, usually an encoded image from an autoencoder, we define the loss as a weighted noise (or vector field) prediction error:

$$\mathcal{L}(\mathbf{x}_0) = \mathbb{E}_{t,\boldsymbol{\epsilon}}\Big[w_t \left\| \boldsymbol{\epsilon} - \epsilon_\Theta(\mathbf{z}_t, t, \mathbf{c}) \right\|^2\Big], \tag{A.1}$$

where $\boldsymbol{\epsilon} \sim \mathcal{N}(\mathbf{0}, \mathbf{I})$ is a noise sample, $\mathbf{c}$ is the conditioning variable (e.g. a text prompt), and the noisy sample $\mathbf{z}_t$ and the weight $w_t$ depend on the chosen forward process.

*Diffusion models (SD1.5, SD2.0):* The forward process is given by

$$\mathbf{z}_t = \sqrt{\bar{\alpha}_t}\,\mathbf{x}_0 + \sqrt{1 - \bar{\alpha}_t}\,\boldsymbol{\epsilon}, \quad \boldsymbol{\epsilon} \sim \mathcal{N}(\mathbf{0}, \mathbf{I}),$$

$$\text{with } \alpha_t = 1 - \beta_t \quad \text{and} \quad \bar{\alpha}_t = \prod_{s=1}^{t} \alpha_s.$$

The noise prediction network $\epsilon_\Theta(\mathbf{z}_t, t, \mathbf{c})$ is trained to predict the noise at each discrete timestep (i.e. performing next-step denoising) by minimizing

$$\mathcal{L}_{\text{diff}}(\mathbf{x}_0) = \mathbb{E}_{t,\boldsymbol{\epsilon}}\Big[w_t \left\| \boldsymbol{\epsilon} - \epsilon_\Theta(\mathbf{z}_t, t, \mathbf{c}) \right\|^2\Big].$$

In practice, SD1.5 and SD2.0 typically use uniform weighting (i.e. $w_t := 1$).

*Rectified Flows (RFs) (SD3):* Rectified Flows [33, 1, 31] define the forward process via a straight-line interpolation between the data and a standard normal:

$$\mathbf{z}_t = (1 - t)\,\mathbf{x}_0 + t\,\boldsymbol{\epsilon}, \quad t \in [0, 1], \quad \boldsymbol{\epsilon} \sim \mathcal{N}(\mathbf{0}, \mathbf{I}), \tag{A.2}$$

and the network directly parameterizes a continuous velocity field $\mathbf{v}_\Theta(\mathbf{z}, t)$. The original conditional flow matching (CFM) objective [31] is defined as

$$\mathcal{L}_{\text{CFM}} = \mathbb{E}_{t,\, p_t(\mathbf{z}|\boldsymbol{\epsilon}),\, p(\boldsymbol{\epsilon})}\Big[\left\| \mathbf{v}_\Theta(\mathbf{z}, t) - \mathbf{u}_t(\mathbf{z} \mid \boldsymbol{\epsilon}) \right\|^2\Big], \tag{A.3}$$

where $\mathbf{u}_t(\mathbf{z} \mid \boldsymbol{\epsilon})$ is the target vector field along the linear path in (A.2). By reparameterizing the CFM objective as a noise-prediction loss (see, e.g., [8]), we obtain

$$\mathcal{L}_{\text{RF}}(\mathbf{x}_0) = \mathbb{E}_{t,\boldsymbol{\epsilon}}\Big[w_t \left\| \epsilon_\Theta(\mathbf{z}_t, t, \mathbf{c}) - \boldsymbol{\epsilon} \right\|^2\Big], \tag{A.4}$$

with a time-dependent weight $w_t$.

For the linear interpolation in (A.2), choosing $w_t^{\text{RF}} = \frac{t}{1-t}$ recovers the original CFM objective. While $w_t^{\text{RF}}$ is the default weighting for RFs, SD3 [8] uses logit-normal weighting. Moreover, SD3 operates in continuous time, so the loss $\mathcal{L}(\mathbf{x}_0)$ is defined continuously with respect to $\mathbf{x}_0$.

Using this formulation we can interpret SD3 under the diffusion loss objective, despite its different underlying architecture. Therefore, we can use it as a classifier in the similar way to earlier diffusion models. The only difference lies in the weighting function $w_t$, which for SD3 follows a logit-normal distribution rather than the uniform weighting used in SD1.5/2.0.

Thus, both Diffusion and RFs ultimately train the network to match a target signal via the unified loss in (A.5). In diffusion the network directly predicts the noise at each discrete timestep, whereas in RFs the network predicts a continuous velocity field whose reparameterized form is trained to match the noise—with conditioning on $\mathbf{c}$ in both cases.

Given the unified loss in Equation (A.5),

$$\mathcal{L}(\mathbf{z}_t, \mathbf{c})(\mathbf{x}_0) = \mathbb{E}_{t,\boldsymbol{\epsilon}}\Big[w_t \left\| \boldsymbol{\epsilon} - \epsilon_\Theta(\mathbf{z}_t, t, \mathbf{c}) \right\|^2\Big], \tag{A.5}$$

**Diffusion Classifiers.** [29] define a diffusion classifier as a network that minimizes this loss with respect to the data sample $\mathbf{x}_0$ and a given conditioning variable $\mathbf{c}$. Since computing $p_\theta(\mathbf{x} \mid \mathbf{c})$ is intractable for diffusion models, the ELBO is used in place of $\log p_\theta(\mathbf{x} \mid \mathbf{c})$. In particular, assuming that

$$\log p_\theta(\mathbf{x} \mid \mathbf{c}) \propto -\mathbb{E}_{t,\boldsymbol{\epsilon}}\left[w_t \left\| \boldsymbol{\epsilon} - \epsilon_\theta(\mathbf{z}_t, t, \mathbf{c}) \right\|^2\right],$$

so that, with a uniform prior over labels (i.e. $p(\mathbf{c}_i) = \frac{1}{N}$), Bayes' rule implies

$$p_\theta(\mathbf{c}_i \mid \mathbf{x}) \propto \exp\left\{ -\mathbb{E}_{t,\boldsymbol{\epsilon}}\left[w_t \left\| \boldsymbol{\epsilon} - \epsilon_\theta(\mathbf{z}_t, t, \mathbf{c}_i) \right\|^2\right] \right\}. \tag{A.6}$$

In practice, we approximate the expectation in Eq. (A.6) via Monte Carlo sampling. For each class label $\mathbf{c}_i$, we sample a fixed set

$$S = \{(t_j, \boldsymbol{\epsilon}_j)\}_{j=1}^N,$$

with $t_j$ drawn from the prescribed distribution and $\boldsymbol{\epsilon}_j \sim \mathcal{N}(\mathbf{0}, \mathbf{I})$. We then compute the empirical weighted error

$$\hat{E}(\mathbf{c}_i) = \frac{1}{N} \sum_{j=1}^N w_{t_j} \left\| \boldsymbol{\epsilon}_j - \epsilon_\theta\left(\mathbf{z}_{t_j}, t_j, \mathbf{c}_i\right) \right\|^2. \tag{A.7}$$

Substituting these estimates into Eq. (A.6) yields the approximate posterior

$$p_\theta(\mathbf{c}_i \mid \mathbf{x}) \approx \frac{\exp\{-\hat{E}(\mathbf{c}_i)\}}{\sum_k \exp\{-\hat{E}(\mathbf{c}_k)\}}. \tag{A.8}$$

The predicted label is then given by

$$\hat{\mathbf{c}} = \arg\max_{\mathbf{c}_i} p_\theta(\mathbf{c}_i \mid \mathbf{x}). \tag{A.9}$$

Using the same sample set $S$ across all classes reduces the variance in the estimated differences in weighted prediction error. This approach extracts a classifier directly from a pretrained conditional diffusion model without any additional training.

## A.2 Details on timestep weighting

For a given latent representation $\mathbf{z}$ and class $y_k \in \{1, \ldots, K\}$, we compute the loss using the fixed set

$$S = \{(t_j, \boldsymbol{\epsilon}_j)\}_{j=1}^{T_s} : e_j(\mathbf{z}, y_k) = \left\| \boldsymbol{\epsilon}_j - \epsilon_\Theta(\mathbf{z}, t_j, \phi(y_k)) \right\|^2,$$

where $\phi(y_k)$ is the text embedding of class $y_k$. The class probabilities are computed using a weighted sum over the fixed timesteps:

$$p(y = y_k \mid \mathbf{z}) = \frac{\exp\left( -\sum_{j=1}^{T_s} w_{t_j} e_j(\mathbf{z}, y_k) \right)}{\sum_{l=1}^K \exp\left( -\sum_{j=1}^{T_s} w_{t_j} e_j(\mathbf{z}, y_l) \right)}.$$

## A.3 Design choices for diffusion classifiers

We found that using diffusion classifiers is tricky in practise. There are a few design choices that differ across experimental setups in previous works [20, 29, 28, 6]. In this subsection we provide a fuller picture of the design choices that matter for diffusion classifiers.

We distinguish between five main factors that potentially differ in previous works and can affect the performance of diffusion classifiers:

① *Loss weighting.* Previous works either used uniform weighting (i.e. $w_t := 1$) [29, 28] or used a time-dependent weighting scheme found empirically on the training set using CIFAR-100 [6, 20], using $w_t := \exp(-7\tilde{t})$ (for normalized $\tilde{t} \in [0, 1]$). These schedules are usually motivated by the generative training objective, which views each timestep $t$ as corresponding to a different noise level — effectively treating $t$ as a proxy for input corruption. However, from a discriminative perspective,

different timesteps correspond to distinct representations within the model, akin to how different layers of a neural network encode features of varying abstraction. In standard supervised models, it has been shown that lower-level features (often found in earlier layers) can be more robust under distribution shifts [51]. Drawing this analogy, early timesteps in diffusion models may similarly preserve more local or low-level features that are useful for generalization.

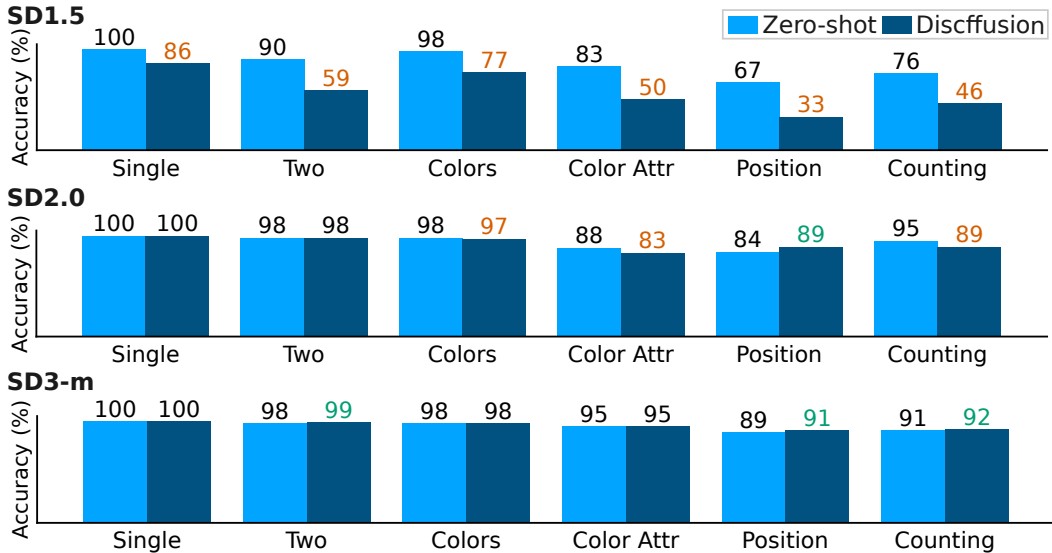

Figure A.1: **Comparison between Zero-shot classifier and Discffusion on SELF-BENCH *in-domain***. Zero-shot Classifier and Discffusion do not show much performance difference, or Discffusion performs worse.

② *Classifier-free guidance.* Classifier-free guidance [14] (CFG) is a de-facto standard in diffusion models for improving the quality of generated images at a cost of lesser variance in generated images. When generating images using CFG using, e.g. DDIM [25] for SD1.5, SD2.0, or Euler sampler for SD3 [8],

All previous works have either not used classifier-free guidance [6, 28] or used it in a limited setting [29] with a conclusion that it does not lead to better classification performances. We largely found the same conclusion to hold.

③ *Model quantization.* Previous works usually do not mention quantization of the model. In practice, all the codebases associated with diffusion classifiers use quantized models, using 16-bit floating-point precision. While the classification accuracy did not vary significantly when switching from float16 to float32 implementation. However, an important impact of the model quantization was measuring errors, and casting the reconstruction distances up to float32 to avoid identical values with varying conditionals.

④ *Variations of empirical weighted error objective.* Diffusion models are usually trained using a form of the weighted error objective (Equation (A.7)), using L2 loss (i.e., squared L2 norm). Most of the previous works use this form of the loss. However, even though [20] has claimed that deviating from the L2 loss will not work, often times this is not the case, and [29] has shown that using the L1 loss can improve performance in non-compositional settings.

⑤ *Sampling strategy.* Previous works mostly have found that using a uniform timestep distribution is the best option, i.e. $\pi_N = \text{Uniform}([0, 1])$. This choice is usually motivated by sticking to the training objective of the model. As we show in Section 3.1, SD3 was trained using timesteps sampled from logit-normal distribution.

⑥ *Alternative methods for diffusion classifiers.* Discffusion [13] uses attention scores from the cross-attention layers of diffusion models, aggregated via LogSumExp (LSE) pooling [3]. Since SD3 replaces traditional cross-attention layers with self-attention layers that incorporate text conditioning, we instead extract attention scores from these self-attention layers. In Figure A.1, we compare the Zero-shot Classifier [29] and Discffusion [13]. In the SELF-BENCH *in-domain* setting, which can

be seen as the fairest setting to evaluate the optimal performance of each method, the performance of Discffusion is worse than the Zero-shot Classifier. This result contradicts Discffusion's claim of achieving better accuracy in some benchmarks.

Overall, *the prevailing notion in the previous works is that good classifiers can be derived by adhering to the generative training objective of the models.*

## A.4 Discussion of misalignment between discriminative and generative performance

We hypothesize that the misalignment between discriminative and generative performance often arises from internal domain-specific biases or spurious correlations acquired during training.

For example, suppose the model frequently sees or generates "small objects" against a blue background. In that case, it becomes easy for the model to generate such scenes. However, during classification, if it encounters a large object with the same blue background, it may still assign a high probability to the "small object" class. This is because it has strongly associated "small object" with "blue background" in its generative space. In other words, the model's discriminative predictions $p(y|x)$ may appear accurate in cases that align with its generative bias but fail when the context shifts. This illustrates that a model's ability to generate realistic samples does not necessarily imply robust or disentangled discriminative representations.

Concretely, we hypothesize that SD3-m focuses on a narrower, high-quality generative domain (akin to sharp spikes in the distribution), which may lead to lower diversity. In contrast, SD2.0, despite lower visual fidelity, may cover a broader range of variations, leading to better discriminative performance on diverse inputs.

To support this hypothesis, we conduct a diversity analysis using SELF-BENCH. For each text prompt, we sample four images from both SD2.0 and SD3-m. We then compute CLIP-B/32 embeddings and analyze: i) mean pairwise cosine similarity among the 4 images (lower indicates higher visual diversity) and ii) mean variance across embedding dimensions (higher indicates more diverse feature space coverage).

Table A.1: Diversity comparison between SD2.0 and SD3-m using CLIP embeddings (per prompt, $n = 4$ images).

| Metric | SD2.0 | SD3-m |
|---|---|---|
| Mean cosine similarity ↓ | $0.845 \pm 0.062$ | $0.895 \pm 0.051$ |
| Mean embedding-dim variance ($\times 10^{-3}$) ↑ | $0.227 \pm 0.190$ | $0.154 \pm 0.250$ |

These results support our hypothesis: SD2.0 produces more diverse samples in terms of both visual similarity and embedding-space variance.

We emphasize that this remains a hypothesis rather than a complete explanation. A more rigorous characterization of the generative–discriminative alignment is left to future work. Nevertheless, recent finding [10] appears consistent with our observations, showing that SD2.0 (and SD1.5) tend to generate more diverse samples than SD3-m.

# B    Ablation Studies

We performed two ablations: (1) varying the resolution for SD3 on the self-bench, and (2) using 30 vs. 100 timesteps for all self-bench experiments, and (3) using a different timestep weighting scheme.

## B.1    Image resolution

As shown in Table B.1 higher resolution generally leads to better performance for SD3-m model.

Table B.1: Geneval ablation using SD3, comparing impact of input resolution. Larger images are always better. Used 100 time samples ($T_s = 100$).

| Task | GenEval Version | Resize Acc | No-Resize Acc | Diff (Resize - No) |
|---|---|---|---|---|
| geneval_color_attr | 1.5 | 33.33% | 55.56% | -22.22% |
| geneval_color_attr | 2 | 69.44% | 59.72% | 9.72% |
| geneval_color_attr | 3-m | 97.22% | 98.09% | -0.87% |
| geneval_colors | 1.5 | 87.67% | 94.75% | -7.08% |
| geneval_colors | 2 | 91.25% | 94.30% | -3.04% |
| geneval_colors | 3-m | 98.41% | 99.68% | -1.27% |
| geneval_counting | 1.5 | 43.88% | 61.22% | -17.35% |
| geneval_counting | 2 | 62.16% | 68.47% | -6.31% |
| geneval_counting | 3-m | 58.26% | 96.30% | -38.04% |
| geneval_position | 1.5 | 66.67% | 66.67% | 0.00% |
| geneval_position | 2 | 52.63% | 44.74% | 7.89% |
| geneval_position | 3-m | 70.80% | 93.81% | -23.01% |
| geneval_single | 1.5 | 90.04% | 91.81% | -1.77% |
| geneval_single | 2 | 93.36% | 94.41% | -1.05% |
| geneval_single | 3-m | 98.41% | 100.00% | -1.59% |
| geneval_two | 1.5 | 62.86% | 69.52% | -6.67% |
| geneval_two | 2 | 72.09% | 78.29% | -6.20% |
| geneval_two | 3-m | 91.50% | 98.11% | -6.61% |

## B.2    Varying number of timesteps

Increasing to 100 timesteps results in improved performance, although this gain is most notable for SD3 (sampled with uniform weights) [8] in Table B.2.

Additionally, we attempted to match the timestep weighting scheme used in the original SD3-m model by employing logit-normal weighting. However, this approach yielded exceptionally poor results: in SELF-BENCH *Cross-domain* experiments, the model did not exceed 11% accuracy, regardless of the generative model or task.

Table B.2: Comparison of 30 vs 100 Timesteps Performance

| Task | GenEval Ver. | Model Ver. | 30 Steps | 100 Steps | Diff |
|---|---|---|---|---|---|
| Color Attr | 1.5 | 1.5 | 84.00% | 88.00% | 4.00% |
| Color Attr | 1.5 | 2 | 55.56% | 55.56% | 0.00% |
| Color Attr | 1.5 | 3-m (no-resize) | 55.56% | 55.56% | 0.00% |
| Color Attr | 2 | 1.5 | 50.00% | 55.56% | 5.56% |
| Color Attr | 2 | 2 | 83.33% | 88.89% | 5.56% |
| Color Attr | 2 | 3-m (no-resize) | 50.00% | 59.72% | 9.72% |
| Color Attr | 3-m | 1.5 | 55.16% | 64.29% | 9.13% |
| Color Attr | 3-m | 2 | 60.71% | 68.65% | 7.94% |
| Color Attr | 3-m | 3-m (no-resize) | 93.25% | 98.09% | 4.84% |
| Colors | 1.5 | 1.5 | 96.80% | 99.09% | 2.28% |
| Colors | 1.5 | 2 | 89.50% | 94.75% | 5.25% |
| Colors | 1.5 | 3-m (no-resize) | 85.84% | 94.75% | 8.90% |
| Colors | 2 | 1.5 | 87.07% | 91.25% | 4.18% |
| Colors | 2 | 2 | 98.86% | 99.62% | 0.76% |
| Colors | 2 | 3-m (no-resize) | 88.97% | 94.30% | 5.32% |
| Colors | 3-m | 1.5 | 80.57% | 84.87% | 4.30% |
| Colors | 3-m | 2 | 90.13% | 92.99% | 2.87% |
| Colors | 3-m | 3-m (no-resize) | 98.73% | 99.68% | 0.96% |
| Counting | 1.5 | 1.5 | 75.51% | 79.59% | 4.08% |
| Counting | 1.5 | 2 | 57.14% | 62.76% | 5.61% |
| Counting | 1.5 | 3-m (no-resize) | 47.96% | 61.22% | 13.27% |
| Counting | 2 | 1.5 | 59.46% | 68.47% | 9.01% |
| Counting | 2 | 2 | 92.79% | 95.95% | 3.15% |
| Counting | 2 | 3-m (no-resize) | 56.76% | 68.47% | 11.71% |
| Position | 1.5 | 1.5 | 66.67% | 83.33% | 16.67% |
| Position | 1.5 | 2 | 50.00% | 41.67% | -8.33% |
| Position | 1.5 | 3-m (no-resize) | 33.33% | 66.67% | 33.33% |
| Position | 2 | 1.5 | 31.58% | 26.32% | -5.26% |
| Position | 2 | 2 | 73.68% | 84.21% | 10.53% |
| Position | 2 | 3-m (no-resize) | 36.84% | 44.74% | 7.89% |
| Position | 3-m | 1.5 | 30.97% | 30.97% | 0.00% |
| Position | 3-m | 2 | 46.90% | 45.58% | -1.33% |
| Position | 3-m | 3-m (no-resize) | 82.30% | 93.81% | 11.50% |
| Single | 1.5 | 1.5 | 100.00% | 99.63% | -0.37% |
| Single | 1.5 | 2 | 98.89% | 99.08% | 0.18% |
| Single | 1.5 | 3-m (no-resize) | 83.39% | 91.81% | 8.42% |
| Single | 2 | 1.5 | 98.89% | 98.89% | 0.00% |
| Single | 2 | 2 | 100.00% | 100.00% | 0.00% |
| Single | 2 | 3-m (no-resize) | 89.67% | 94.41% | 4.74% |
| Single | 3-m | 1.5 | 99.68% | 100.00% | 0.32% |
| Single | 3-m | 2 | 99.36% | 99.84% | 0.48% |
| Single | 3-m | 3-m (no-resize) | 100.00% | 100.00% | 0.00% |
| Two | 1.5 | 1.5 | 91.43% | 95.24% | 3.81% |
| Two | 1.5 | 2 | 85.71% | 90.00% | 4.29% |
| Two | 1.5 | 3-m (no-resize) | 52.38% | 69.52% | 17.14% |
| Two | 2 | 1.5 | 89.15% | 92.25% | 3.10% |
| Two | 2 | 2 | 97.67% | 99.61% | 1.94% |
| Two | 2 | 3-m (no-resize) | 56.59% | 78.29% | 21.71% |
| Two | 3-m | 1.5 | 87.91% | 91.83% | 3.92% |
| Two | 3-m | 2 | 92.81% | 93.63% | 0.82% |
| Two | 3-m | 3-m (no-resize) | 96.41% | 98.11% | 1.71% |

## B.3 On universality of timestep weighting

Previous work has shown that learning a task-specific timestep weighting function, while beneficial, typically results in only modest gains, on average around 1% improvement in performance. These results have mostly been reported on classification tasks involving single, non-compositional queries.

Here, we test whether such universal weighting functions, as proposed in earlier work, can also be effective in our setting. To do so, we evaluate all models on our proposed datasets using an exponential timestep weighting function defined as $\exp(-7t)$, where $t$ is the normalized timestep ranging from 0 to 1.

We present the results in Table B.3. Overall, we find that uniform and exponentially weighted models perform quite similarly for 1.52 diffusion models, although the gap is often quite large between the two. While for SD-3m model, uniform weighting is almost always better.

Table B.3: **Mean accuracies by version and task.** Bold indicates the larger of the uniform or exponentially–weighted scores.

| Version | Task | Uniform | Exp. Weighted |
|---|---|---|---|
| | COCO QA | 0.44 | **0.48** |
| | VG QA | 0.44 | **0.49** |
| | CLEVR Binding–Color | 0.67 | **0.70** |
| | CLEVR Spatial | **0.50** | 0.48 |
| 1.5 | Spec Count | **0.20** | 0.12 |
| | Sugar Attributes | **0.70** | 0.61 |
| | Sugar Objects | **0.85** | 0.78 |
| | WhatsUp A | **0.27** | 0.27 |
| | WhatsUp B | 0.26 | **0.28** |
| | COCO QA | 0.42 | **0.47** |
| | VG QA | 0.48 | **0.49** |
| | CLEVR Binding–Color | **0.82** | 0.74 |
| | CLEVR Spatial | 0.49 | **0.50** |
| 2 | Spec Count | **0.23** | 0.12 |
| | Sugar Attributes | **0.76** | 0.64 |
| | Sugar Objects | **0.85** | 0.81 |
| | WhatsUp A | 0.26 | **0.30** |
| | WhatsUp B | **0.26** | 0.23 |
| | COCO QA | **0.56** | 0.55 |
| | VG QA | **0.54** | 0.52 |
| | CLEVR Binding–Color | 0.60 | **0.75** |
| | CLEVR Spatial | **0.62** | 0.51 |
| 3-m | Spec Count | **0.19** | 0.11 |
| | Sugar Attributes | **0.72** | 0.61 |
| | Sugar Objects | **0.74** | 0.71 |
| | WhatsUp A | **0.28** | 0.27 |
| | WhatsUp B | **0.40** | 0.29 |

## B.4 Biased timestep sampling

We explore whether further gains could be achieved with SD3 based on our prior analysis. Motivated by findings from timestep weighting, we tested a simplified variant where only mid-to-late timesteps were used for discrimination. We find that these strategies did not yield meaningful improvements. The results are presented in Table B.4.

Table B.4: Results using later timesteps (sampling from $t \sim [0.5, 1]$ for classification.

| Method/Dataset | Self-Bench2.0 Single | Self-Bench2.0 Counting | WhatsUpA | CLEVR Binding |
|---|---|---|---|---|
| **SD3-m** | 0.87 | 0.57 | 0.30 | 0.63 |
| **SD3-m (supervised)** | 0.91 | 0.72 | 0.42 | 0.98 |
| Later Timesteps | **0.90** | **0.66** | **0.31** | 0.59 |

# C   Details of Experiments settings

## C.1   Choice of Baselines

We select these versions for the following reasons: i) SD1.5 was previously used in diffusion classifier evaluations [28], ii) SD2.0 demonstrated better performance compared to SD2.1 [62] in discriminative tasks, and iii) SD3-m is one of the state-of-the-art generative models, and we use its medium variant since it offers effective performance while being more lightweight than the full SD3 model. It has not been studied in the context of diffusion classifiers. We have also considered distilled models as baselines (i.e., FLUX [60] and SDXL-Turbo [45]). However, we did not include them since it is difficult to determine their performance under no classifier-free guidance (CFG-free) settings. Some analysis is provided in Section E.2.

## C.2   Implementation Details

**Evaluations.** For evaluation, we use a single A100 GPU for all tasks with a batch size of 4. The evaluation time depends on the SD model, the number of negative prompts, and the dataset size. While SD1.5 and SD2.0 require similar amounts of time, SD3-m takes significantly longer. Specifically, for a dataset with 230 images and 4 prompts per image (one positive and three negative), evaluation takes approximately 15 minutes for SD 1.5 and 30 minutes for SD 3-m. We also used stuned library for running the experiments [43].

**Training details on timestep weighting.** We train on 100 timesteps using the Adam optimizer. For the low-shot setting, we fit a third-degree polynomial, while for the Self-bench experiments, we use the full timestep vector. We use no regularization, set the learning rate to $\ell = 0.05$, and train for 5,000 epochs. The entire optimization procedure is performed on frozen scores, allowing us to infer weights in under a minute for datasets with fewer than 1,000 samples.

**Benchmarks.** Table C.1 presents the benchmarks used in our study, categorized into Attribute, Object, Position, Counting, Complex Relation, Action, Size, and others. The benchmarks include Vismin [2], EQBench [57], MMVP [50], CLEVR [23], Whatsup [24], Spec [36], ARO [63], Sugarcrepe [16], COLA [40], and Winoground [49].

Table C.1: **Categorization of compositional benchmarks**. For EQBench and Vismin, an official subset is used.

| Category | Datasets |
|---|---|
| Attribute | Aro (Attribute) [63]
SugarCrepe (Attribute) [16]
Vismin (Attribute) [2]
EQBench (EQ-Kubric Attribute, EQ-SD) [57]
MMVP (Color) [50]
CLEVR (pair binding color, recognition color, recognition shape, binding color shape, binding shape color) [23]
COLA (Multi Object) [40] |
| Object | Winoground (Object) [49]
SugarCrepe (Object) [16]
Vismin (Object) [2] |
| Position (Spatial Relation) | WhatsUp (WhatsUp A, WhatsUp B, COCO-spatial one, COCO-spatial two, GQA-spatial one, GQA-spatial two) [24]
SPEC (Absolute Spatial, Relative Spatial) [36]
EQBench (Location) [57]
Vismin (Relation) [2]
MMVP (Spatial, Orientation, Perspective) [50]
CLEVR (spatial) [23] |
| Counting | SPEC (Count) [36], EQBench (EQ-Kubric Counting) [57], Vismin (Counting) [2] |
| Complex Relation | Aro (Relation, COCO order, Flickr order) [63]
SugarCrepe (Relation) [16]
Winoground (Relation, Both) [49]
MMVP (State, Structural Character) [50] |
| Action | EQBench (YouCook2, GEBC, AG) [57] |
| Size | SPEC (Absolute Size, Relative Size) [36]
CLEVR (Size) [23] |
| ETC | MMVP (Text) [50] |

Task formulations vary across benchmarks. For example, Winoground consists of two images paired with two captions, each describing two objects. The negative caption is created by swapping the objects in the text. In contrast, Vismin (Object) includes prompts that modify the object by replacing it with another randomly selected object that is not present in the image. The captions in Vismin can describe either a single object or multiple objects.

We use a subset of EQBench and Vismin in our evaluation. Additionally, we include COLA only for multi-object tasks due to the difficulty of selecting negative prompts. However, we found that the name COLA (Multi Object) does not align well with the task's focus, as it primarily deals with object attributes in the prompts. Therefore, we classify COLA under the Attribute category.

As mentioned earlier, some benchmarks consist of a single image paired with multiple text prompts, while others feature two images with two matching captions. The latter category includes Winoground, COLA, Vismin, EQBench, and MMVP, whereas all other benchmarks belong to the former category.

**Categories** To enable a structured analysis, we group the tasks into four categories: *Object*, *Attribute*, *Position*, and *Counting*. Each category is designed to target a specific aspect of compositional understanding through carefully crafted text prompts.

- *Object*: Evaluates object recognition (often in context) within a given context by introducing modifications such as swapping, replacing, or removing objects to assess the model's ability to distinguish between different entities.
- *Attribute*: Focuses on descriptive properties (e.g., adjectives) associated with objects, such as variations in color or shape, to determine the model's sensitivity to attribute-object relationships.
- *Position*: Examines spatial relationships between objects or the perspective of a single object (e.g., "a dog on the left side of the image").
- *Counting*: Assesses numerical reasoning by prompting the model to count specific objects in an image.

**Image-Text matching scores.** As mentioned above, the task can be divided into two parts. First, there is one image with multiple texts. Second, there are two images and two texts in one pair. For the first setting, we simply pick the best prompt with Equation A.9. However, for the second task, following previous approaches [29] to get the text score, if we have a pair of <image1, text1, image2, text2>, we use the following equation:

$$
\mathbb{I}\begin{bmatrix} \text{score}(\texttt{text1}, \texttt{image1}) > \text{score}(\texttt{text2}, \texttt{image1}) \\ \textbf{AND} \\ \text{score}(\texttt{text2}, \texttt{image2}) > \text{score}(\texttt{text1}, \texttt{image2}) \end{bmatrix} \tag{C.1}
$$

where score follows Equation A.9.

# D  Details of SELF-BENCH

## D.1  Design and filtering

**Creating discrimination tasks.** We use generation prompts from Geneval [11]. The prompt template "a photo of" is used in the experiment for both generation and discrimination.

These are the possible choices for each discrimination task.

- **Colors**: "red", "orange", "yellow", "green", "blue", "purple", "pink", "brown", "black", "white"
- **Positions**: "left of", "right of", "above", "below"
- **Counting**: "one", "two", "three", "four"
- **Single Object**: "person", "bicycle", "car", "motorcycle", "airplane", "bus", "train", "truck", "boat", "traffic light", "fire hydrant", "stop sign", "parking meter", "bench", "bird", "cat", "dog", "horse", "sheep", "cow", "elephant", "bear", "zebra", "giraffe", "backpack", "umbrella", "handbag", "tie", "suitcase", "frisbee", "skis", "snowboard", "sports ball", "kite", "baseball bat", "baseball glove", "skateboard", "surfboard", "tennis racket", "bottle", "wine glass", "cup", "fork", "knife", "spoon", "bowl", "banana", "apple", "sandwich", "orange", "broccoli", "carrot", "hot dog", "pizza", "donut", "cake", "chair", "couch", "potted plant", "bed", "dining table", "toilet", "tv", "laptop", "computer mouse", "tv remote", "computer keyboard", "cell phone", "microwave", "oven", "toaster", "sink", "refrigerator", "book", "clock", "vase", "scissors", "teddy bear", "hair drier", "toothbrush"

For *Two Objects* and *Color Attribution*, the possible choices are the same as those in *Single Object* and *Colors*, respectively. Since both *Two Objects* and *Color Attribution* involve two objects, the latter additionally requires binding two color choices to the respective objects (e.g., "a red dog and a yellow cat"). As a result, the *Color Attribution* category can have up to 100 different prompt combinations.

For the *Two Objects* category, the number of possible prompts is large (6,400). To manage this, we consider only 101 prompts: one containing both true objects and 100 cases where one true object is paired with a randomly selected object.

**Crowdsourcing with manual filtering.** We recruited three annotators and provided them with detailed instructions for the filtering process. The task took approximately 2–3 hours, and no compensation was provided. Each annotator was instructed to filter images based on category-specific criteria, following the Geneval framework [11]. For example, the *Two Objects* category requires generating two objects mentioned in the original prompt clearly. Additionally, *Single Object* category requires the presence of the mentioned object, and the number of the objects does not matter; it simply checks if the mentioned object is in the image. It does not matter if non-mentioned object is also in the same image.

Figure D.1 displays an example of the filtering interface where the annotator can label the images. Since there are ambiguous examples, such as an image with only half of an object, we provided three options: "Good", "Ambiguous", and "Wrong". If any one of the three annotators labels the example as ambiguous or wrong, the image is filtered out later. Examples of filtered images are provided in Figure D.2.

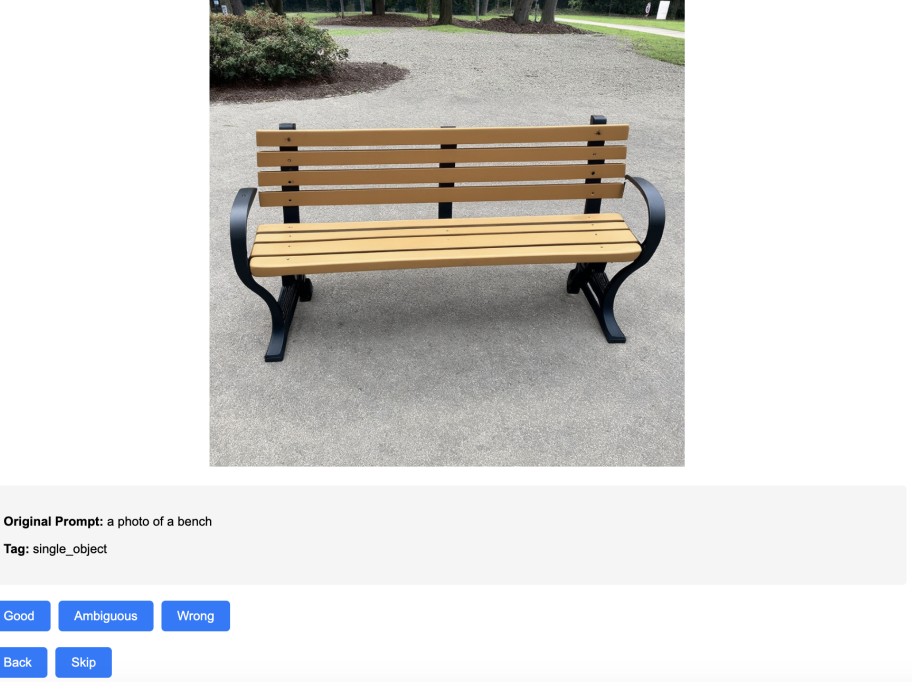

Figure D.1: **Example of a filtering userspace.** The annotator should select one among three options. Before they annotate, they get the instructions for each category.

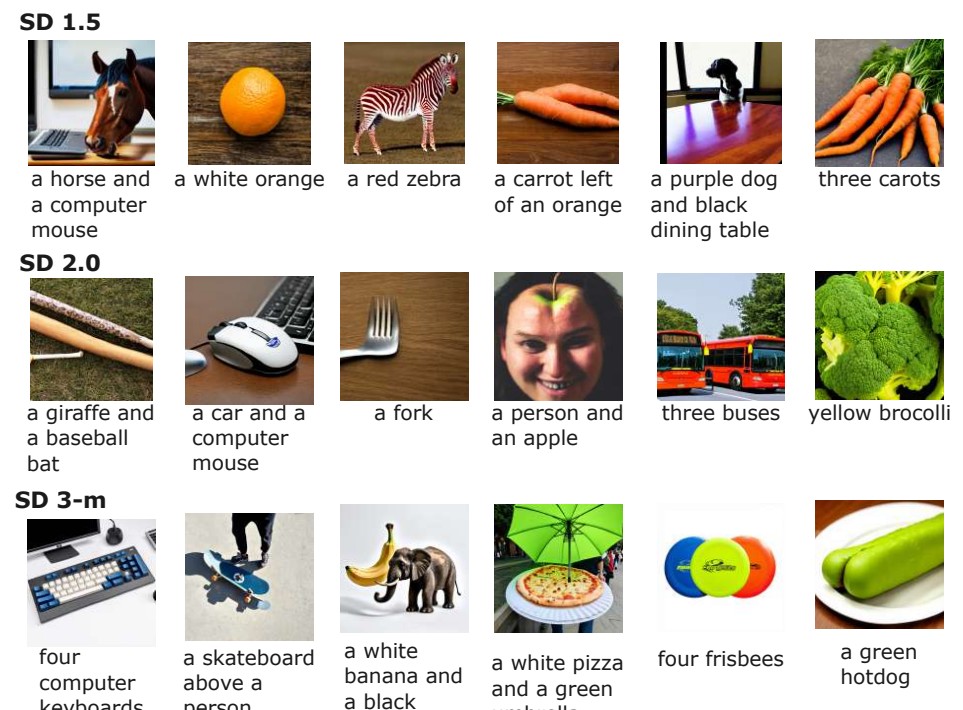

Figure D.2: **Images filtered out by annotators.** Annotators removed images that did not meet the criteria specified in the category instructions.

## D.2 Qualitative examples

Figure D.3 illustrates the domain differences between existing compositional benchmarks and SD3-m generated results, using prompts from SELF-BENCH, SPEC [36], and WhatsUP [24]. Figure D.4 presents additional examples from SELF-BENCH. Figure D.5 highlights failure cases of the Diffusion Classifier, even in *in-domain* scenarios.

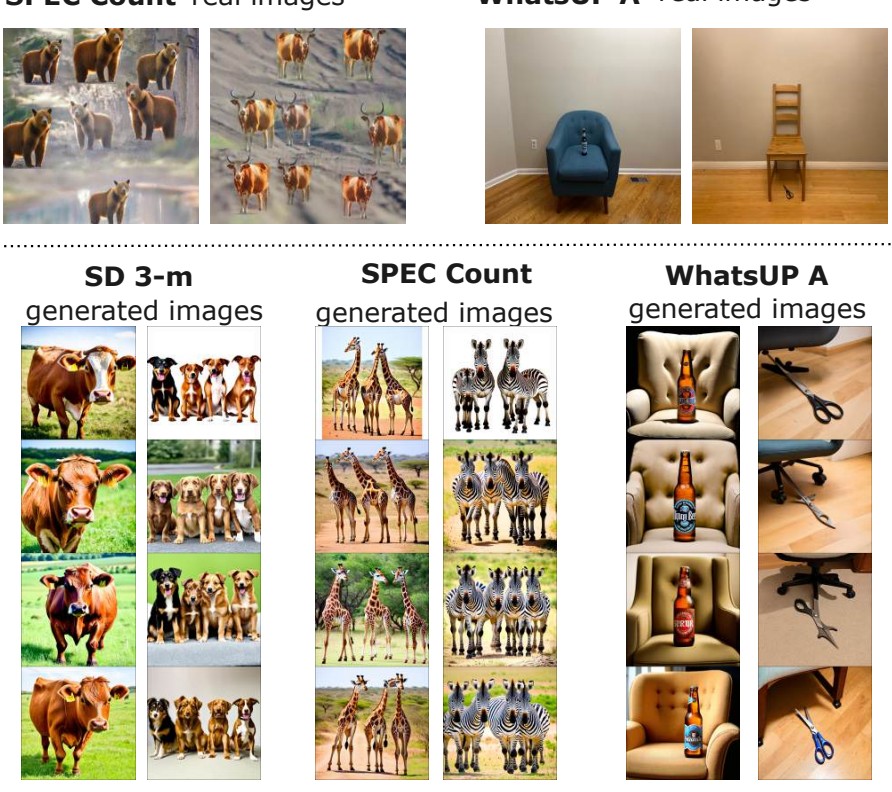

Figure D.3: **Domain differences between existing compositional benchmarks and SD3-m generated results.** (top) Examples from existing compositional benchmarks: SPEC [36] and WhatsUP A [24]. (bottom left) SELF-BENCH examples. (bottom middle) Generated images using prompts from SPEC count tasks. (bottom right) Generated images using prompts from WhatsUPA tasks.

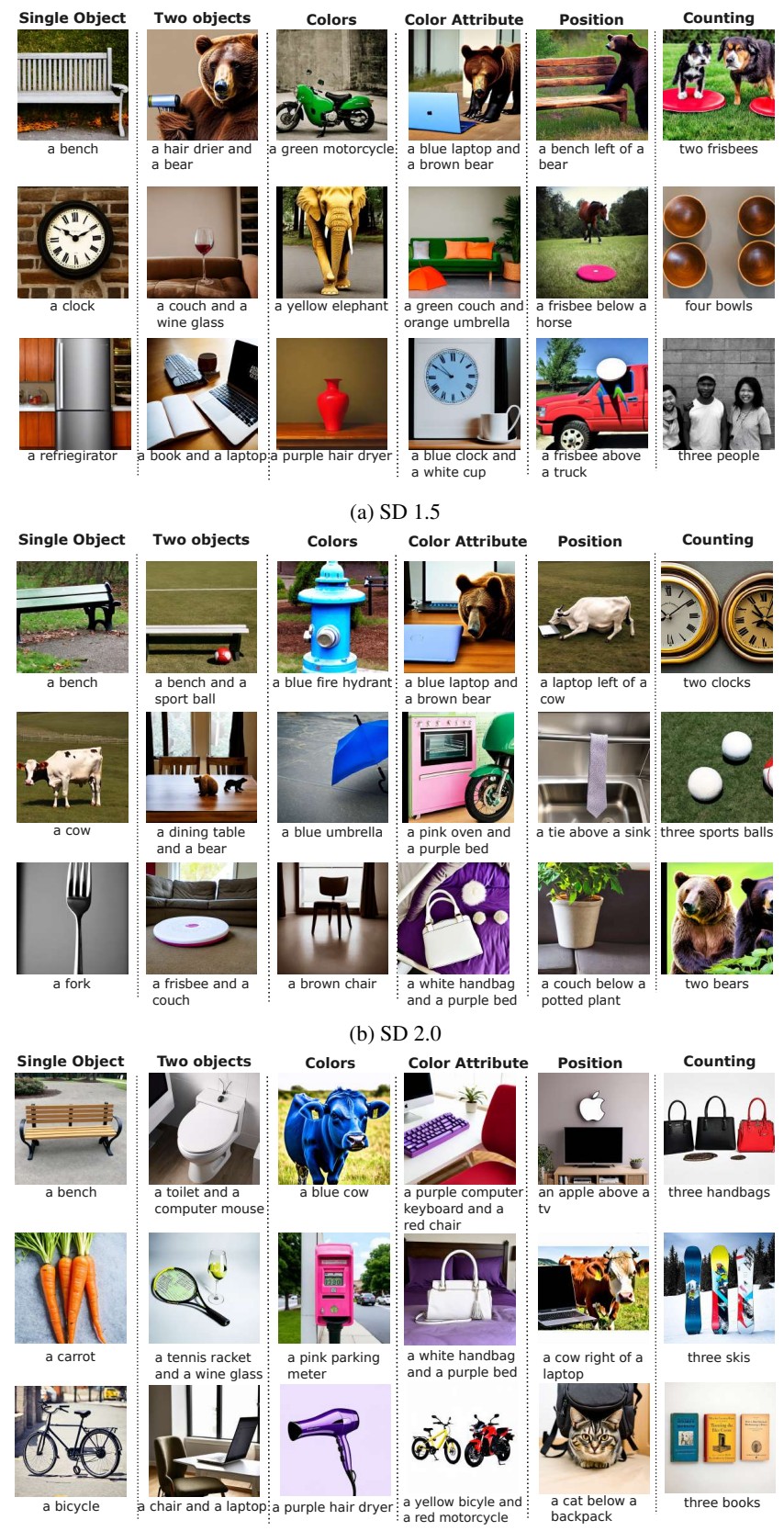

Figure D.4: **More examples from SELF-BENCH.** Representative samples illustrating the range of tasks and model outputs used in our benchmark.

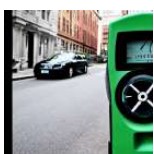

**GT**: a black car and a green parking meter
**Prediction**: a green car and a black parking meter

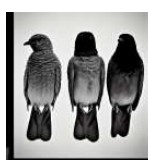

**GT**: three birds
**Prediction**: four birds

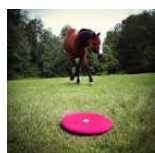

**GT**: a frisbee below a horse
**Prediction**: a frisbee above a horse

(a) SD 1.5

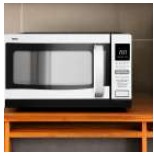

**GT**: A microwave and a bench
**Prediction**: A microwave and a cup

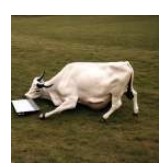

**GT**: a laptop left of a cow
**Prediction**: a laptop right of a cow



**GT**: a green skateboard
**Prediction**: a yellow skateboard

(b) SD 2.0

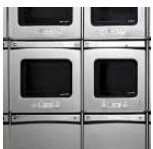

**GT**: four microwaves
**Prediction**: three microwaves

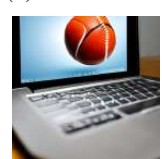

**GT**: a laptop below a sports ball
**Prediction**: a laptop above a sports ball

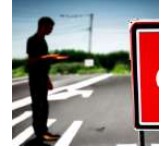

**GT**:a person and a stop sign
**Prediction**: a skateboard and a stop sign

(c) SD 3-m

Figure D.5: SELF-BENCH *in-Domain* **prediction failure cases.** Examples where the model fails despite being evaluated in an in-domain setting.

# E  Additional results

## E.1  Full results in compositional benchmarks

Table E.1, E.2, E.3, E.4, E.5, E.6, E.7, E.8, E.9 and E.10 report quantitative results for Winoground [49], COLA [40], EQBench [57], VisMin [2], MMVP [50], ARO [63], CLEVR [23], SugarCrepe [16], WhatsUP [24], and SPEC [36], respectively. Figure E.1 provides an overview of results across all prior benchmarks considered in our study.

**SELF-BENCH.** Table E.11 summarizes performance on SELF-BENCH. Given its high accuracy, one might suspect data bias in SELF-BENCH or selection bias in the model. To address this, we report both macro and micro accuracy in Figure E.2, which also provides a comprehensive overview of performance on SELF-BENCH. Figure E.3 shows performance degradation in *cross-domain* settings. Table E.12, Table E.13, and Table E.14 present fine-grained performance across color, position, and counting tasks, respectively.

**Compositional Benchmarks vs SELF-BENCH.** Figure E.4 compares results from existing compositional benchmarks with those from SELF-BENCH, revealing a clear domain shift across all tasks (see gray background).

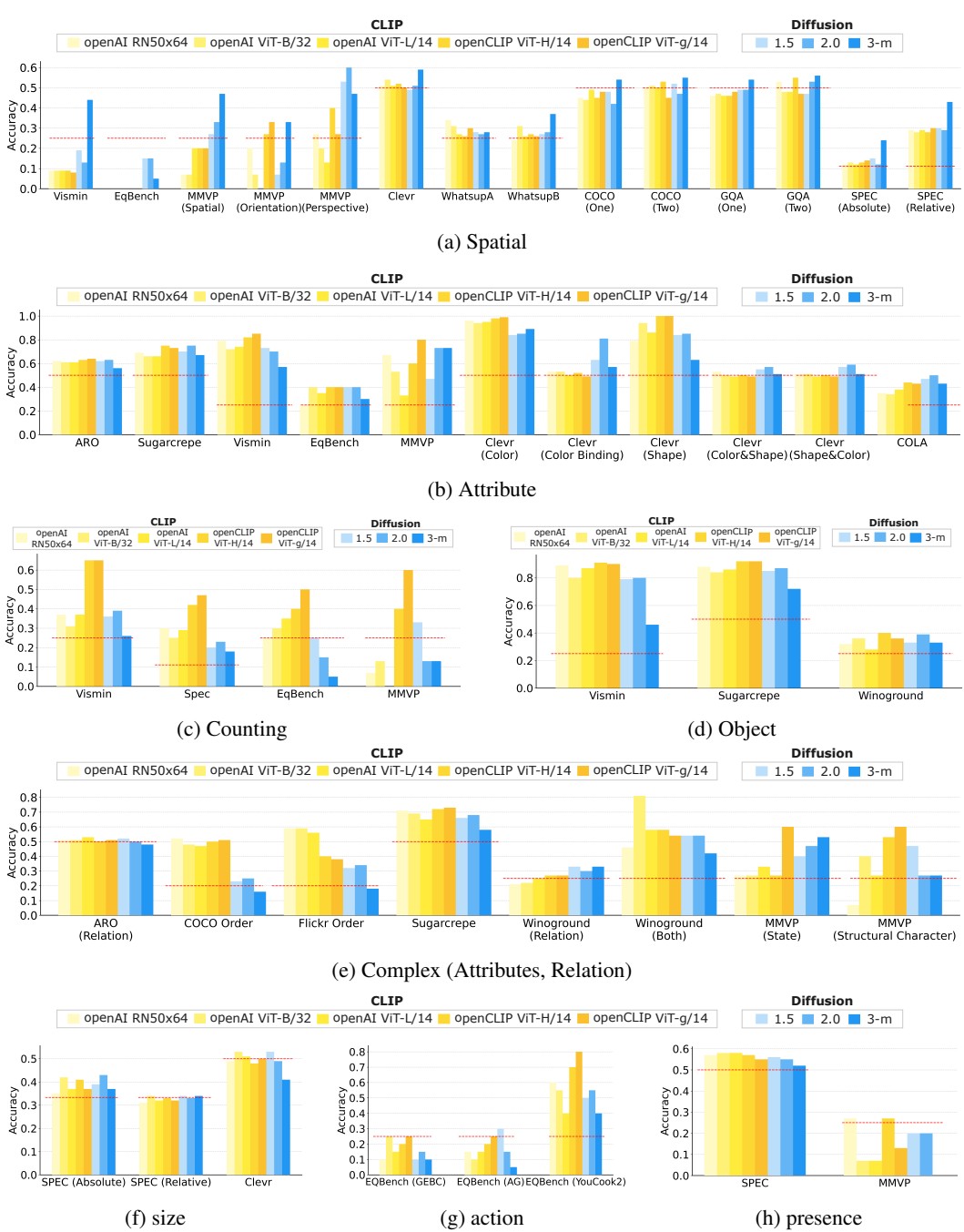

Figure E.1: **Overview of performance on compositional benchmarks beyond the four main categories.** Summarizes model accuracy on additional compositional tasks, highlighting generalization beyond the core evaluation categories. The red dotted horizontal line represents the random chance level.

Table E.1: Image-to-text retrieval results on Winoground [49].

| Version | Timesteps | Resolution | Method | All | Both | Object | Relation |
|---|---|---|---|---|---|---|---|
| CLIP RN50x64 | | 224 | | 0.27 | 0.46 | 0.32 | 0.21 |
| CLIP ViT-B/32 | | 224 | | 0.31 | **0.81** | 0.36 | 0.22 |
| CLIP ViT-L/14 | - | 224 | cosine-sim | 0.28 | 0.58 | 0.28 | 0.25 |
| openCLIP ViT-H/14 | | 224 | | 0.34 | 0.58 | **0.40** | 0.27 |
| openCLIP ViT-G/14 | | 224 | | 0.32 | 0.54 | 0.36 | 0.27 |
| SD 1.5 | 30 | 512 | zero-shot | 0.32 | 0.54 | 0.33 | 0.33 |
| | | | discffusion | 0.20 | 0.35 | 0.27 | 0.18 |
| SD 2.0 | 30 | 512 | zero-shot | **0.36** | 0.54 | 0.39 | 0.30 |
| | | | discffusion | 0.3 | 0.54 | 0.33 | 0.25 |
| SD 3-m | 30 | 1024 | zero-shot | 0.34 | 0.42 | 0.33 | 0.33 |
| | | | discffusion | 0.34 | 0.46 | 0.34 | **0.34** |

Table E.2: Image-to-Text retrieval accuracy on COLA [40].

| Version | Timesteps | Resolution | Method | Cola multi |
|---|---|---|---|---|
| CLIP RN50x64 | | 224 | | 0.35 |
| CLIP ViT-B/32 | | 224 | | 0.34 |
| CLIP ViT-L/14 | - | 224 | cosine-sim | 0.38 |
| openCLIP ViT-H/14 | | 224 | | 0.44 |
| openCLIP ViT-G/14 | | 224 | | 0.43 |
| SD 1.5 | 30 | 512 | zero-shot | 0.47 |
| | | | discffusion | 0.33 |
| SD 2.0 | 30 | 512 | zero-shot | **0.50** |
| | | | discffusion | 0.44 |
| SD 3-m | 30 | 1024 | zero-shot | 0.43 |
| | | | discffusion | 0.43 |

Table E.3: Image-to-text retrieval accuracy on EQbench [57].

| Version | Timesteps | Resolution | Method | EQ-YouCook2 | EQ-GEBC | EQ-AG | Attribute | EQ-Kubric Counting | Location | EQ-SD |
|---|---|---|---|---|---|---|---|---|---|---|
| CLIP RN50x64 | | 224 | | 0.6 | 0.1 | 0.15 | 0.25 | 0.25 | 0.0 | 0.9 |
| CLIP ViT-B/32 | | 224 | | 0.55 | **0.25** | 0.1 | 0.4 | 0.3 | 0.0 | 0.85 |
| CLIP ViT-L/14 | - | 224 | cosine-sim | 0.4 | 0.15 | 0.15 | 0.35 | 0.35 | 0.0 | 0.85 |
| openCLIP ViT-H/14 | | 224 | | 0.7 | 0.2 | 0.2 | 0.4 | 0.4 | 0.0 | **0.95** |
| openCLIP ViT-G/14 | | 224 | | **0.8** | **0.25** | 0.25 | 0.4 | **0.5** | 0.0 | 0.9 |
| SD 1.5 | 30 | 512 | zero-shot | 0.5 | 0.1 | **0.3** | 0.4 | 0.25 | **0.15** | 0.9 |
| | | | discffusion | 0.5 | 0.1 | 0.1 | 0.2 | 0.1 | 0.0 | 0.8 |
| SD 2.0 | 30 | 512 | zero-shot | 0.55 | 0.15 | 0.15 | 0.4 | 0.15 | **0.15** | 0.9 |
| | | | discffusion | 0.55 | 0.2 | 0.1 | 0.4 | 0.15 | 0.05 | 0.75 |
| SD 3-m | 30 | 1024 | zero-shot | 0.4 | 0.1 | 0.05 | 0.3 | 0.05 | 0.05 | 0.6 |
| | | | discffusion | 0.35 | 0.1 | 0.0 | 0.3 | 0.05 | 0.05 | 0.55 |

Table E.4: Image-to-text retrieval accuracy on Vismin[2].

| Version | Timesteps | Resolution | Method | Relation | Attribute | Object | Counting |
|---|---|---|---|---|---|---|---|
| CLIP RN50x64 | | 224 | | 0.09 | 0.79 | 0.89 | 0.37 |
| CLIP ViT-B/32 | | 224 | | 0.09 | 0.72 | 0.80 | 0.31 |
| CLIP ViT-L/14 | - | 224 | cosine-sim | 0.09 | 0.74 | 0.87 | 0.37 |
| openCLIP ViT-H/14 | | 224 | | 0.09 | 0.82 | **0.91** | **0.65** |
| openCLIP ViT-G/14 | | 224 | | 0.08 | **0.85** | 0.90 | **0.65** |
| SD 1.5 | 30 | 512 | zero-shot | 0.19 | 0.73 | 0.79 | 0.36 |
| | | | discffusion | 0.07 | 0.58 | 0.66 | 0.13 |
| SD 2.0 | 30 | 512 | zero-shot | 0.13 | 0.70 | 0.80 | 0.39 |
| | | | discffusion | 0.09 | 0.71 | 0.79 | 0.31 |
| SD 3-m | 30 | 1024 | zero-shot | **0.44** | 0.57 | 0.46 | 0.26 |
| | | | discffusion | **0.44** | 0.57 | 0.48 | 0.26 |

Table E.5: Image-to-Text retrieval accuracy on MMVP-VLM [50].

| Version | Timesteps | Resolution | Method | Camera Perspective | Color | Orientation | Presence | Quantity | Spatial | State | Structural Character | Text |
|---|---|---|---|---|---|---|---|---|---|---|---|---|
| CLIP RN50x64 | | 224 | | 0.27 | 0.67 | 0.2 | **0.27** | 0.07 | 0.07 | 0.27 | 0.07 | 0.4 |
| CLIP ViT-B/32 | | 224 | | 0.2 | 0.53 | 0.07 | 0.07 | 0.13 | 0.07 | 0.27 | 0.4 | **0.33** |
| CLIP ViT-L/14 | - | 224 | cosine-sim | 0.13 | 0.33 | 0.0 | 0.07 | 0.0 | 0.2 | 0.33 | 0.27 | 0.27 |
| openCLIP ViT-H/14 | | 224 | | 0.4 | 0.6 | 0.27 | **0.27** | 0.4 | 0.2 | 0.27 | 0.53 | 0.13 |
| openCLIP ViT-G/14 | | 224 | | 0.27 | **0.8** | 0.13 | **0.33** | 0.6 | 0.2 | **0.6** | **0.6** | 0.27 |
| SD 1.5 | 30 | 512 | zero-shot | 0.53 | 0.47 | 0.07 | 0.2 | **0.33** | 0.27 | 0.4 | 0.47 | **0.33** |
| | | | discffusion | 0.4 | 0.4 | 0.0 | 0.13 | 0.2 | 0.13 | 0.2 | 0.13 | 0.07 |
| SD 2.0 | 30 | 512 | zero-shot | **0.6** | 0.73 | 0.13 | 0.2 | 0.13 | **0.33** | 0.47 | 0.27 | 0.27 |
| | | | discffusion | 0.47 | 0.67 | 0.27 | 0.07 | 0.07 | 0.13 | 0.4 | 0.2 | 0.2 |
| SD 3-m | 30 | 1024 | zero-shot | 0.47 | 0.67 | **0.33** | 0.0 | 0.13 | 0.47 | 0.53 | 0.27 | 0.0 |
| | | | discffusion | 0.33 | 0.73 | 0.2 | 0.2 | 0.13 | 0.4 | 0.33 | 0.27 | 0.2 |

Table E.6: Image-to-Text retrieval accuracy on ARO [63].

| Version | Timesteps | Resolution | Method | VG relation | VG Attribution | Flickr30k order | COCO order |
|---|---|---|---|---|---|---|---|
| CLIP RN50x64 | | 224 | | 0.51 | 0.62 | 0.59 | 0.52 |
| CLIP ViT-B/32 | | 224 | | 0.51 | 0.61 | 0.59 | 0.48 |
| CLIP ViT-L/14 | - | 224 | cosine-sim | 0.53 | 0.61 | 0.56 | 0.47 |
| openCLIP ViT-H/14 | | 224 | | 0.50 | 0.63 | 0.40 | 0.33 |
| openCLIP ViT-G/14 | | 224 | | 0.51 | 0.64 | 0.38 | 0.33 |
| SD 1.5 | 30 | 512 | zero-shot | 0.52 | 0.62 | 0.32 | 0.23 |
| | | | discffusion | **0.62** | 0.67 | **0.85** | 0.72 |
| SD 2.0 | 30 | 512 | zero-shot | 0.50 | 0.63 | 0.34 | 0.25 |
| | | | discffusion | 0.58 | **0.73** | 0.77 | 0.58 |
| SD 3-m | 30 | 1024 | zero-shot | 0.48 | 0.56 | 0.18 | 0.16 |
| | | | discffusion | 0.49 | 0.57 | 0.20 | 0.17 |

Table E.7: Image-to-text retrieval accuracy on CLEVR [23].

| Version | Timesteps | Resolution | Method | All | pair binding size | pair binding color | recognition color | recognition shape | spatial | binding color shape | binding shape color |
|---|---|---|---|---|---|---|---|---|---|---|---|
| CLIP RN50x64 | | 224 | | 0.60 | 0.35 | 0.53 | 0.96 | 0.79 | 0.51 | 0.53 | 0.51 |
| CLIP ViT-B/32 | | 224 | | 0.64 | 0.50 | 0.53 | 0.94 | 0.94 | 0.54 | 0.50 | 0.51 |
| CLIP ViT-L/14 | - | 224 | cosine-sim | 0.64 | 0.70 | 0.50 | 0.95 | 0.86 | 0.51 | 0.49 | 0.49 |
| openCLIP ViT-H/14 | | 224 | | 0.67 | 0.66 | 0.52 | 0.98 | **1.0** | 0.52 | 0.50 | 0.50 |
| openCLIP ViT-G/14 | | 224 | | 0.65 | 0.60 | 0.49 | **0.99** | **1.0** | 0.50 | 0.49 | 0.49 |
| SD 1.5 | 30 | 512 | zero-shot | 0.66 | 0.67 | 0.63 | 0.84 | 0.84 | 0.49 | 0.55 | 0.57 |
| | | | discffusion | **0.73** | **0.81** | 0.64 | 0.86 | 0.88 | **0.70** | 0.55 | 0.59 |
| SD 2.0 | 30 | 512 | zero-shot | 0.69 | 0.61 | 0.81 | 0.85 | 0.88 | 0.51 | 0.57 | 0.59 |
| | | | discffusion | **0.73** | 0.66 | **0.86** | 0.89 | 0.89 | 0.63 | **0.59** | **0.61** |
| SD 3-m | 30 | 1024 | zero-shot | 0.56 | 0.52 | 0.57 | 0.63 | 0.59 | 0.59 | 0.51 | 0.51 |
| | | | discffusion | 0.56 | 0.53 | 0.58 | 0.64 | 0.59 | 0.59 | 0.51 | 0.51 |

Table E.8: Image-to-Text retrieval accuracy on SugarCrepe [16].

| Version | Timesteps | Resolution | Method | attribute | object | relation |
|---|---|---|---|---|---|---|
| CLIP RN50x64 | | 224 | | 0.69 | 0.88 | 0.71 |
| CLIP ViT-B/32 | | 224 | | 0.66 | 0.84 | 0.69 |
| CLIP ViT-L/14 | - | 224 | cosine-sim | 0.66 | 0.86 | 0.65 |
| openCLIP ViT-H/14 | | 224 | | 0.75 | **0.92** | 0.72 |
| openCLIP ViT-G/14 | | 224 | | 0.73 | **0.92** | 0.73 |
| SD 1.5 | 30 | 512 | zero-shot | 0.70 | 0.85 | 0.66 |
| | | | discffusion | 0.80 | 0.67 | 0.59 |
| SD 2.0 | 30 | 512 | zero-shot | 0.75 | 0.87 | 0.68 |
| | | | discffusion | **0.86** | 0.87 | **0.76** |
| SD 3-m | 30 | 1024 | zero-shot | 0.67 | 0.72 | 0.58 |
| | | | discffusion | 0.68 | 0.72 | 0.60 |

Table E.9: Image-to-Text retrieval accuracy on WhatsUP [24].

| Version | Timesteps | Resolution | Method | WhatsUp A | WhatsUp B | COCO-spatial (one) | COCO-spatial (two) | GQA-spatial (one) | GQA-spatial (two) |
|---|---|---|---|---|---|---|---|---|---|
| CLIP RN50x64 | | 224 | | **0.34** | 0.24 | 0.45 | 0.50 | 0.46 | 0.53 |
| CLIP ViT-B/32 | | 224 | | 0.31 | 0.31 | 0.44 | 0.51 | 0.47 | 0.48 |
| CLIP ViT-L/14 | - | 224 | cosine-sim | 0.27 | 0.26 | 0.49 | 0.50 | 0.46 | 0.48 |
| openCLIP ViT-H/14 | | 224 | | 0.26 | 0.27 | 0.45 | 0.53 | 0.46 | 0.55 |
| openCLIP ViT-G/14 | | 224 | | 0.30 | 0.26 | 0.48 | 0.45 | 0.48 | 0.47 |
| SD 1.5 | 30 | 512 | zero-shot | 0.28 | 0.27 | 0.48 | 0.52 | 0.49 | 0.47 |
| | | | discffusion | 0.23 | 0.32 | **0.69** | 0.52 | **0.55** | 0.56 |
| SD 2.0 | 30 | 512 | zero-shot | 0.27 | 0.28 | 0.42 | 0.47 | 0.49 | 0.53 |
| | | | discffusion | 0.25 | 0.21 | 0.59 | **0.58** | **0.55** | **0.58** |
| SD 3-m | 30 | 1024 | zero-shot | 0.28 | **0.37** | 0.54 | 0.55 | 0.54 | 0.56 |
| | | | discffusion | 0.31 | **0.37** | 0.55 | 0.57 | 0.54 | 0.54 |

Table E.10: Image-to-text retrieval accuracy on SPEC [36].

| Version | Timesteps | Resolution | Method | Absolute Size | Absolute Spatial | Count | Existence | Relative Size | Relative Spatial |
|---|---|---|---|---|---|---|---|---|---|
| CLIP RN50x64 | | 224 | | 0.35 | 0.12 | 0.30 | 0.57 | 0.31 | 0.29 |
| CLIP ViT-B/32 | | 224 | | 0.42 | 0.13 | 0.25 | **0.58** | 0.34 | 0.28 |
| CLIP ViT-L/14 | - | 224 | cosine-sim | 0.37 | 0.12 | 0.29 | **0.58** | 0.32 | 0.29 |
| openCLIP ViT-H/14 | | 224 | | 0.41 | 0.13 | 0.42 | 0.57 | 0.33 | 0.28 |
| openCLIP ViT-G/14 | | 224 | | 0.37 | 0.14 | **0.47** | 0.55 | 0.32 | 0.30 |
| SD 1.5 | 30 | 512 | zero-shot | 0.39 | 0.15 | 0.20 | 0.56 | 0.34 | 0.30 |
| | | | discffusion | 0.33 | 0.11 | 0.12 | 0.52 | 0.33 | 0.26 |
| SD 2.0 | 30 | 512 | zero-shot | **0.43** | 0.12 | 0.23 | 0.55 | 0.33 | 0.29 |
| | | | discffusion | 0.33 | 0.11 | 0.12 | 0.52 | 0.33 | 0.26 |
| SD 3-m | 30 | 1024 | zero-shot | 0.37 | 0.24 | 0.18 | 0.52 | 0.34 | **0.43** |
| | | | discffusion | 0.33 | 0.14 | 0.14 | 0.51 | 0.33 | 0.32 |

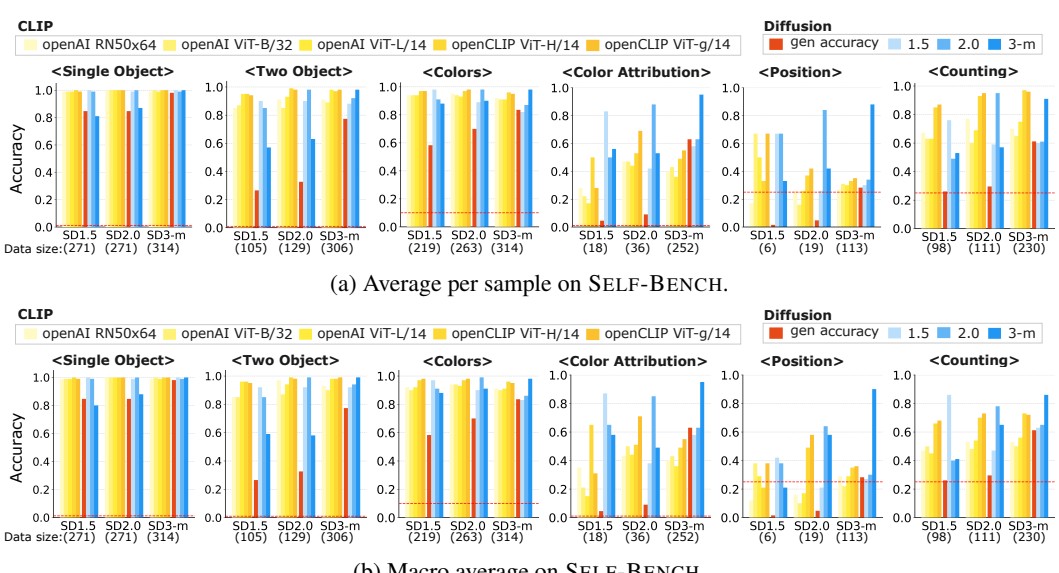

(a) Average per sample on SELF-BENCH.

(b) Macro average on SELF-BENCH.

Figure E.2: **Micro accuracy (Top) and Macro accuracy (Bottom) on SELF-BENCH.** We evaluate CLIP and SD models in our SELF-BENCH. The X-axis represents the model used for generating the dataset, with the number of images for each dataset indicated below. The results clearly show that models perform well only when evaluated *in-domain*, not *cross-domain*. The red dotted horizontal line represents the random chance level.

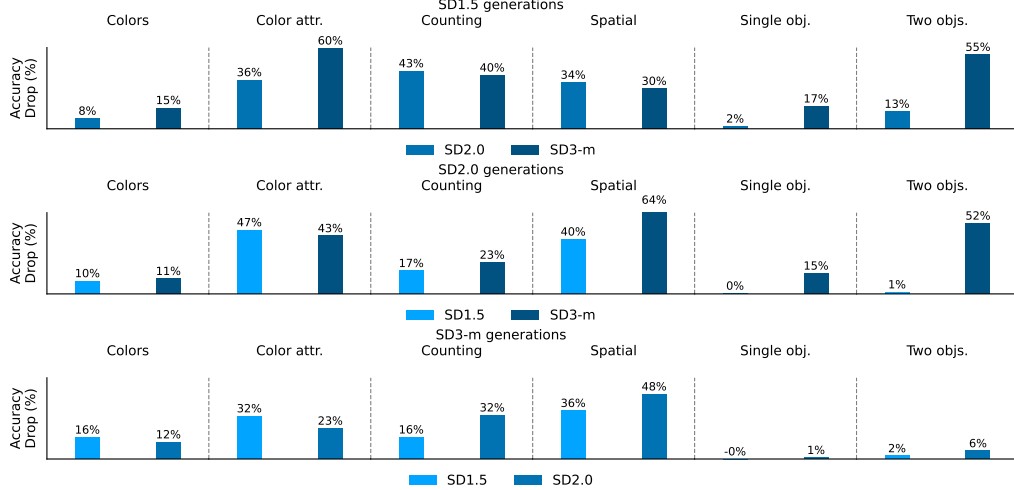

Figure E.3: **SELF-BENCH *cross-domain* drop rate.** WE show the performance drop when evaluating models in *cross-domain* settings.

Table E.11: Complete image-to-text retrieval accuracy results on SELF-BENCH across all tasks. Bold entries indicate in-domain evaluations for the model.

| Version | Timesteps | Resolution | Method | Single Object Full | Single Object Correct | Two Objects Full | Two Objects Correct | Colors Full | Colors Correct | Color Attribution Full | Color Attribution Correct | Position Full | Position Correct | Counting Full | Counting Correct |
|---|---|---|---|---|---|---|---|---|---|---|---|---|---|---|---|
| CLIP RN50x64 | - | 224 | | 0.97 | 0.99 | 0.46 | 0.85 | 0.86 | 0.94 | 0.28 | 0.28 | 0.33 | 0.17 | 0.47 | 0.67 |
| CLIP ViT-B/32 | - | 224 | | 0.96 | 0.99 | 0.46 | 0.87 | 0.87 | 0.94 | 0.25 | 0.22 | 0.24 | 0.67 | 0.52 | 0.63 |
| CLIP ViT-L/14 | - | 224 | cosine-sim | 0.97 | 0.99 | 0.54 | 0.95 | 0.87 | 0.94 | 0.29 | 0.17 | 0.31 | 0.5 | 0.49 | 0.63 |
| openCLIP ViT-H/14 | - | 224 | | 0.97 | 1.0 | 0.52 | 0.95 | 0.89 | 0.97 | 0.31 | 0.5 | 0.30 | 0.33 | 0.48 | 0.85 |
| openCLIP ViT-G/14 | - | 224 | | 0.97 | 0.99 | 0.51 | 0.94 | 0.87 | 0.97 | 0.35 | 0.28 | 0.34 | 0.67 | 0.49 | 0.87 |
| **SD 1.5 (in-domain)** | 30 | 512 | # of samples | 320 | 271 | 396 | 105 | 376 | 219 | 400 | 18 | 400 | 6 | 320 | 98 |
| | | | zero-shot | 0.98 | 1.0 | 0.69 | 0.90 | 0.93 | 0.98 | 0.56 | 0.83 | 0.49 | 0.67 | 0.65 | 0.76 |
| | | | discffusion | 0.88 | 0.86 | 0.36 | 0.59 | 0.75 | 0.77 | 0.26 | 0.5 | 0.36 | 0.33 | 0.5 | 0.46 |
| SD 2.0 (cross-domain) | 30 | 512 | zero-shot | 0.96 | 0.99 | 0.48 | 0.85 | 0.82 | 0.91 | 0.26 | 0.5 | 0.36 | 0.67 | 0.46 | 0.49 |
| | | | discffusion | 0.96 | 0.98 | 0.47 | 0.83 | 0.80 | 0.91 | 0.24 | 0.56 | 0.35 | 0.33 | 0.40 | 0.59 |
| SD 3-m (cross-domain) | 30 | 512 | zero-shot | 0.82 | 0.86 | 0.26 | 0.50 | 0.68 | 0.75 | 0.16 | 0.44 | 0.31 | 0.83 | 0.39 | 0.44 |
| | | | discffusion | 0.82 | 0.87 | 0.24 | 0.50 | 0.68 | 0.74 | 0.17 | 0.44 | 0.31 | 0.83 | 0.40 | 0.46 |
| | | 1024 | zero-shot | 0.78 | 0.81 | 0.29 | 0.57 | 0.88 | 0.88 | 0.26 | 0.56 | 0.29 | 0.33 | 0.37 | 0.53 |
| | | | discffusion | 0.80 | 0.83 | 0.29 | 0.57 | 0.82 | 0.88 | 0.26 | 0.56 | 0.29 | 0.33 | 0.37 | 0.52 |
| CLIP RN50x64 | - | 224 | | 0.99 | 1.0 | 0.61 | 0.91 | 0.93 | 0.95 | 0.30 | 0.47 | 0.30 | 0.26 | 0.50 | 0.77 |
| CLIP ViT-B/32 | - | 224 | | 1.0 | 1.0 | 0.54 | 0.85 | 0.92 | 0.94 | 0.35 | 0.47 | 0.22 | 0.16 | 0.50 | 0.60 |
| CLIP ViT-L/14 | - | 224 | cosine-sim | 0.99 | 1.0 | 0.64 | 0.93 | 0.91 | 0.93 | 0.28 | 0.44 | 0.26 | 0.26 | 0.52 | 0.69 |
| openCLIP ViT-H/14 | - | 224 | | 1.0 | 1.0 | 0.69 | 0.99 | 0.94 | 0.97 | 0.44 | 0.53 | 0.44 | 0.37 | 0.53 | 0.93 |
| openCLIP ViT-G/14 | - | 224 | | 1.0 | 1.0 | 0.65 | 0.98 | 0.94 | 0.98 | 0.45 | 0.69 | 0.38 | 0.42 | 0.53 | 0.95 |
| **SD 2.0 (in-domain)** | 30 | 512 | # of samples | 320 | 271 | 396 | 129 | 376 | 263 | 400 | 36 | 400 | 19 | 320 | 111 |
| | | | zero-shot | 1.0 | 1.0 | 0.82 | 0.98 | 0.97 | 0.98 | 0.70 | 0.88 | 0.63 | 0.84 | 0.78 | 0.95 |
| | | | discffusion | 0.99 | 1.0 | 0.78 | 0.98 | 0.92 | 0.97 | 0.56 | 0.83 | 0.61 | 0.89 | 0.64 | 0.89 |
| SD 1.5 (cross-domain) | 30 | 512 | zero-shot | 0.99 | 0.99 | 0.61 | 0.90 | 0.85 | 0.89 | 0.37 | 0.42 | 0.28 | 0.26 | 0.49 | 0.59 |
| | | | discffusion | 0.79 | 0.76 | 0.28 | 0.62 | 0.62 | 0.60 | 0.18 | 0.28 | 0.21 | 0.32 | 0.45 | 0.31 |
| SD 3-m (cross-domain) | 30 | 512 | zero-shot | 0.89 | 0.91 | 0.35 | 0.57 | 0.78 | 0.80 | 0.19 | 0.39 | 0.33 | 0.63 | 0.46 | 0.51 |
| | | | discffusion | 0.89 | 0.91 | 0.34 | 0.56 | 0.80 | 0.81 | 0.20 | 0.42 | 0.32 | 0.63 | 0.48 | 0.51 |
| | | 1024 | zero-shot | 0.86 | 0.87 | 0.40 | 0.63 | 0.87 | 0.90 | 0.28 | 0.53 | 0.30 | 0.42 | 0.45 | 0.57 |
| | | | discffusion | 0.87 | 0.88 | 0.41 | 0.66 | 0.87 | 0.90 | 0.27 | 0.58 | 0.31 | 0.47 | 0.44 | 0.57 |
| CLIP RN50x64 | - | 224 | | 0.99 | 0.99 | 0.90 | 0.91 | 0.89 | 0.92 | 0.38 | 0.40 | 0.26 | 0.27 | 0.66 | 0.7 |
| CLIP ViT-B/32 | - | 224 | | 1.0 | 1.0 | 0.86 | 0.89 | 0.88 | 0.91 | 0.43 | 0.43 | 0.28 | 0.31 | 0.61 | 0.65 |
| CLIP ViT-L/14 | - | 224 | cosine-sim | 0.99 | 0.99 | 0.95 | 0.98 | 0.89 | 0.91 | 0.34 | 0.36 | 0.32 | 0.30 | 0.68 | 0.75 |
| openCLIP ViT-H/14 | - | 224 | | 1.0 | 1.0 | 0.95 | 0.97 | 0.91 | 0.96 | 0.47 | 0.49 | 0.36 | 0.33 | 0.84 | 0.97 |
| openCLIP ViT-G/14 | - | 224 | | 1.0 | 1.0 | 0.95 | 0.98 | 0.91 | 0.95 | 0.51 | 0.55 | 0.37 | 0.35 | 0.84 | 0.96 |
| **SD 3-m (in-domain)** | 30 | 1024 | # of samples | 320 | 314 | 396 | 306 | 376 | 314 | 400 | 252 | 400 | 113 | 320 | 230 |
| | | | zero-shot | 1.0 | 1.0 | 0.98 | 0.98 | 0.97 | 0.98 | 0.91 | 0.95 | 0.72 | 0.89 | 0.85 | 0.91 |
| | | | discffusion | 1.0 | 1.0 | 0.98 | 0.99 | 0.98 | 0.98 | 0.91 | 0.95 | 0.72 | 0.91 | 0.86 | 0.92 |
| SD 1.5 (cross-domain) | 30 | 512 | zero-shot | 1.0 | 1.0 | 0.87 | 0.88 | 0.78 | 0.82 | 0.53 | 0.58 | 0.32 | 0.30 | 0.57 | 0.60 |
| | | | discffusion | 0.85 | 0.84 | 0.75 | 0.54 | 0.60 | 0.61 | 0.27 | 0.25 | 0.25 | 0.28 | 0.47 | 0.55 |
| SD 2.0 (cross-domain) | 30 | 512 | zero-shot | 0.99 | 0.99 | 0.91 | 0.92 | 0.82 | 0.87 | 0.56 | 0.63 | 0.33 | 0.34 | 0.57 | 0.61 |
| | | | discffusion | 0.98 | 0.98 | 0.88 | 0.90 | 0.82 | 0.85 | 0.52 | 0.57 | 0.38 | 0.38 | 0.56 | 0.65 |

Table E.12: Fine-grained accuracy on SELF-BENCH *Colors*.

| Version | Timesteps | Resolution | Method | Red Full | Red Correct | Orange Full | Orange Correct | Yellow Full | Yellow Correct | Green Full | Green Correct | Blue Full | Blue Correct | Purple Full | Purple Correct | Pink Full | Pink Correct | Brown Full | Brown Correct | Black Full | Black Correct | White Full | White Correct | Macro Average Full | Macro Average Correct |
|---|---|---|---|---|---|---|---|---|---|---|---|---|---|---|---|---|---|---|---|---|---|---|---|---|---|
| CLIP RN50x64 | - | 224 | | 0.98 | 1.00 | 0.82 | 0.67 | 0.77 | 1.00 | 1.00 | 1.00 | 0.85 | 1.00 | 0.90 | 1.00 | 0.83 | 0.93 | 0.82 | 0.92 | 0.84 | 0.91 | 0.62 | 0.74 | 0.84 | 0.92 |
| CLIP ViT-B/32 | - | 224 | | 0.92 | 1.00 | 0.82 | 0.56 | 0.82 | 0.96 | 0.98 | 1.00 | 0.88 | 1.00 | 0.88 | 0.95 | 0.79 | 0.87 | 0.82 | 0.85 | 0.93 | 0.94 | 0.75 | 0.89 | 0.86 | 0.90 |
| CLIP ViT-L/14 | - | 224 | cosine-sim | 0.94 | 0.97 | 0.82 | 0.56 | 0.68 | 0.92 | 1.00 | 1.00 | 0.82 | 0.95 | 0.93 | 1.00 | 0.88 | 1.00 | 0.89 | 1.00 | 0.91 | 0.88 | 0.81 | 0.89 | 0.87 | 0.92 |
| openCLIP ViT-H/14 | - | 224 | | 1.00 | 1.00 | 0.93 | 0.89 | 0.70 | 0.96 | 0.98 | 1.00 | 0.88 | 1.00 | 0.90 | 0.95 | 0.88 | 1.00 | 0.86 | 1.00 | 0.86 | 0.88 | 0.88 | 1.00 | 0.89 | 0.97 |
| openCLIP ViT-G/14 | - | 224 | | 0.96 | 1.00 | 0.96 | 1.00 | 0.70 | 0.96 | 0.98 | 1.00 | 0.85 | 1.00 | 0.90 | 1.00 | 0.92 | 1.00 | 0.86 | 1.00 | 0.82 | 0.88 | 0.78 | 0.95 | 0.87 | 0.98 |
| **SD 1.5** | 30 | 512 | # of samples | 52 | 31 | 28 | 9 | 44 | 26 | 44 | 29 | 40 | 22 | 40 | 22 | 24 | 15 | 28 | 13 | 44 | 33 | 32 | 19 | - | - |
| | | | zero-shot | 0.98 | 1.0 | 0.96 | 0.89 | 0.93 | 1.0 | 0.98 | 1.0 | 0.95 | 0.95 | 0.98 | 1.0 | 0.88 | 1.0 | 0.68 | 0.92 | 0.97 | 0.91 | 0.53 | 0.74 | 0.92 | 0.97 |
| | | | discffusion | 0.92 | 0.94 | 0.89 | 0.78 | 0.95 | 1.0 | 0.80 | 0.59 | 0.80 | 0.91 | 0.88 | 0.91 | 0.75 | 0.87 | 0.21 | 0.46 | 0.52 | 0.52 | 0.53 | 0.74 | 0.72 | 0.77 |
| SD 2.0 | 30 | 512 | zero-shot | 0.81 | 0.94 | 0.93 | 0.89 | 0.68 | 0.88 | 0.95 | 0.97 | 0.78 | 0.86 | 0.83 | 0.86 | 0.88 | 1.0 | 0.64 | 0.77 | 0.86 | 0.81 | 0.88 | 1.0 | 0.82 | 0.91 |
| | | | discffusion | 0.81 | 0.97 | 0.71 | 1.0 | 0.64 | 0.88 | 0.95 | 0.97 | 0.75 | 0.86 | 0.80 | 0.82 | 0.88 | 0.93 | 0.71 | 0.92 | 0.86 | 0.82 | 0.88 | 1.0 | 0.8 | 0.92 |
| SD 3-m | 30 | 1024 | zero-shot | 0.71 | 0.74 | 0.79 | 0.67 | 0.82 | 1.0 | 0.95 | 1.0 | 0.85 | 0.95 | 0.9 | 1.0 | 0.88 | 1.0 | 0.79 | 0.85 | 0.59 | 0.61 | 0.91 | 1.0 | 0.82 | 0.88 |
| | | | discffusion | 0.71 | 0.74 | 0.75 | 0.67 | 0.84 | 1.0 | 0.98 | 1.0 | 0.85 | 0.95 | 0.90 | 1.0 | 0.88 | 1.0 | 0.82 | 0.85 | 0.64 | 0.64 | 0.91 | 1.0 | 0.83 | 0.89 |
| CLIP RN50x64 | - | 224 | | 0.96 | 0.94 | 0.86 | 0.78 | 0.91 | 0.97 | 1.00 | 1.00 | 1.00 | 1.00 | 1.00 | 1.00 | 1.00 | 1.00 | 0.82 | 0.94 | 0.80 | 0.84 | 0.88 | 0.96 | 0.92 | 0.94 |
| CLIP ViT-B/32 | - | 224 | | 0.90 | 0.89 | 0.86 | 0.78 | 0.91 | 0.89 | 0.91 | 1.00 | 0.97 | 1.00 | 0.97 | 1.00 | 1.00 | 1.00 | 0.75 | 0.81 | 0.93 | 0.91 | 0.84 | 1.00 | 0.91 | 0.94 |
| CLIP ViT-L/14 | - | 224 | cosine-sim | 0.87 | 0.86 | 0.86 | 0.78 | 0.80 | 0.94 | 1.00 | 1.00 | 0.95 | 0.97 | 1.00 | 1.00 | 1.00 | 1.00 | 0.79 | 0.88 | 0.86 | 0.88 | 0.97 | 1.00 | 0.91 | 0.93 |
| openCLIP ViT-H/14 | - | 224 | | 0.92 | 0.89 | 0.93 | 0.89 | 0.84 | 0.97 | 1.00 | 1.00 | 0.97 | 1.00 | 0.97 | 1.00 | 1.00 | 1.00 | 0.89 | 1.00 | 0.98 | 1.00 | 0.97 | 1.00 | 0.95 | 0.97 |
| openCLIP ViT-G/14 | - | 224 | | 0.96 | 0.94 | 0.96 | 0.94 | 0.82 | 0.97 | 1.00 | 1.00 | 0.97 | 1.00 | 0.97 | 1.00 | 1.00 | 1.00 | 0.86 | 0.94 | 0.93 | 0.97 | 0.94 | 1.00 | 0.94 | 0.98 |
| **SD 2.0** | 30 | 512 | # of samples | 52 | 35 | 28 | 18 | 44 | 35 | 44 | 36 | 40 | 34 | 40 | 22 | 24 | 12 | 28 | 16 | 44 | 32 | 32 | 23 | - | - |
| | | | zero-shot | 0.94 | 0.97 | 1.0 | 1.0 | 0.93 | 1.0 | 0.95 | 0.94 | 0.98 | 1.0 | 0.98 | 1.0 | 1.0 | 1.0 | 0.89 | 0.94 | 1.0 | 1.0 | 1.0 | 1.0 | 0.97 | 0.99 |
| | | | discffusion | 0.92 | 0.94 | 0.96 | 1.0 | 0.80 | 0.94 | 0.95 | 0.92 | 0.93 | 1.0 | 0.95 | 1.0 | 1.0 | 0.92 | 0.82 | 1.0 | 0.98 | 0.97 | 0.94 | 1.0 | 0.93 | 0.97 |
| SD 1.5 | 30 | 512 | zero-shot | 0.67 | 0.69 | 0.96 | 0.94 | 0.84 | 0.94 | 0.95 | 0.94 | 0.85 | 0.88 | 0.90 | 0.95 | 0.92 | 0.92 | 0.71 | 0.88 | 0.84 | 0.88 | 0.88 | 0.96 | 0.85 | 0.90 |
| | | | discffusion | 0.81 | 0.77 | 0.68 | 0.56 | 0.89 | 0.97 | 0.52 | 0.39 | 0.83 | 0.82 | 0.48 | 0.36 | 0.79 | 0.92 | 0.21 | 0.44 | 0.41 | 0.28 | 0.50 | 0.59 | 0.61 | 0.59 |
| SD 3-m | 30 | 1024 | zero-shot | 0.73 | 0.74 | 0.96 | 0.89 | 0.84 | 0.86 | 1.0 | 1.0 | 0.98 | 1.0 | 0.95 | 1.0 | 0.96 | 0.92 | 0.82 | 0.88 | 0.74 | 0.78 | 0.88 | 1.0 | 0.89 | 0.91 |
| | | | discffusion | 0.73 | 0.74 | 0.93 | 0.89 | 0.84 | 0.89 | 1.0 | 1.0 | 0.98 | 1.0 | 0.95 | 1.0 | 0.96 | 0.92 | 0.70 | 0.78 | 0.88 | 1.0 | 0.88 | 0.90 | | |
| CLIP RN50x64 | - | 224 | | 0.73 | 0.85 | 0.96 | 0.96 | 0.91 | 0.89 | 0.98 | 1.00 | 1.00 | 1.00 | 0.97 | 0.97 | 1.00 | 1.00 | 0.68 | 0.77 | 0.91 | 0.92 | 0.75 | 0.79 | 0.89 | 0.91 |
| CLIP ViT-B/32 | - | 224 | | 0.81 | 0.87 | 0.82 | 0.81 | 0.91 | 0.89 | 0.91 | 1.00 | 1.00 | 1.00 | 1.00 | 1.00 | 1.00 | 1.00 | 0.64 | 0.73 | 0.86 | 0.89 | 0.88 | 0.88 | 0.89 | 0.90 |
| CLIP ViT-L/14 | - | 224 | cosine-sim | 0.94 | 0.92 | 0.93 | 0.92 | 0.84 | 0.84 | 0.91 | 1.00 | 1.00 | 1.00 | 0.93 | 0.92 | 1.00 | 1.00 | 0.79 | 0.89 | 0.89 | 0.78 | 0.79 | 0.89 | 0.91 |  |
| openCLIP ViT-H/14 | - | 224 | | 0.81 | 0.95 | 1.00 | 1.00 | 0.89 | 0.89 | 0.95 | 1.00 | 1.00 | 1.00 | 0.93 | 0.92 | 1.00 | 1.00 | 0.71 | 0.82 | 0.98 | 1.00 | 0.91 | 1.00 | 0.92 | 0.96 |
| openCLIP ViT-G/14 | - | 224 | | 0.83 | 0.95 | 0.93 | 0.92 | 0.84 | 0.84 | 0.93 | 1.00 | 1.00 | 1.00 | 1.00 | 1.00 | 1.00 | 1.00 | 0.71 | 0.82 | 0.98 | 1.00 | 0.91 | 1.00 | 0.91 | 0.95 |
| **SD 3-m** | 30 | 1024 | # of samples | 52 | 39 | 28 | 26 | 44 | 38 | 44 | 34 | 40 | 39 | 40 | 36 | 24 | 18 | 28 | 22 | 44 | 38 | 32 | 24 | - | - |
| | | | zero-shot | 0.92 | 0.92 | 0.96 | 0.96 | 1.0 | 1.0 | 1.0 | 1.0 | 1.0 | 1.0 | 1.0 | 1.0 | 1.0 | 1.0 | 1.0 | 1.0 | 0.98 | 0.93 | 0.91 | 1.0 | 0.97 | 0.98 |
| | | | discffusion | 0.94 | 0.92 | 0.96 | 0.96 | 1.0 | 1.0 | 1.0 | 1.0 | 1.0 | 1.0 | 1.0 | 1.0 | 1.0 | 1.0 | 0.96 | 1.0 | 0.98 | 0.97 | 0.91 | 1.0 | 0.97 | 0.98 |
| SD 1.5 | 30 | 512 | zero-shot | 0.65 | 0.77 | 0.82 | 0.81 | 0.80 | 0.84 | 0.82 | 0.88 | 0.83 | 0.82 | 0.98 | 0.97 | 1.0 | 1.0 | 0.68 | 0.68 | 0.64 | 0.75 | 0.71 | 0.79 | 0.83 |  |
| | | | discffusion | 0.75 | 0.87 | 0.57 | 0.46 | 0.93 | 1.0 | 0.59 | 0.56 | 0.83 | 0.72 | 0.70 | 0.58 | 0.67 | 0.78 | 0.18 | 0.23 | 0.25 | 0.34 | 0.34 | 0.33 | 0.58 | 0.59 |
| SD 2.0 | 30 | 512 | zero-shot | 0.69 | 0.79 | 0.89 | 0.88 | 0.75 | 0.82 | 0.86 | 0.88 | 0.90 | 0.90 | 0.90 | 0.89 | 0.92 | 0.94 | 0.68 | 0.77 | 0.89 | 0.95 | 0.78 | 0.83 | 0.83 | 0.86 |
| | | | discffusion | 0.73 | 0.85 | 0.82 | 0.81 | 0.70 | 0.76 | 0.82 | 0.85 | 0.93 | 0.90 | 0.98 | 0.86 | 0.92 | 0.89 | 0.68 | 0.82 | 0.86 | 0.89 | 0.84 | 0.88 | 0.83 | 0.85 |

Table E.13: Fine-grained accuracy on SELF-BENCH *Position*

| Version | Timesteps | Resolution | Method | left of | | right of | | above | | below | | Macro Average | |
|---|---|---|---|---|---|---|---|---|---|---|---|---|---|
| | | | | Full | Correct | Full | Correct | Full | Correct | Full | Correct | Full | Correct |
| CLIP RN50x64 | - | 224 | | 0.33 | 0.00 | 0.37 | 0.00 | 0.38 | 0.50 | 0.25 | 0.00 | 0.33 | 0.12 |
| CLIP ViT-B/32 | - | 224 | | 0.14 | 0.00 | 0.19 | 0.00 | 0.40 | 0.50 | 0.20 | 1.00 | 0.23 | 0.38 |
| CLIP ViT-L/14 | - | 224 | cosine-sim | 0.28 | 0.00 | 0.43 | 0.00 | 0.29 | 0.50 | 0.23 | 0.67 | 0.31 | 0.29 |
| openCLIP ViT-H/14 | - | 224 | | 0.38 | 0.00 | 0.43 | 0.00 | 0.19 | 0.50 | 0.21 | 0.33 | 0.30 | 0.21 |
| openCLIP ViT-G/14 | - | 224 | | 0.41 | 0.00 | 0.41 | 0.00 | 0.27 | 0.50 | 0.29 | 1.00 | 0.34 | 0.38 |
| **SD 1.5** | 30 | 512 | # of samples | 76 | 1 | 108 | 0 | 104 | 2 | 112 | 3 | - | - |
| | | | zero-shot | 0.47 | 0.00 | 0.44 | 0.00 | 0.46 | 1.00 | 0.56 | 0.67 | 0.48 | 0.42 |
| | | | discffusion | 0.42 | 0.0 | 0.25 | 0.0 | 0.42 | 1.0 | 0.36 | 0.0 | 0.36 | 0.25 |
| SD 2.0 | 30 | 512 | zero-shot | 0.34 | 0.00 | 0.39 | 0.00 | 0.46 | 0.50 | 0.26 | 1.00 | 0.36 | 0.38 |
| | | | discffusion | 0.62 | 1.0 | 0.32 | 0.0 | 0.38 | 0.5 | 0.17 | 0.0 | 0.37 | 0.38 |
| SD 3-m | 30 | 1024 | zero-shot | 0.28 | 0.0 | 0.27 | 0.0 | 0.29 | 0.5 | 0.32 | 0.33 | 0.29 | 0.21 |
| | | | discffusion | 0.29 | 0.0 | 0.27 | 0.0 | 0.29 | 0.5 | 0.32 | 0.33 | 0.29 | 0.21 |
| CLIP RN50x64 | - | 224 | | 0.33 | 0.00 | 0.29 | 0.00 | 0.29 | 0.50 | 0.29 | 0.14 | 0.30 | 0.16 |
| CLIP ViT-B/32 | - | 224 | | 0.14 | 0.00 | 0.17 | 0.00 | 0.38 | 0.25 | 0.17 | 0.14 | 0.22 | 0.10 |
| CLIP ViT-L/14 | - | 224 | cosine-sim | 0.28 | 0.00 | 0.37 | 0.00 | 0.18 | 0.38 | 0.22 | 0.29 | 0.26 | 0.17 |
| openCLIP ViT-H/14 | - | 224 | | 0.57 | 1.00 | 0.62 | 0.33 | 0.25 | 0.50 | 0.35 | 0.14 | 0.45 | 0.49 |
| openCLIP ViT-G/14 | - | 224 | | 0.55 | 1.00 | 0.44 | 0.67 | 0.26 | 0.50 | 0.32 | 0.14 | 0.39 | 0.58 |
| **SD 2.0** | 30 | 512 | # of samples | 76 | 1 | 108 | 3 | 104 | 8 | 112 | 7 | - | - |
| | | | zero-shot | 0.53 | 0.0 | 0.73 | 0.67 | 0.61 | 0.88 | 0.63 | 1.0 | 0.62 | 0.64 |
| | | | discffusion | 0.72 | 1.0 | 0.52 | 1.0 | 0.53 | 0.88 | 0.46 | 0.86 | 0.56 | 0.93 |
| SD 1.5 | 30 | 512 | zero-shot | 0.34 | 0.0 | 0.30 | 0.33 | 0.21 | 0.38 | 0.29 | 0.14 | 0.28 | 0.21 |
| | | | discffusion | 0.22 | 0.0 | 0.10 | 0.0 | 0.32 | 0.50 | 0.21 | 0.29 | 0.21 | 0.20 |
| SD 3-m | 30 | 1024 | zero-shot | 0.39 | 1.0 | 0.33 | 0.67 | 0.25 | 0.38 | 0.23 | 0.29 | 0.30 | 0.58 |
| | | | discffusion | 0.43 | 1.0 | 0.31 | 0.33 | 0.24 | 0.75 | 0.27 | 0.57 | 0.31 | 0.66 |
| CLIP RN50x64 | - | 224 | | 0.13 | 0.22 | 0.27 | 0.47 | 0.38 | 0.27 | 0.23 | 0.19 | 0.25 | 0.29 |
| CLIP ViT-B/32 | - | 224 | | 0.09 | 0.06 | 0.19 | 0.07 | 0.65 | 0.66 | 0.13 | 0.11 | 0.27 | 0.22 |
| CLIP ViT-L/14 | - | 224 | cosine-sim | 0.16 | 0.06 | 0.51 | 0.47 | 0.27 | 0.41 | 0.29 | 0.22 | 0.31 | 0.29 |
| openCLIP ViT-H/14 | - | 224 | | 0.20 | 0.00 | 0.56 | 0.73 | 0.27 | 0.32 | 0.36 | 0.33 | 0.35 | 0.35 |
| openCLIP ViT-G/14 | - | 224 | | 0.28 | 0.17 | 0.56 | 0.60 | 0.33 | 0.36 | 0.30 | 0.31 | 0.37 | 0.36 |
| **SD 3-m** | 30 | 1024 | # of samples | 76 | 18 | 108 | 15 | 104 | 44 | 112 | 36 | - | - |
| | | | zero-shot | 0.68 | 0.89 | 0.66 | 0.93 | 0.72 | 0.86 | 0.80 | 0.92 | 0.72 | 0.90 |
| | | | discffusion | 0.71 | 0.94 | 0.66 | 1.0 | 0.71 | 0.86 | 0.80 | 0.92 | 0.72 | 0.93 |
| SD 1.5 | 30 | 512 | zero-shot | 0.28 | 0.28 | 0.30 | 0.13 | 0.33 | 0.36 | 0.37 | 0.31 | 0.32 | 0.27 |
| | | | discffusion | 0.22 | 0.17 | 0.20 | 0.0 | 0.39 | 0.61 | 0.18 | 0.06 | 0.25 | 0.21 |
| SD 2.0 | 30 | 512 | zero-shot | 0.17 | 0.06 | 0.29 | 0.40 | 0.43 | 0.43 | 0.38 | 0.33 | 0.32 | 0.30 |
| | | | discffusion | 0.51 | 0.33 | 0.27 | 0.27 | 0.48 | 0.50 | 0.29 | 0.31 | 0.39 | 0.35 |

Table E.14: Fine-grained accuracy on SELF-BENCH *Counting*

| Version | Timesteps | Resolution | Method | one | | two | | three | | four | | Macro Average | |
|---|---|---|---|---|---|---|---|---|---|---|---|---|---|
| | | | | Full | Correct | Full | Correct | Full | Correct | Full | Correct | Full | Correct |
| CLIP RN50x64 | - | 224 | | 0.00 | 0.00 | 0.58 | 0.78 | 0.32 | 0.55 | 0.51 | 0.56 | 0.35 | 0.47 |
| CLIP ViT-B/32 | - | 224 | | 0.00 | 0.00 | 0.56 | 0.71 | 0.31 | 0.50 | 0.69 | 0.78 | 0.39 | 0.50 |
| CLIP ViT-L/14 | - | 224 | cosine-sim | 0.00 | 0.00 | 0.47 | 0.71 | 0.43 | 0.55 | 0.58 | 0.56 | 0.37 | 0.45 |
| openCLIP ViT-H/14 | - | 224 | | 0.00 | 0.00 | 0.62 | 0.86 | 0.37 | 0.79 | 0.48 | 1.00 | 0.37 | 0.66 |
| openCLIP ViT-G/14 | - | 224 | | 0.00 | 0.00 | 0.56 | 0.86 | 0.43 | 0.84 | 0.50 | 1.00 | 0.37 | 0.68 |
| **SD 1.5** | 30 | 512 | # of samples | 4 | 1 | 104 | 51 | 108 | 37 | 104 | 9 | - | - |
| | | | zero-shot | 0.50 | 1.0 | 0.74 | 0.78 | 0.49 | 0.68 | 0.72 | 1.0 | 0.61 | 0.86 |
| | | | discffusion | 0.75 | 1.0 | 0.08 | 0.06 | 0.86 | 0.95 | 0.54 | 0.67 | 0.56 | 0.67 |
| SD 2.0 | 30 | 512 | zero-shot | 0.25 | 0.0 | 0.51 | 0.55 | 0.31 | 0.38 | 0.57 | 0.67 | 0.41 | 0.40 |
| | | | discffusion | 0.50 | 0.0 | 0.61 | 0.84 | 0.20 | 0.32 | 0.40 | 0.33 | 0.43 | 0.37 |
| SD 3-m | 30 | 1024 | zero-shot | 0.5 | 0.0 | 0.38 | 0.53 | 0.41 | 0.54 | 0.33 | 0.56 | 0.41 | 0.41 |
| | | | discffusion | 0.50 | 0.0 | 0.38 | 0.51 | 0.44 | 0.54 | 0.30 | 0.56 | 0.41 | 0.40 |
| CLIP RN50x64 | - | 224 | | 0.00 | 0.00 | 0.69 | 0.87 | 0.37 | 0.58 | 0.44 | 0.67 | 0.38 | 0.53 |
| CLIP ViT-B/32 | - | 224 | | 0.00 | 0.00 | 0.55 | 0.64 | 0.23 | 0.27 | 0.75 | 1.00 | 0.38 | 0.48 |
| CLIP ViT-L/14 | - | 224 | cosine-sim | 0.00 | 0.00 | 0.59 | 0.69 | 0.44 | 0.62 | 0.55 | 0.87 | 0.39 | 0.54 |
| openCLIP ViT-H/14 | - | 224 | | 0.00 | 0.00 | 0.74 | 0.93 | 0.44 | 0.96 | 0.51 | 1.00 | 0.40 | 0.70 |
| openCLIP ViT-G/14 | - | 224 | | 0.00 | 0.00 | 0.75 | 0.94 | 0.31 | 0.96 | 0.55 | 1.00 | 0.40 | 0.73 |
| **SD 2.0** | 30 | 512 | # of samples | 4 | 3 | 104 | 70 | 108 | 23 | 104 | 15 | - | - |
| | | | zero-shot | 1.00 | 0.75 | 0.94 | 0.89 | 0.91 | 0.64 | 1.00 | 0.83 | 0.96 | 0.78 |
| | | | discffusion | 0.75 | 0.67 | 0.88 | 1.00 | 0.37 | 0.61 | 0.66 | 0.87 | 0.67 | 0.79 |
| SD 1.5 | 30 | 512 | zero-shot | 0.0 | 0.0 | 0.57 | 0.60 | 0.37 | 0.48 | 0.55 | 0.80 | 0.37 | 0.47 |
| | | | discffusion | 1.00 | 0.67 | 0.05 | 0.03 | 0.88 | 0.91 | 0.38 | 0.60 | 0.58 | 0.55 |
| SD 3-m | 30 | 1024 | zero-shot | 0.5 | 0.67 | 0.48 | 0.5 | 0.43 | 0.61 | 0.44 | 0.8 | 0.46 | 0.65 |
| | | | discffusion | 0.00 | 0.0 | 0.50 | 0.53 | 0.33 | 0.39 | 0.62 | 0.73 | 0.36 | 0.41 |
| CLIP RN50x64 | - | 224 | | 0.00 | 0.00 | 0.73 | 0.82 | 0.50 | 0.52 | 0.75 | 0.78 | 0.50 | 0.53 |
| CLIP ViT-B/32 | - | 224 | | 0.00 | 0.00 | 0.69 | 0.80 | 0.36 | 0.40 | 0.79 | 0.80 | 0.46 | 0.50 |
| CLIP ViT-L/14 | - | 224 | cosine-sim | 0.00 | 0.00 | 0.67 | 0.75 | 0.73 | 0.79 | 0.64 | 0.70 | 0.51 | 0.56 |
| openCLIP ViT-H/14 | - | 224 | | 0.00 | 0.00 | 0.88 | 0.96 | 0.82 | 0.95 | 0.83 | 0.98 | 0.63 | 0.73 |
| openCLIP ViT-G/14 | - | 224 | | 0.00 | 0.00 | 0.89 | 0.98 | 0.79 | 0.94 | 0.85 | 0.97 | 0.63 | 0.72 |
| **SD 3-m** | 30 | 1024 | # of samples | 4 | 3 | 104 | 84 | 108 | 83 | 104 | 60 | - | - |
| | | | zero-shot | 0.75 | 0.67 | 0.89 | 0.92 | 0.78 | 0.86 | 0.88 | 0.98 | 0.82 | 0.86 |
| | | | discffusion | 0.75 | 0.67 | 0.90 | 0.94 | 0.80 | 0.87 | 0.88 | 0.98 | 0.83 | 0.86 |
| SD 1.5 | 30 | 512 | zero-shot | 0.75 | 0.67 | 0.54 | 0.62 | 0.47 | 0.52 | 0.68 | 0.70 | 0.61 | 0.63 |
| | | | discffusion | 1.00 | 1.0 | 0.09 | 0.13 | 0.91 | 0.95 | 0.38 | 0.55 | 0.59 | 0.66 |
| SD 2.0 | 30 | 512 | zero-shot | 0.50 | 0.67 | 0.58 | 0.64 | 0.32 | 0.37 | 0.83 | 0.90 | 0.56 | 0.65 |
| | | | discffusion | 0.75 | 0.67 | 0.66 | 0.79 | 0.38 | 0.43 | 0.63 | 0.77 | 0.60 | 0.67 |

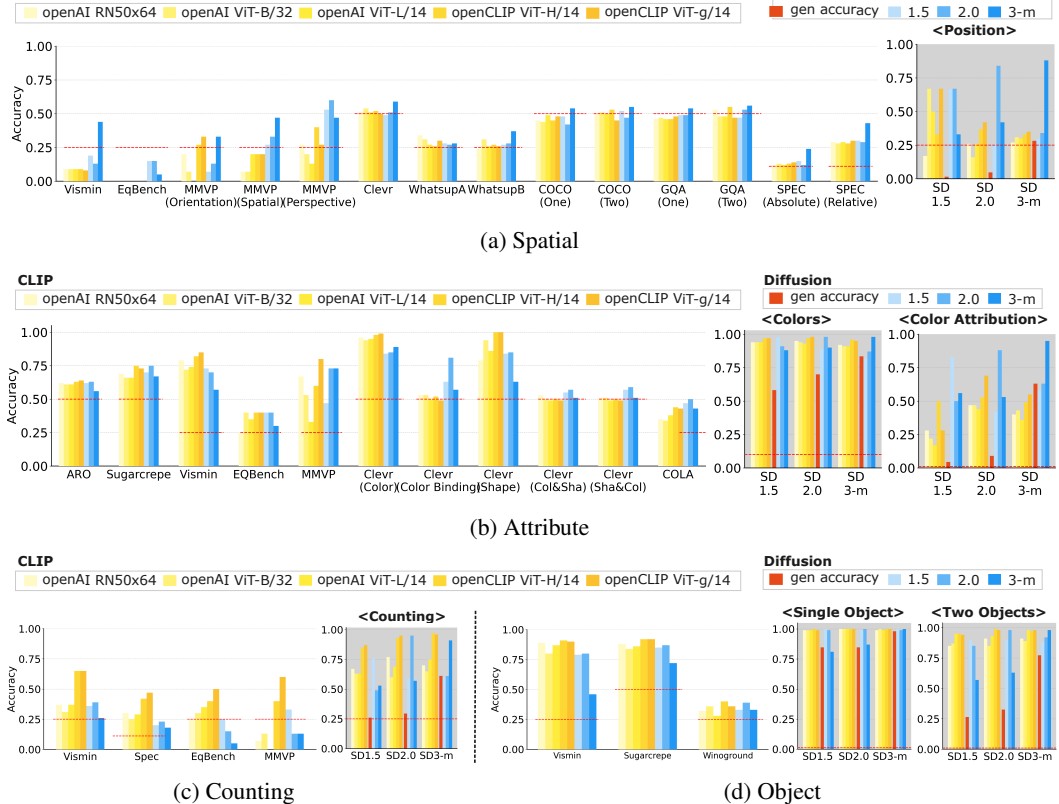

Figure E.4: **Additional results on compositional benchmarks and SELF-BENCH.** The detailed breakdown of our analysis within four categories: (a) *Object*, (b) *Attribute*, (c) *Position*, and (d) *Counting*. Results on a white background correspond to performance across ten existing compositional benchmarks (33 sub-tasks), while those on a gray background represent results on SELF-BENCH. The red dotted horizontal line represents the random chance level.

## E.2 Results on a distilled SD3.5 and FLUX model

Given that timestep reweighting has a strong positive effect on the performance of the Stable Diffusion 3 model, we further investigate whether distilled versions of these models (capable of generating images in as few as 4 steps) behave differently from their corresponding base models. Specifically, we evaluate Stable Diffusion 3 and its distilled counterpart, Stable Diffusion 3.5 Large Turbo, on Self-Bench at a resolution of 512. While the distilled model supports extremely fast generation, we ensure a fair comparison by running both models at 30 inference steps.

This experiment tests whether aggressive distillation, while beneficial for generation quality and efficiency, compromises the discriminative performance of the model—or not. The results suggest that it does: the distilled model underperforms significantly compared to its base version. We present the results in Table E.15. Additionally, FLUX [60] has been introduced with impressive generative quality. We include FLUX in our analysis as an exploratory case, focusing on Position and Counting tasks in real-world datasets—categories where diffusion models typically perform particularly well and poorly, respectively. As shown in Table E.16 and Table E.17, we again observe a significant drop in discriminative performance for the distilled model.

Table E.15: **SD3 Large turbo (distilled) / SD3-m accuracies on Self-Bench tasks**.

| Geneval Version | Color Attr | Position | Counting | Colors | Single | Two |
|---|---|---|---|---|---|---|
| **1.5** | 0.17 / 0.15 | 0.67 / 0.63 | 0.33 / 0.30 | 0.53 / 0.72 | 0.48 / 0.74 | 0.19 / 0.49 |
| **2** | 0.31 / 0.34 | 0.47 / 0.48 | 0.41 / 0.42 | 0.65 / 0.71 | 0.51 / 0.78 | 0.16 / 0.57 |
| **3-m** | 0.49 / 0.72 | 0.40 / 0.68 | 0.43 / 0.57 | 0.83 / 0.84 | 0.73 / 0.88 | 0.43 / 0.75 |

Table E.16: **Performance of FLUX in *Position*.**

| Method | WhatsUP A | WhatsUP B | COCO one | COCO two | GQA one | GQA two | SPEC relative | EQbench | Vismin | MMVP spatial | MMVP orientation | MMVP perspective | CLEVR | SPEC absolute |
|---|---|---|---|---|---|---|---|---|---|---|---|---|---|---|
| SD1.5 | 0.28 | 0.27 | 0.48 | 0.52 | 0.49 | 0.47 | 0.30 | **0.15** | 0.19 | 0.27 | 0.07 | **0.53** | 0.49 | 0.15 |
| SD2.0 | 0.27 | 0.28 | 0.42 | 0.47 | 0.49 | 0.53 | 0.29 | **0.15** | 0.13 | 0.33 | 0.13 | 0.6 | 0.51 | 0.12 |
| SD3-m | 0.28 | **0.37** | **0.54** | **0.55** | **0.54** | **0.56** | **0.43** | 0.05 | **0.44** | **0.47** | **0.33** | 0.47 | **0.59** | **0.24** |
| FLUX | 0.28 | 0.29 | 0.47 | **0.55** | 0.52 | 0.50 | 0.35 | 0.0 | 0.24 | 0.27 | 0.2 | 0.33 | 0.57 | 0.17 |

Table E.17: **Performance of FLUX in *Counting*.**

| Method | Vismin | EQbench | MMVP | SPEC |
|---|---|---|---|---|
| SD1.5 | **0.33** | **0.25** | **0.33** | 0.2 |
| SD2.0 | 0.13 | 0.15 | 0.13 | **0.23** |
| SD3-m | 0.13 | 0.05 | 0.13 | 0.18 |
| FLUX | 0.17 | 0.05 | 0.13 | 0.15 |

## E.3 Style alignment

In this section, we present additional results on attempts to close the domain gap. We perform style alignment experiments using the textual inversion technique. The core idea is to adapt the model's style representation to better fit the target domain. Specifically, for each dataset, we aim to reduce the domain gap for the Stable Diffusion 3 Medium model.

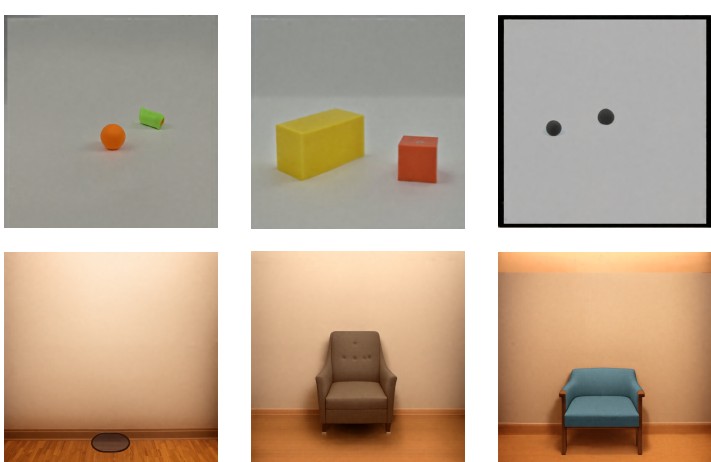

Figure E.5: **Image generations of SD3-m with style alignment.** Top: CLEVR dataset; Bottom: Whatsup-A dataset.

To achieve this, we learn a style token denoted as $\mathcal{S}^*$ via textual inversion. Given a dataset, we train $\mathcal{S}^*$ such that it enables accurate reconstruction of the dataset's images when used in text prompts.

The training prompts follow the structure: "a clear photo in the style of $\mathcal{S}^*$". We performed these experiments on the original dataset size of 512.

At inference time, when evaluating diffusion classifiers, we include the learned style token in the prompt by appending the phrase "in the style of $\mathcal{S}^*$". This style-conditioned prompting is used across several datasets, including the Whatsup-A and CLEVR-ColorBinding benchmarks.

The generations are shown in Fig. E.5. The results of these experiments are summarised in the table below. Overall, we find that style alignment through textual inversion may not be an effective way to mitigate the domain gap.

Table E.18: **Diffusion classifier accuracy before and after style alignment via textual inversion.**

| Dataset | Before Alignment (%) | After Alignment (%) |
|---|---|---|
| WhatsApp-A | 26.5 | 29.3 |
| CLEVR-ColorBinding | 59.1 | 57.1 |

## E.4 Additional qualitative results

We illustrate a qualitative example of SD2.0 and SD3-m generation results from different timesteps on Whatsup-A dataset in Figure E.6.

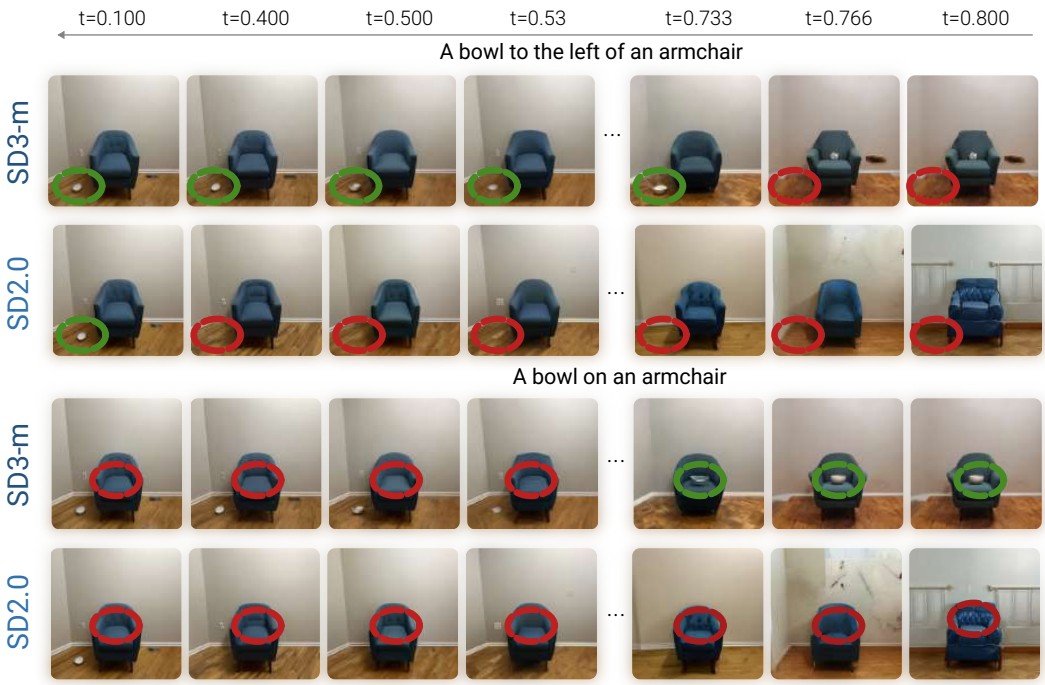

Figure E.6: **Image generation results on Whatsup-A using SD2.0 and SD3-m models.**

## E.5 Upperbounding Self-Bench performance with timestep weighting

**Timestep weights benefit all models cross-domain, but SD3 the most.** We investigate whether timestep weighting can mitigate performance issues, particularly for the SD3-m model; we follow Section 3.1 and report upper-bound performance by fitting timestep weights on all data.

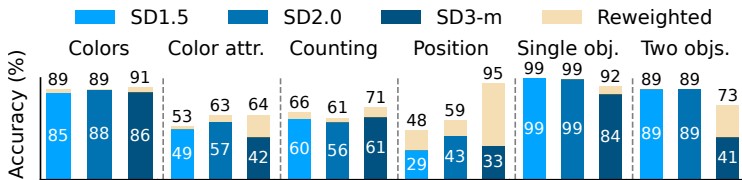

Figure E.7: **SELF-BENCH: Timestep reweighting helps address the cross-domain problem.** We show the performance of each SD model on cross-domain data (e.g., for the SD1.5 model, the bars depict its performance on SD2.0 and SD3-m generations, averaged). All models benefit from timestep reweighting, with SD3-m benefiting the most.

Our upper bound analysis in SELF-BENCH reveals that optimizing timestep weights improves performance across most tasks and models (Figure E.7). The improvements are particularly dramatic for SD3-m: in spatial tasks, accuracy increases from 33% to 95%, and in the two-objects task from 41% to 73%. While SD1.5 and SD2.0 also benefit from reweighting, showing improvements of 1-10% across tasks, the gains are more modest compared to SD3-m. This suggests that SD3-m's lower baseline performance is not due to fundamental model limitations, but rather suboptimal weighting of timestep information.

Contrary to previously argued uniform weighting or decaying weighting schemes, we find that timesteps near the end of the diffusion process (i.e., large $t$) tend to perform better for classification. This is in contrast to previous works [6, 20] that have argued for using uniform or exponentially decaying weights across timesteps.

## E.6 CLIP scores and domain gaps

As an extension of Figure 9 in Section 5.4 of the main paper, Table E.19 shows CLIP embedding distances between real-world datasets and SELF-BENCH generations, and corresponding accuracy gains from timestep weighting. We can see the positive correlation in SD3 but not in SD1 and SD2.

Table E.19: **Timestep Weighting and Domain Gap.** CLIP embedding distances between real-world datasets and SELF-BENCH generations, and corresponding accuracy gains from timestep weighting.

| Dataset | CLIP Distance (SD 1.5) | Δ Accuracy (SD1.5) | CLIP Distance (SD2.0) | Δ Accuracy (SD2.0) | CLIP Distance (SD 3-m) | Δ Accuracy (SD3-m) |
|---|---|---|---|---|---|---|
| COCO QA‡ | 3.494 | 5% | 3.133 | 5% | 3.576 | +4% |
| VQ QA‡ | 3.666 | 5% | 3.584 | 1% | 3.918 | +5% |
| SPEC Count† | 2.646 | 2% | 2.598 | 0% | 3.191 | +5% |
| WhatsUp B‡ | 4.254 | 0% | 4.016 | 0% | 4.047 | +4% |
| WhatsUp A‡ | 4.656 | -1% | 4.344 | 4% | 4.600 | +12% |
| CLEVR Binding* | 5.64 | 9% | 4.348 | 8% | 4.926 | +35% |
| CLEVR Spatial‡ | 5.57 | -3% | 4.63 | 2% | 5.023 | +16% |

‡ Position   † Counting   ⋆ Attribute

## E.7 SigLIP results

We also report SigLIP and SigLIP2 (ViT-SO400M-14 and ViT-L-16-256) as additional discriminative baselines in Table E.20. While SigLIP sometimes exceeds CLIP (e.g., SugarCrepe-Attribute), the conclusion remains the same: classifiers remain best in Position, and CLIP/SigLIP dominate Object/Counting.

Table E.20: **SigLIP baselines**. We report the best CLIP variant, SigLIP, SigLIP2, and the best diffusion classifier per task.

| Benchmark | Best Diffusion | Best CLIP | SigLIP | SigLIP2 |
|---|---|---|---|---|
| Self-Bench (1.5) | 0.88 | 0.79 | 0.80 | 0.81 |
| Self-Bench (2.0) | 0.94 | 0.84 | 0.86 | 0.89 |
| Self-Bench (3-m) | 0.95 | 0.80 | 0.79 | 0.82 |
| COCO QA two | 0.55 | 0.53 | 0.50 | 0.49 |
| VQ QA two | 0.56 | 0.55 | 0.49 | 0.51 |
| SPEC Count | 0.23 | 0.47 | 0.40 | 0.41 |
| WhatsUP A | 0.28 | 0.34 | 0.28 | 0.30 |
| WhatsUP B | 0.37 | 0.31 | 0.28 | 0.28 |
| CLEVR Colors Bind. | 0.81 | 0.53 | 0.50 | 0.51 |
| CLEVR Spatial | 0.59 | 0.54 | 0.50 | 0.50 |
| SugarCrepe attrib. | 0.75 | 0.75 | 0.78 | 0.77 |
| SugarCrepe object | 0.87 | 0.92 | 0.91 | 0.92 |

