# OpenReview forum: "Diffusion Classifiers Understand Compositionality, but Conditions Apply"
_NeurIPS.cc/2025/Datasets_and_Benchmarks_Track — NeurIPS 2025 Datasets and Benchmarks Track poster_

### Official Review · Reviewer_HGDo · 2025-06-02

**Rating:** 5
**Confidence:** 4

**Summary:**

This paper proposes a comprehensive study of the discriminative capabilities of diffusion classifiers on a wide range of compositional tasks. Moreover, this paper proposes the SELF-BENCH benchmark to evaluate the performance of the diffusion model.

**Dataset Code Accessibility:**

No

**Dataset Code Comments:**

As a benchmark, this paper has sufficient detail to support reproducibility. The code and data are readily accessible and reproducible, and made available in an executable format.

**Ethical Considerations:**

No, there are no or only very minor ethics concerns

**Final Justification:**

The author's response effectively addressed my concerns, and I provided a positive evaluation and recommended acceptance of this paper.

**Limitations Weaknesses:**

1. The timestep weighting is very important in the diffusion model. Some related works [1-3] also find this conclusion. Therefore, I suggest the author have a short discussion in the paper.

[1] A closer look at time steps is worthy of triple speed-up for diffusion model training. CVPR-2025

[2] Beta-tuned timestep diffusion model. ECCV-2024

[3] Adaptive Non-Uniform Timestep Sampling for Diffusion Model Training. arXiv-2024

2. In Eq.1 $\omega_{t}$ should be loss function weights. The timestep weight should the outside of Eq. 1 (e.g., t~Unfiorm).

**Strengths Contributions:**

1. The finding that timestep weighting is especially effective in scenarios with large domain gaps is interesting. Moreover, the author conducted in-depth discussions and analyses, which are very valuable for the application of diffusion models.

2. The experiments in this paper are extensive and include various mainstream diffusion models, which makes the experiments in this paper very convincing.

---

> ### Author Rebuttal · Authors · 2025-07-31
>
> We greatly appreciate that the reviewer acknowledges the effectiveness of timestep weighting, as well as our extensive and convincing experiments across various diffusion model series. The reviewer’s suggestions are addressed in detail below.
>
> ### **1. Timestep weighting related work**
>
> We thank the reviewer for providing many valuable references. While our work focuses solely on inference, we will include [1–3] in post as they provide useful context on how non-uniform timestep strategies affect both training and generation. In particular, [2] is especially relevant as it demonstrates the benefit of replacing uniform timestep sampling with a beta-based strategy during inference to better match the forward diffusion process. Meanwhile, [1,3] highlight how non-uniform sampling influences the training dynamics, which, although outside our direct scope, help frame the broader discussion. All support the conclusion that different timesteps carry different information, which aligns with our Finding 3. We will include these references in the updated manuscript.
>
> [1] A closer look at time steps is worthy of triple speed-up for diffusion model training. (CVPR 2025)
> [2] Beta-tuned timestep diffusion model. (ECCV 2024)
> [3] Adaptive Non-Uniform Timestep Sampling for Diffusion Model Training. (ArXiv 2024)
>
> &nbsp;&nbsp;
>
> ### **2. Eq 1. interpretation**
>
> Thank you for pointing this out. We used the unified loss formulation from [1], which is also used in SD3 [2]. Under such a formulation we write
>
> $$\mathcal{L}(z, c) = \mathbb{E}_{t \sim U(0,1), \epsilon} [ w_t \|\| \epsilon - \hat \epsilon ({z}_t, t, c) \|\|^2],$$
>
> where $\hat \epsilon := \epsilon_\Theta$ in standard notation, due to openreview's math rendering we made a substitution.
>
> We understand that your comment refers to specifying the sampling distribution of t. While we originally described it in general terms in the paper, we will now specify it explicitly as being drawn from a uniform distribution. Please let us know if we misunderstood.
>
>
> [1] Understanding diffusion objectives as the elbo with simple data augmentation. (NeurIPS 2023)
> [2] Scaling rectified flow transformers for high-resolution image synthesis. (ICML 2024)

---

> > ### Comment · Reviewer_HGDo · 2025-08-01
> >
> > Thank the author for their reply. I will keep my positive rating.

---

> > > ### Author Response · Authors · 2025-08-01
> > >
> > > Thank you for the positive rating! Let us know if you have any other concerns or would like to discuss anything further.

---

### Official Review · Reviewer_o8Cg · 2025-06-19

**Rating:** 4
**Confidence:** 3

**Summary:**

The paper focuses on analyzing diffusion models' capability to understand compositional attributes. It demonstrates that diffusion-based generative models exhibit strong discriminative performance within their own generated image domain but perform poorly outside this domain. The authors substantiate this claim by introducing Self-Bench. To bridge this gap, they propose timestep weighting, which enhances the generalization ability of diffusion models across diverse domains.

**Dataset Code Accessibility:**

Yes

**Ethical Considerations:**

No, there are no or only very minor ethics concerns

**Final Justification:**

The authors have supplemented their analysis with experiments on timestep sampling and shift, along with additional model comparisons. These results partially address my concerns. I appreciate the authors' efforts. Considering the previously highlighted strengths, I have raised my final score to 4.

**Limitations Weaknesses:**

1. The dataset proposed in the paper has relatively limited generalizability. It is constructed based on generation results from different models using GenEval prompts, which appears to impose significant constraints on its practical applicability.

2. Although this paper highlights the unique performance characteristics of SD3-m, it lacks in-depth analysis. For instance, according to the authors' conclusions, timestep reweighting significantly enhances the discriminative performance of SD3-m but has a smaller impact on other models. But why? Regarding the timestep settings, to my knowledge, SD3 employs lognorm timestep sampling during training and applies a timestep shift for high-resolution images. It is expected to have a more fundamental explanation of the timestep issue in the paper, rather than attributing it solely to specific model-related factors.

3. I am confused about the authors' claim in A.3.2 stating that "it is a consensus that CFG does not impact classifier performance." If this were truly the case, then why, in A.6, do they exclude FLUX and SDXL-Turbo on the grounds that "it is challenging to ascertain their performance under classifier-free guidance (CFG-free) settings"?

4. I believe the state-of-the-art FLUX model should at least be included in the evaluation. We could identify commonalities in this issue from more similar architectures (from U-Net to mmDiT) and training methods (from DDPM to flow matching), rather than simplistically attributing this property strictly to "SD3-Medium."

**Strengths Contributions:**

1. In my view, the research direction of this paper can be summarized as: how to leverage generative models for visual understanding of composition. This task itself is valuable. As understanding and generative models increasingly converge, how generative models can facilitate understanding remains a highly worthwhile problem to explore.

2. The paper identifies a critical issue: there exists a significant performance gap between in-distribution and out-of-distribution understanding. To substantiate this, the authors establish Self-Bench—a meaningful finding.

3. This work primarily investigates three distinct models from the Stable Diffusion series, highlighting the unique performance characteristics of the SD3-m model—another intriguing observation.

---

> ### Author Rebuttal · Authors · 2025-07-31
>
> We sincerely thank the reviewer for acknowledging the value of our task and for recognizing the domain gap findings revealed through Self-Bench to be worthwhile. We have addressed the reviewer’s concerns below.
>
> ### **1. The dataset proposed in the paper has relatively limited generalizability**
>
> Our Self-Bench dataset was designed with a specific purpose: to serve as a diagnostic compositional benchmark across six distinct categories and to isolate the domain effects.
> - While the dataset is built using generations from diffusion models prompted via GenEval, its intent is diagnostic, targeting both discriminative and generative model evaluation under controlled variations.
> - It is not meant to replace the existing real-world compositional benchmarks, but rather to isolate domain-specific effects in a systematic way.
> - At the same time, we believe that building off of a well-studied benchmark such as GenEval is more reliable than building own dataset from scratch.
> - Additionally, the Self-Bench framework is extensible: new categories, model families, or prompt sources can be added with minimal effort.
>
> &nbsp;&nbsp;
>
> ### **2. Timestep weighting in SD3**
>
> We appreciate this point.
>
> - First, we include ablations on the timestep reweighting strategy to provide more empirical evidence (see table below). Our results show that the log-normal timestep sampling used in SD3’s training does not solely account for its improved performance. This observation aligns with previous work [1], which employed a heuristic weighting scheme, $w_t = \exp(-7t),$ different from the original training scheme, $w_t = \mathrm{SNR}(t),$ for SD 1.4 and Imagen and found it to perform better than using the same training weighting scheme. (*However, as illustrated in Table A.7 in the Appendix, this heuristic weighting $w_t = \exp(-7t),$ can also underperform compared to uniform sampling for SD 1.5, which is generally considered to be a fine-tuned version of SD 1.4.*) This suggests that other mechanisms are contributing to the performance gains.
> - As the reviewer mentioned regarding the timestep shift, we have conducted ablations with a 3.0 timestep shift (used for higher resolution). We hypothesize that the performance gains may arise from how SD3 handles distribution shifts toward higher noise levels (i.e., early timesteps). However, we have not observed big improvements in discriminative performance.
>
> We believe this direction warrants further exploration, and we will mention it as a future avenue for understanding such model-specific effects. If the reviewer believes there is a more fundamental way to address these questions, we would be happy to consider further analyses.
>
> [1] Text-to-image diffusion models are zero shot classifiers. (NeurIPS 2023)
>
> The table below reports text-to-image retrieval accuracy (higher is better). For Self-Bench, we report the aggregated average for all categories.
>
> | Benchmarks \ Models | SD3-m default (uniform sampling & No shift) | SD3-m (lognorm timestep sampling) | SD3-m (3.0 timestep shift) |
> | :---- | :---- | :---- | :---- |
> | Self-Bench (1.5) | **0.61** | 0.06 | **0.61** |
> | Self-Bench (2.0) | 0.65 |  0.12 | **0.66** |
> | Self-Bench (3-m)  | **0.95** | 0.19 | **0.95** |
> | COCO QA two | **0.59** | 0.54 | 0.55 |
> | VQ QA two | 0.53 | 0.42 | **0.56** |
> | SPEC Count | **0.20** | 0.11 | 0.18 |
> | WhatsUP A | 0.30 | **0.34** | 0.28 |
> | WhatUP B | **0.46**  | 0.35 | 0.38 |
> | CLEVR colors Binding | **0.63** | 0.55 | 0.57 |
> | CLEVR Spatial |**0.63** | 0.55 | 0.59 |
> | Sugarcrepe attribution | 0.70 | 0.55 | **0.68** |
> | Sugarcrepe object | **0.76** | 0.50 | 0.72 |
>
> &nbsp;&nbsp;
>
> ### **3. On CFG and diffusion classifiers**
>
> We apologise for the confusion. Our statement that “CFG does not impact classifier performance” was intended to mean that CFG does not improve classification performance. This is established in previous works [1,2]. We will clarify this in the manuscript.
>
> [1] Your diffusion model is secretly a zero-shot classifier." Proceedings of the IEEE/CVF International Conference on Computer Vision (ICCV 2023)
> [2] Text-to-image diffusion models are zero shot classifiers. (NeurIPS 2023)
>
> &nbsp;&nbsp;
>
> ### **4. Inclusion of FLUX and transformer-based models**
>
> We appreciate the suggestion and have carefully considered a wide range of models as potential baselines. Below we summarise the models we examined and our rationales:
> 1. *Stable Diffusion 3.5 Medium [1]* – This model is architecturally similar to SD3-Medium, with the main difference being the use of MMDiT-X instead of MMDiT, (new QK normalization and dual attention layers are reportedly applied to SD3.5-Large, though it is unclear whether these changes are present in SD3.5-Medium). **We have now added this baseline and show that its performance trends are very similar to SD3**.
> 2. *Muse [2] / Amuse [3]* – These are masked image models, not diffusion-based, and therefore not directly comparable within our current setup.
> 3. *FLUX [4] & SD3.5 Large Turbo [5]* – Both use the mmDiT architecture with flow matching loss, and are distilled models. Their training setup makes it difficult to isolate behaviour in a consistent, non-CFG setting, which is a critical factor, as noted in our response to Point 3 above. Moreover, according to previous work [6], FLUX does not even outperform SD3-medium on the compositional generative benchmark (Geneval). Due to this ambiguity and lack of comparability, we excluded them from the main results.
>     * Nevertheless, we have included its results (see Appendix Section A.8 for reference), in table below. Due to the models being distilled versions with CFG, they predictably show inferior performance. *Additionally, we noticed a typo in Table A.5 in Appendix: the labels for SD3.5 and SD3-m were reversed. We apologise for the oversight and will correct this in the final version.*
> 4. *Parti [7]* – This is an autoregressive model, incompatible with our evaluation setup focused on diffusion models.
> 5. *Auraflow [8]* – Unfortunately, no public training scripts or documentation are available, making reproduction and inclusion infeasible.
> 6. *HiDream-I1 [6]* – The HiDream-I1-full model has 17B parameters, which is prohibitively large for our experiments (for comparison, Flux-dev.1 has 12B and already takes approximately two times longer than SD3-m, which has 8B). The other variants, HiDream-I1-dev and HiDream-I1-fast, are distilled versions and are therefore excluded for the reasons mentioned above.
>
> We are happy to consider additional baselines. If the reviewer has specific suggestions or models in mind that fit within the scope of our evaluation framework, we would greatly appreciate them. We also want to note that we have included REPA (which is not a foundation model) in response to reviewer jvhk’s comment that *3. REPA is worth discussing*. This supports our view that only strong foundation models are necessary as meaningful baselines for comparison.
>
> [1] Stable Diffusion 3.5 Medium model (available on Hugging Face; external link omitted)
> [2] Muse: Text-to-image generation via masked generative transformers (Arxiv 2023)
> [3] aMUSEd: An Open MUSE Reproduction (Arxiv 2024)
> [4] Flux model (available on Hugging Face; external link omitted)
> [5] SD3.5 Large Turbo (available on Hugging Face; external link omitted)
> [6] HiDream-I1: A High-Efficient Image Generative Foundation Model with Sparse Diffusion Transformer (Arxiv 2025)
> [7] Scaling Autoregressive Models for Content-Rich Text-to-Image Generation (TMLR 2022)
> [8] Auraflow model (available on Hugging Face; external link omitted)
>
> The table below reports text-to-image retrieval accuracy (higher is better). For Self-Bench, we report the aggregated average for all categories.
> | Benchmarks \ Models | sd 1.5 | SD 2.0 | SD 3-m | SD 3.5-medium | FLUX-dev.1 |
> | :---- | :---- | :---- | :---- | :---- | :---- |
> | Self-Bench (1.5)  | **0.86** | 0.74 | 0.61 | 0.70 | 0.31 |
> | Self-Bench (2.0) | 0.67 | **0.94**  | 0.65 | 0.63 | 0.38 |
> | Self-Bench (3-m) | 0.70 | 0.65 | **0.95** | 0.89 | 0.86 |
> | Self-Bench (3.5-m)| 0.69 | 0.77 | 0.84 | **0.89**  | 0.53 (preliminary) |
> | COCO QA two | 0.42 | 0.43 | **0.59** | 0.55 | 0.55 |
> | VQ QA two | 0.44 | 0.46 | **0.53** | 0.52 | 0.50 |
> | WhatsUP A | 0.28 | 0.27 | **0.30** | **0.30** | 0.28 |
> | WhatUP B | 0.32 | 0.29 | **0.46**  | 0.40 | 0.29 |
> | CLEVR colors Binding | 0.70 | **0.84** | 0.63 | 0.61 | 0.63 |
> | CLEVR Spatial | 0.51 | 0.48 | **0.63** | 0.62 | 0.57 |
> | sugarcrepe attribution | 0.71 | **0.74** | 0.70 | 0.73 | 0.62 |
> | sugarcrepe object | 0.85 | **0.86** | 0.76 | 0.76 | 0.59 |
>
> **preliminary = The evaluation is still running, the result will be included in the next update.*

---

> > ### Comment · Reviewer_o8Cg · 2025-08-04
> >
> > I appreciate the authors' experimental work, which has partially addressed my concerns. Accordingly, I have raised my score from 3 to 4.

---

### Official Review · Reviewer_jvhk · 2025-07-01

**Rating:** 4
**Confidence:** 4

**Summary:**

This paper studied using diffusion-based T2I model as discriminative model. The authors propose self-bench, a new benchmark to evaluate the performance of different diffusion models as classifier, which uses images generated by diffusion model to mitigate the effects of out-of-domain data. The results show that diffusion model can serve as strong discriminative models, bette than CLIP baseline, after applying timestep reweighting.

**Dataset Code Accessibility:**

Yes

**Ethical Considerations:**

No, there are no or only very minor ethics concerns

**Final Justification:**

I recommend accept this paper.

**Limitations Weaknesses:**

- The only baseline is CLIP, some stronger models e.g. SigLIP was not tested.
-  It seems that for some sub-domain e.g. position/counting, the current model fails, which may need more explanations.
-  Recent papers e.g. REPA tries to align diffusion feature with CLIP/Dino feature, which are worth being included into this benchmark.

**Strengths Contributions:**

- This paper explores using diffusion model as classifier in a systematic way: (1) removing the effect of out-of-domain data (2) conduct large scale evaluation across different models. The results offer some insight into how to use diffusion model as classifier: use sd3 with reweighting.
- The collected benchmark / dataset is clean, well-defined and easy to use.

---

> ### Author Rebuttal · Authors · 2025-07-31
>
> We are grateful for the reviewer’s positive feedback, noting that our work is systematic and that Self-Bench is clean, well-defined, and easy to use. The reviewer’s concerns and questions are addressed below.
>
> ### **1. SigLIP was not tested**
>
> Thank you for the suggestion. In the main paper, we focused on CLIP because it is used in SD models' text encoders and is commonly used with comparatively good performance (especially newer OpenCLIP models we've used as well).
>
> We have now added SigLIP and SigLIP2 as baselines (ViT-SO400M-14 SigLIP and ViT-L-16-256 SigLIP2 variants, both trained on WebLI). While the inclusion of SigLIP does not alter the main claims/findings of our paper, we observe that its performance in certain categories is superior to CLIP, which we highlight in the following results.
>
> The table below reports text-to-image retrieval accuracy (higher is better). For Self-Bench; we report the aggregated average for all categories.
>
> | Benchmarks \ Models | Best Diffusion Classifier | Best CLIP | SigLIP | SigLIP2 |
> | :---- | :---- | :---- | :---- | :---- |
> | Self-Bench (1.5) |**0.88** | 0.79 | 0.80| 0.81|
> | Self-Bench (2.0) | **0.94** | 0.84 | 0.86 | 0.89 |
> | Self-Bench (3-m)  | **0.95** | 0.80 | 0.79 | 0.82 |
> | COCO QA two | **0.55** | 0.53 | 0.50 | 0.49 |
> | VQ QA two | **0.56** | 0.55 | 0.49 | 0.51 |
> | SPEC Count | 0.23 | **0.47** | 0.40 | 0.41 |
> | WhatsUP A | 0.28 | **0.34** | 0.28 | 0.30 |
> | WhatUP B | **0.37** | 0.31 | 0.28 | 0.28 |
> | CLEVR colors Binding | **0.81** | 0.53 | 0.50 | 0.51 |
> | CLEVR Spatial | **0.59** | 0.54 | 0.50 | 0.50 |
> | sugarcrepe attribution | 0.75 | 0.75 | **0.78** | 0.77 |
> | sugarcrepe object | 0.87 | **0.92** | 0.91 | **0.92** |
>
> &nbsp;&nbsp;
>
> ### **2. For position/counting, the current model fails, which may need more explanations**
>
> Thanks for raising this question. Although the random chance for Position is similar to that of Object, the diffusion classifier’s accuracy is notably lower. For Counting, it performs even below random chance. This suggests that Position and Counting can be challenging compared to the other subsets. (Figure A.13 in the Appendix provides a helpful reference.)
>
>
> - Discriminative abilities $p(y \mid x)$ are generally tied to generative capabilities $p(x \mid y)$, through the relationship $p(y \mid x) \propto p(y) p(x \mid y)$.  In Self-Bench, we observe that diffusion models struggle with generation in both the Position and Counting tasks. This aligns with prior work [1,2], which also reports poor generative performance $p(x \mid y)$ on these categories, indicating that they remain challenging even for state-of-the-art models. Such deficiencies in generation can naturally lead to degraded discriminative performance.
> - We emphasize that these challenges are not unique to diffusion models. Vision-language models, including CLIP-based models, are also known to struggle with such tasks [3,4]. Since diffusion models rely on CLIP text encoders, some of these issues may originate from there [5,6]. However, it remains underexplored whether conditional text encoder models are the sole cause of these problems.
> - We have empirically observed that Position errors often stem from confusion between concepts like “left” and “right,” which depend on camera perspectives, something not explicitly captured in the training datasets. Similarly, “above” and “below” are particularly challenging; for instance, generating “desk below elephant” is more difficult than “elephant below desk.” On the other hand, prompts like “elephant on the desk” are easier to generate, indicating that the dataset distribution plays a significant role. We believe these issues come from a relatively infrequent occurance of such concepts in the training data.
> - For Counting, diffusion models generally struggle to generate multiple objects in a single scene, likely due to skewed distributions of counting terms (e.g., “one” appears far more often than “ten”). We provide fine-grained position and counting accuracy in the Appendix (see Tables A.21 and A.22). We will include this discussion in the final version.
> - Based on all of the above, we do not expect outstanding discriminative abilities since generative abilities remain limited in these cases.
>
> [1] Spatial Transport Optimization by Repositioning Attention Map for Training-Free Text-to-Image Synthesis (CVPR 2025)
> [2] Make It Count: Text-to-Image Generation with an Accurate Number of Objects (CVPR 2025)
> [3] Teaching CLIP to Count to Ten (ICCV 2023)
> [4] Thinking in Space: How Multimodal Large Language Models See, Remember, and Recall Spaces (CVPR 2025)
> [5] Mass-Producing Failures of Multimodal Models with Language Models (Arxiv 2024)
> [6] Understanding and Mitigating Compositional Issues in Text-to-Image Generative Models (Arxiv 2024)
>
> &nbsp;&nbsp;
>
> ### **3. REPA is worth discussing**
>
> We appreciate the reviewer’s suggestion to include additional discussion on REPA. On a high level, we believe this highlights the connection between generative and discriminative modelling, and the underlying features being learnt in both cases.
>
> Our initial evaluation focused on foundation pre-trained models. However, to give a broader understanding of how well generative diffusion models can be repurposed for classification, we include REPA as an additional baseline.
>
> - We trained REPA from scratch (as the pre-trained weights are not available) using DINOv2 and used MSE with the v-prediction loss as the discriminative objective for diffusion classifiers, as the official pretrained weights are not publicly available. As REPA is trained exclusively on the COCO dataset, we compared its results on both COCO-related and non-COCO datasets, alongside a Stable Diffusion model fine-tuned with LoRA on the same COCO training dataset (Please refer to our answer to reviewer RGqz on *2-2. Ablation studies regarding model architecture, learning process, or training data?* for detailed experimental settings).
> - We observed that REPA’s performance is at near-random levels, indicating that it is not a competitive baseline compared to foundation models such as Stable Diffusion. *Since our implementation does not use the original pretrained weights reported by the authors, we will include a more detailed discussion of REPA in the related works section of our final camera-ready version to avoid potential misinterpretations.*
>
> The table reports text-to-image retrieval accuracy (higher is better):
> | COCO related benchmarks \ Models | random chance | SD1.5 | SD2.0  | SD3-m  | REPA |
> | :----| :---- | :---- | :---- | :---- | :---- |
> | COCO QA one |0.5 |  0.45  | 0.42| **0.58**  |0.52|
> | COCO QA two |0.5 |  0.53  |0.50  |  **0.62** |0.55|
> | COCO order |0.2 | 0.33 |  **0.35** | 0.27 |  0.14|
>
> | non-COCO related benchmarks \ Models | random chance | SD1.5 | SD2.0  | SD3-m  | REPA |
> | :----| :---- | :---- | :---- | :---- | :---- |
> | Clevr spatial |0.5 | 0.49 | 0.50 | **0.66** |0.51|
> | Clevr binding color |0.5 | 0.64 | **0.84** |0.72| 0.51|
> | Whatsup A |0.25 | 0.26 |0.29 | **0.32** | 0.25|
> | Whatsup B |0.25 | 0.31 | 0.27 |  **0.46**| 0.27|
> | Sugarcrepe object |0.5 |  0.85 |**0.87**  |0.82 | 0.48 |
> | Sugarcrepe attribution |0.5 |  0.72 |  **0.78** | 0.77 | 0.53|

---

### Official Review · Reviewer_RGqz · 2025-07-03

**Rating:** 5
**Confidence:** 3

**Summary:**

This paper inverstigates the discriminative capablities of diffusion-based generative models in handling compositional tasks, mainly including the SD family (SD 1.5, 2.0 and 3-m). The authors present an extensive empirical study across 10 compositional benchmarks (33 tasks) and introduce a novel diagnostic dataset, SELF-BENCH, which contains images generated by diffusion models themselves. Specfically, the benchmark is categorized into four broad task groups, Object, Attribute, Position, and Counting, and the evaluation consists of in-domain and cross-domain settings.

The work tests three hypotheses:
- Diffusion classifiers outperform CLIP in compositional reasoning.
- Generative capabilities imply discriminative understanding (within the same domain, where the test samples are generated by the same model).
- Timestep weighting can improve performance, especially in cross-domain settings.

Key findings include that diffusion classifiers:
- Excel in spatial reasoning, perform on par with "CLIP" in attribute tasks, but struggle with object recognition and counting.
- Perform well when tested in-domain, with a strong drop in accuracy on cross-domain tasks.
- Benefit significantly from task-specific timestep weighting, especially the newer SD3-m model.

**Dataset Code Accessibility:**

Yes

**Dataset Code Comments:**

The authors have publicly released the dataset and code.

**Ethical Considerations:**

No, there are no or only very minor ethics concerns

**Final Justification:**

The authors have addressed my concerns with comprehensive analysis and discussion. The proposed benchmark and investigation will help advance this research area and I recommend accepting this paper.

**Limitations Weaknesses:**

1. The results and discussion of this work are mostly empirical. It would be more interesting to include deeper analysis on why generative capabilities don’t always align with discriminative performance.
2. The ablation studies on SD3-m model is a bit shallow at this point, concerning only resolution and different timesteps. Is it possible to conduct ablation studies regarding model architecture, learning process, or training data? Since its behaviour is significantly different from SD 1.5 and 2.0.
3. What about the behavior of SDXL, which is not included in the comparisons?

A minor issue:
In Line 280, "they perform better on Correct (yellow bars) than on Full (blue bars) samples". The colors used in the text do not seem to match those presented in Figure 5.

**Strengths Contributions:**

1. The study addresses an interesting and timely question in vision-language modeling: whether generative performance translates to discriminative understanding, especially for compositional tasks.
2. The authors conduct extensive empirical assessments of diffusion classifiers in compositional settings, using 10 benchmarks and 33 tasks.
3. The proposed Self-Bench benchmark isolates the domain effect and compare the performance across in-domain and out-of-domain scenarios, providing an approach to verify "if a diffusion model can generate images of certain type, it can also discriminate them."
4. The hypotheses and findings are clearly stated. Additionally, this paper provides more valuable insights, such as: (1) The best generative model (SD3-m) does not always yield the best discriminative results. (2) Timestep sensitivity plays a crucial role in classification performance.
5. The paper is well-structured, with clear hypotheses, systematic experiments, and direct conclusions.

---

> ### Author Rebuttal · Authors · 2025-07-31
>
> We appreciate the reviewer acknowledging that our work is interesting, has extensive assessments, clear hypotheses and findings, is well-structured, and proposes Self-Bench for studying/isolating the domain effect. Below, we address the reviewer’s concerns and suggestions.
>
> ### **1. Why generative capabilities don’t always align with discriminative performance?**
>
> We believe that the misalignment between generative and discriminative performance often arises from the model’s internal domain-specific biases and spurious correlations it picks up during training. We will discuss this more in the final version.
>
>
> - For instance, suppose the model frequently sees/generates “small objects” against a blue background. Generating such small objects within a blue background will be easy for the model.
> - However, during classification, if we show the model a *large* object with a blue background, it may still assign a high probability to the “small object” class rather than the “large object,” because it has strongly associated “small object” with the blue background in its generative space.
> - In other words, the model’s $p(y \mid x)$ (discriminative prediction) may appear accurate for cases that *align with its generative bias* but fail when the context changes. This demonstrates that the model’s generative capability (ability to produce realistic samples) does not necessarily imply that it learns disentangled or robust discriminative features.
>
> ---
>
>
> As such, we hypothesize that SD3-m may focus on a more narrow, high-quality generative domain (sharp spikes in a distribution), but may fail to generate diverse samples; this is akin to the example given above where a model generates the target query with a domain-specific bias. Compared to it, SD2.0, despite lower visual fidelity, may cover a broader range of variations (higher variance), leading to better discriminative performance.
>
> To support the hypothesis that SD3-m's domain has less diverse than SD2.0:
>
> - We run an experiment on the diversity of generated samples using Self-Bench. For each text prompt, there are 4 associated images from both models.
> - We extract image embeddings using CLIP-B/32 and compare:
>   - (1) average cosine similarity between embeddings (measures how visually similar the samples are to each other for the same prompt), and
>   - (2) average variance across embedding dimensions (measures how much the features vary for the same prompt)
>
> | Metric per prompt (n=4 images)         | SD-2.0             | SD-3 m             |
> |----------------------------------------|--------------------|--------------------|
> | **Mean cosine similarity ↓**           | **0.845 ± 0.062**  | 0.895 ± 0.051      |
> | **Mean embedding-dim variance ↑ (×1e-3)** | **0.227 ± 0.190**  | 0.154 ± 0.250      |
>
> These results support the hypothesis: SD2.0 is more diverse both in terms of cross-sample similarity and variance measures.
>
> We note that this remains a *hypothesis* rather than a complete explanation; future work is needed to characterise the connection more rigorously. At the same time, recent findings appear to support our observation, e.g., [1] shows that SD2.0 (and SD1.5) tend to generate more diverse samples than SD3-m.
>
> [1] Beyond Overcorrection: Evaluating Diversity in T2I Models with DIVBENCH, (Arxiv 2025)
>
>
> &nbsp;&nbsp;
>
> ### **3. Behavior of SDXL**
>
> We appreciate the reviewer’s suggestion to include SDXL as a baseline.
>
> - We have now incorporated SDXL and reported its performance on both Self-Bench and real-world benchmarks - *please refer to the table in the section below*. While SDXL is known to have better generative performance than SD1.5, we observe that its accuracy on compositional tasks remains limited. _We also want to highlight that we have included SD3.5-m as baseline in the answer to reviewer o8Cg (*4. Inclusion of FLUX and transformer-based models*)._
>
> &nbsp;&nbsp;
>
> ### **2. Ablation studies regarding model architecture, learning process, or training data?**
>
> Conducting detailed ablations on pretrained foundation models is challenging due to (1) the lack of publicly available information about their training datasets and (2) the difficulty of modifying their architectures or loss functions using pretrained weights.
>
> - Still, to give insight into this, we performed two additional experiments, as much as is feasible in the current setting: (1) text encoder alignment to create more comparable architectures, and (2) LoRA fine-tuning on COCO to isolate the effects of different training datasets.
>
> ---
> 2-1) _Text encoder alignment_. Across SD1.5, SD2.0, SDXL, and SD3-m, the models employ different conditional text encoders. Notably, SDXL uses two encoders, while SD3-m employs three. As we cannot modify the internal architecture of these diffusion models, we "aligned" the text encoders by zeroing out the embeddings from the additional text encoders in SDXL and SD3-m to match the single-encoder setup of SD1.5 (CLIP ViT-L/14). **This modification increased SDXL’s accuracy on Self-Bench but decreased its performance on real-world benchmarks. Interestingly, we did not observe the same trends in SD3-m**, which we believe could be due to its transformer-based architecture, and its distinct training objectives compared to SDXL. In contrast, SDXL shares a similar training setup and U-Net architecture with SD1.5.
>
> Unfortunately, this could not be applied to SD2.0, which uses OpenCLIP ViT/H, a different encoder not shared by the other models. As a result, we evaluated all models using Self-Bench (SD2.0), which serves as a cross-domain benchmark for all models.
>
> The table below reports text-to-image retrieval accuracy (higher is better). For Self-Bench, we report the aggregated average for all categories.
> | Benchmarks \ Models | SD1.5 default (CLIP ViT-L/14) | SDXL default (ViT-L/14,  ViT/G) | SD XL* (ViT-L/14, dropping ViT/G) | SD3-m default (ViT-L/14, ViT/G & T5) | SD3-m* (ViT-L/14, dropping ViT/G & dropping T5) |SD3-m (ViT-L/14 &  ViT/G, dropping T5)
> | :---- | :---- | :---- | :---- | :---- | :---- |:---- |
> | Self-Bench (SD2.0) | **0.67** | 0.40 | 0.45 |0.65 | 0.52  | 0.56 |
> | COCO QA two | 0.42 | 0.53 | 0.50 | **0.59** | 0.56 | 0.51
> | VQ QA two | 0.44 | 0.47 | 0.49 | **0.53** | 0.49 | 0.49
> | WhatsUP A | 0.28 | 0.25 | 0.27 | 0.30 | 0.29 | **0.31**
> | WhatUP B | **0.32** | 0.25 | 0.22 | 0.46  | 0.31 | 0.33
> | CLEVR Colors Binding | **0.70** | 0.52 | 0.52 | 0.63 | 0.60 | 0.64
> | CLEVR Spatial | 0.51 | 0.51 | 0.52 | **0.63** | 0.61 | 0.59
> | Sugarcrepe attribution | **0.71** | 0.70 | 0.64 | 0.70 | 0.68 | 0.68
> | Sugarcrepe object | **0.85** | 0.67 | 0.66 | 0.76 | 0.74 |0.75
>
> Caveat: `*` indicates a modified version not officially released by the original authors. Technically, these configurations may depart from how the models were trained (due to zeroed-out embeddings) relative to the model’s original training setup.
>
>
> ---
> 2-2) _LoRA fine-tuning on COCO_. SD1.5 and SD2.0 are known to be trained on LAION-2B-en, a subset of LAION-5B, and further fine-tuned with aesthetic or preference datasets, often filtered using NSFW detectors. In contrast, the training data for SD3-m is not publicly disclosed. This lack of transparency makes it difficult to directly compare how differences in training data affect downstream performance. Therefore, to understand better if the domain gap is the issue preventing SD3 from discriminating well, we attempt to *align SD1.5, SD2.0, and SD3-m with the COCO domain*. We hypothesize that SD3 would see larger benefits relative to its original performance due to better alignment with the target domain (when testing on real-world datasets). For that, we run fine-tuning experiments on the training set of COCO using LoRA to enable efficient adaptation while minimizing extensive changes to the base model.
>
> For SD1.5 and SD2.0, we used rank 64, whereas for SD3-m (due to its larger size), we used rank 32 and extended training by 5000 additional steps.
>
> We then evaluated performance on compositional benchmarks whose image domains are derived from COCO. **We observed a notable accuracy improvement for all models, with SD3-m showing the largest relative gain. This supports our Finding 2, highlighting that SD3-m suffers from a larger domain gap vs. real-world benchmarks compared to SD1.5 and SD2.0.**
>
> The table below reports text-to-image retrieval accuracy (higher is better), along with the Δ (delta) indicating accuracy gains or drops after LoRA fine-tuning.
>
> | Benchmarks \ Models | SD1.5 ($\Delta$) | SD2.0 ($\Delta$) | SD3-m ($\Delta$) |
> | :---- | :---- | :---- | :---- |
>  | COCO order | 0.33 (+0.10)| 0.35 (+0.10) | 0.27 (**+0.11**) |
> | COCO QA one | 0.45 (-0.03) | 0.42 (+0.00)| 0.58 (**+0.04**) |
> | COCO QA two | 0.53 (-0.01) | 0.50  (+0.03)  | 0.62 (**+0.07**)| 0.63
>
> &nbsp;&nbsp;
>
> ### **Typo**
>
> Thank you for catching this! We will fix it in post.

---

> > ### Comment · Reviewer_RGqz · 2025-08-07
> >
> > I apologize for the delayed response due to a misunderstanding of the deadline. I appreciate the authors' thorough and thoughtful rebuttal. The clarifications have addressed most of my concerns, and I have no follow-up questions. I remain positive about the paper and its contributions.

---

> > > ### Author Response · Authors · 2025-08-07
> > >
> > > Thank you for maintaining the positive rating! If any further questions or concerns arise, we would be happy to discuss them.

---

> ### Author Response · Authors · 2025-08-06
>
> Dear Reviewer, thank you again for your thoughtful review! If you have any remaining questions or concerns after reading our rebuttal, we’d really appreciate the opportunity to discuss them further. Thanks!

---

### Comment · Area_Chair_KgD6 · 2025-08-03

Dear reviewers,

Thank you for agreeing to review at NeurIPS 2025. This is a friendly reminder that the authors have submitted their rebuttal and we are now in author-discussion phase until **August 6th AoE**.

Please **acknowledge reading the rebuttal** if you have not done it yet and update your score accordingly. Keep the deadline in mind when asking the authors for further clarification so they have **enough time to reply**.

Thank you,

AC

---

### Note · Authors · 2025-08-12

We thank the reviewers again for their constructive feedback, which has helped us strengthen both the clarity and scope of our work. In our rebuttal, we have:
- Clarified that Self-Bench is a diagnostic benchmark to isolate domain effects, not a replacement for real-world datasets; it is easily extendable to new categories, models, and prompts.
- Expanded on generative vs. discriminative mismatch, showed that SD 2.0 produces more varied samples than SD 3-m, which helps explain its stronger cross-domain performance.
- Added SDXL, SigLIP/SigLIP2, SD 3.5-m, REPA, and FLUX; main conclusions remain unchanged
- Ran text encoder alignment and LoRA fine-tuning, showing SD 3-m’s larger domain gap vs. SD 1.5/2.0.
- Provided error analysis linking poor performance to generative limitations and dataset biases in counting and position settings.
- Extended tests on SD 3-m with log-normal sampling and timestep shifts: gains stem from model-specific handling of timesteps, not only training-time schedules.
- Specified how timesteps are sampled in Eq. 1 and discussed relevant prior work on non-uniform timestep strategies.
- Clarified that CFG does not improve classification, consistent with prior findings.

These additions strengthen our main takeaway: diffusion classifiers can show compositional understanding, but performance depends on domain alignment, diversity, and timestep strategy.

We believe our rebuttal substantially addresses the reviewers’ concerns and that our work provides timely, broadly applicable insights for the community, setting a foundation for future research in this area.

---

### Decision · Program_Chairs · 2025-09-18

**Decision:**

Accept (poster)

**Comment:**

This work introduces a study of the compositional understanding abilities of diffusion models when used as zero-shot classifiers. Concretely, they cover three diffusion models (SD 1.5, 2.0, and, for the first time, 3-m) spanning 10 datasets and over 30 tasks, and they generate the images with the diffusion models themselves. The hypothesis is that if models can generate in a compositional manner, they should also be able to understand what they generate (in a zero-shot manner). Their three main findings are that i) and ii) models seem to "understand" what they create but they are outperformed by CLIP. iii) Performance drops when tested with out-domain samples (not self-generated). However the performance loss can be recovered by tuning the noise schedule to make the classifier more sensitive to the kind of compositional task.

Overall this is a good work that contains both a benchmark and interesting insights that can further our understanding of generative models. Reviews are positive (5,4,4,5) and **I support its acceptance**.

During discussion reviewers o8Cg, RGqz, HGDo agreed with this assessment.

Comment to the authors:

The caption of Figure 4 contains a typo (compositioanal).